# EMPIRICAL PRIORS FOR BAYESIAN NEURAL NETWORKS VIA WEIGHT PRUNING

## ABSTRACT

Designing informative priors for Bayesian neural networks remains a fundamental challenge, yet it plays a crucial role in determining both model performance and robustness. While numerous studies have explored effective strategies for prior selection, the suitable choice of prior distribution is often architecture-dependent and requires extensive time to determine. To address this, we propose *Sparsity-Informed priors for Bayesian neural networks*—SPIN—a *simple* method that empirically determines both the prior mean and variance of Gaussian priors: weights that survive pruning are considered important and are assigned low-variance Gaussian priors centered at their post-pruning values, while pruned weights are treated as less informative and given high-variance, zero-mean Gaussian priors. Our empirical results demonstrate that SPIN enhances performance across diverse architectures and datasets. Furthermore, we discuss why this prior design contributes to improved performance in Bayesian neural networks.

## 1 INTRODUCTION

Bayesian neural networks (BNNs) aim to model the posterior distribution over network parameters given training data (Neal & MacKay, 1992; Blundell et al., 2015; Gal et al., 2016; Jospin et al., 2022). Instead of training a single deterministic model, BNNs maintain a distribution over weights (Graves, 2011), thereby representing an *ensemble of networks* where each sample corresponds to a different network of the ensemble (Wilson & Izmailov, 2020).

A key component in BNN is the *specification of the prior distribution* $p(\mathbf{w})$, which encodes our *beliefs* about the parameters. Several prior formulations have been proposed in the literature, such as *isotropic Gaussian* (Hernández-Lobato & Adams, 2015; Louizos & Welling, 2017; Dusenberry et al., 2020; Wenzel et al., 2020), *correlated Gaussian* (Fortuin et al., 2021), and *heavy-tailed* distributions (e.g., *Laplace*, *Student-t*) (Fortuin, 2022; Hafner et al., 2020). However, selecting a meaningful prior is challenging: it is often highly *dependent on the network architecture* (Fortuin et al., 2021), *difficult and computationally expensive to find empirically* (Louizos & Welling, 2017), and *sometimes fails to provide good performance* (Wenzel et al., 2020).

To address the challenges of prior selection in BNN, we propose—SPIN[1] (**SP**arsity-**I**nformed priors for Bayesian **N**eural networks)—a simple yet effective approach that leverages network pruning to construct informative, data-driven priors. Pruning techniques remove redundant weights from a neural network, significantly reducing model size while often *preserving* (Han et al., 2015; Liu et al., 2018)—or *even enhancing* (Solla et al., 1990; You et al., 2019; Louizos et al., 2017)—*generalization performance*. Crucially, pruning *implicitly reveals the relative importance of individual weights* (Frankle & Carbin, 2019; Hoefler et al., 2021): those that survive are likely essential for generalization, while pruned weights tend to be less informative.

Inspired by these observations, we design priors that reflect the pruning-induced importance of weights: *low-variance priors are assigned to important (unpruned) weights, while high-variance priors are assigned to less critical (pruned) ones*. As illustrated in Fig. 1, where the pruning step enforces a binary mask over weights, and the resulting importance structure (*Sparse DNN*) is transferred into a prior distribution $p(\mathbf{w})$ of BNN. We evaluate our pruning-based prior across various architectures—including *convolutional neural networks* (CNNs) (He et al., 2016; Huang et al., 2017)

---

[1]The implementation is available `https://anonymous.4open.science/r/anonymous-50BE/`

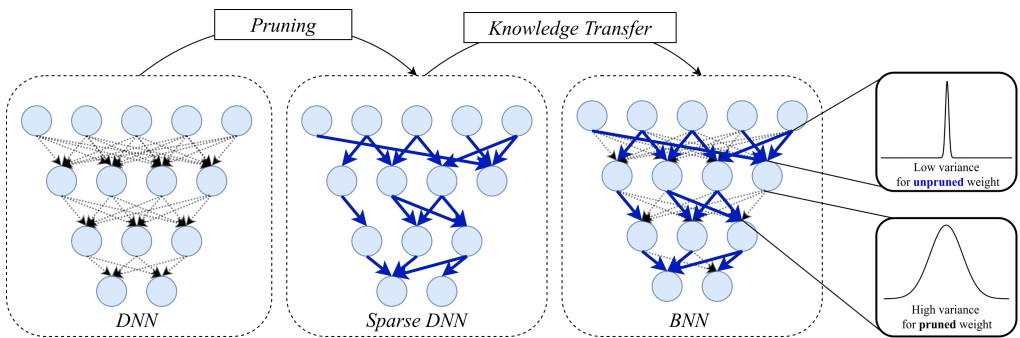

Figure 1: Overview of SPIN. We first train a baseline deterministic Deep Neural Network (DNN) (left), and apply global unstructured pruning to remove unimportant weights, yielding a *Sparse DNN*[2] (middle). We then construct the informative prior for the *BNN* by leveraging the *Sparse DNN* (right). The surviving weights, represented by **blue** solid lines, are used to form a high-confidence (low-variance) prior, while **pruned** weights follow a low-confidence prior (high-variance).

and *vision transformer* (ViT) (Dosovitskiy et al., 2020)—on *various benchmark datasets*. Overall, our results highlight SPIN as a *simple yet powerful* prior design that consistently surpasses existing priors across diverse architectures and datasets.

## 2 BACKGROUND

BNNs aim to model the full predictive distribution

$$p(\mathbf{y} \mid \mathbf{x}, \mathcal{D}) = \int p(\mathbf{y} \mid \mathbf{x}, \mathbf{w}) \, p(\mathbf{w} \mid \mathcal{D}) \, d\mathbf{w},$$

where $\mathcal{D} = \{(\mathbf{x}_i, \mathbf{y}_i)\}_{i=1}^{N}$ denotes the training dataset and $p(\mathbf{w} \mid \mathcal{D})$ is the posterior distribution over weights. Unlike traditional deep neural networks (DNNs) that yield point estimates $f(\mathbf{x}; \hat{\mathbf{w}})$, BNNs capture epistemic uncertainty through weight distributions (Neal, 2012; Gal & Ghahramani, 2016).

By Bayes' rule, the posterior is given as:

$$p(\mathbf{w} \mid \mathcal{D}) = \frac{p(\mathcal{D} \mid \mathbf{w}) \, p(\mathbf{w})}{p(\mathcal{D})}, \quad \text{where} \quad p(\mathcal{D}) = \int p(\mathcal{D} \mid \mathbf{w}) \, p(\mathbf{w}) \, d\mathbf{w}.$$

However, the marginal likelihood $p(\mathcal{D})$ is generally intractable, rendering exact Bayesian inference impractical for deep models (MacKay, 1992; Blundell et al., 2015). To approximate the posterior, two primary inference strategies have emerged: **Markov Chain Monte Carlo (MCMC)** (Neal, 2012) and **Variational Inference (VI)** (Blundell et al., 2015; Gal & Ghahramani, 2016; Krishnan et al., 2020). MCMC methods, such as Hamiltonian Monte Carlo (Duane et al., 1987; Hoffman et al., 2014; Chen et al., 2014), generate asymptotically exact samples from the true posterior $p(\mathbf{w} \mid \mathcal{D})$, but are computationally prohibitive and poorly scalable to modern deep architectures. In contrast, VI introduces a tractable family of distributions $q_\phi(\mathbf{w})$ to approximate the posterior by minimizing the Kullback-Leibler (KL) divergence:

$$q_\phi^*(\mathbf{w}) = \arg\min_{q_\phi} D_{\mathrm{KL}}\big(q_\phi(\mathbf{w}) \,\|\, p(\mathbf{w} \mid \mathcal{D})\big).$$

This is equivalent to maximizing the *Evidence Lower Bound* (ELBO):

$$\mathcal{L}_{\mathrm{ELBO}}(\phi) = \mathbb{E}_{q_\phi(\mathbf{w})}\big[\log p(\mathcal{D} \mid \mathbf{w})\big] - D_{\mathrm{KL}}\big(q_\phi(\mathbf{w}) \,\|\, p(\mathbf{w})\big).$$

Both approaches bypass the intractability of $p(\mathcal{D})$, but hinge critically on the choice of prior $p(\mathbf{w})$ (Sam et al., 2024; Cui et al., 2022; Tran et al., 2022; Shen et al., 2021; Fortuin, 2022). In this work, we address this challenge by introducing a pruning-induced prior that encodes the structural importance of weights, enabling more effective variational inference in BNN.

---

[2]We refer to a *Sparse DNN* as a deterministic network obtained by pruning a baseline DNN, where a substantial number of weights are set exactly to zero while preserving the original layer topology.

## 3 RELATED WORK

Foundational methods for training BNNs, such as Bayes by Backprop (Blundell et al., 2015), often rely on simple *isotropic Gaussian* priors. However, this choice is based on the unrealistic assumption that all weights are independent and equally important (Fortuin, 2022). To overcome this, subsequent work has explored priors that better reflect the varying importance of weights. These include heavy-tailed distributions like the *Laplace* or *Student-t* (Fortuin, 2022), as well as explicit sparsity-inducing priors like the *Spike-and-Slab* and *horseshoe* (Jantre et al., 2025; Sun et al., 2022; Ghosh et al., 2019; Carvalho et al., 2010). Both approaches permit a few weights to attain large magnitudes while shrinking the rest, thereby better reflecting the sparse structure of salient parameters in overparameterized networks. Despite these advances, specifying a prior that accurately captures this importance hierarchy remains a central challenge (Krishnan et al., 2020; Fortuin, 2022).

A powerful empirical tool for identifying important weights is network pruning. Beyond simply reducing model size, pruning reveals a fundamental hierarchy of parameter importance (LeCun et al., 1989), a principle articulated in the well-known *lottery ticket hypothesis* (Frankle & Carbin, 2019). This hypothesis posits that dense networks contain sparse subnetworks (*winning tickets*) that are critical for achieving high performance. To identify these subnetworks, pruning methodologies are generally categorized into structured and unstructured approaches. Structured pruning eliminates coherent units, such as entire filters or neurons. While this coarse-grained removal facilitates hardware acceleration, it risks severing critical connections embedded within pruned structures. In contrast, unstructured pruning targets individual weights based on fine-grained saliency criteria, such as magnitude (Han et al., 2015) or gradient sensitivity (Liu et al., 2023). Although the resulting irregular sparsity often necessitates specialized hardware for acceleration, this approach excels at isolating specific salient parameters, thereby preserving the intrinsic winning ticket structure with high fidelity.

Building on the high-fidelity weight identification of unstructured pruning, Our work bridges these two fields: BNN prior design and network pruning. Building on data-driven approaches that use pre-trained models to inform priors (Krishnan et al., 2020), we leverage the insights from the lottery ticket hypothesis. Our key contribution is a framework that uses an empirically identified sparse, high-performing subnetwork to define a more targeted and informative prior. Unlike priors that regard every weight as equally important, this approach encodes the network's importance structure into the prior beliefs, leading to substantially improved performance.

## 4 METHODOLOGY

The prior distribution $p(\mathbf{w})$ shapes the posterior and predictive behavior of BNNs. Existing works often adopt homogenous priors (e.g., *isotropic Gaussian*, and *Laplace*), overlooking the heterogeneous importance of different weights. To address this, in this section, we introduce **SP**arsity-**I**nformed priors for Bayesian **N**eural networks (SPIN), a framework that identifies the varying importance of weights and embeds this knowledge into the BNN prior. Let $\mathcal{W} = \{w_j\}_{j=1}^d$ denote the set of all weights in the network assuming an i.i.d. prior:

$$p(\mathbf{w}) = \prod_{j=1}^{d} \mathcal{N}(w_j \mid 0, \sigma^2).$$

We introduce an inductive bias that reflects the network's varying parameter importance by decomposing the weight space $\mathcal{W}$ into two disjoint subsets:

$$\mathcal{W} = \mathcal{W}_{\text{important}} \cup \mathcal{W}_{\text{redundant}}.$$

To identify $\mathcal{W}_{\text{important}}$, we leverage a pretrained DNN that shares the same architecture as our target BNN. We employ iterative global unstructured pruning and subsequent fine-tuning to identify unimportant connections while preserving performance. The resulting pruned model, referred to as a *Sparse DNN*, contains the set of unpruned weights $\mathbf{w}^* \in \mathcal{W}_{\text{important}}$, which are treated as high-confidence parameters. Conversely, the pruned weights form $\mathcal{W}_{\text{redundant}}$, representing connections deemed non-essential for predictive performance and, therefore, assigned a weakly informative

---

**Algorithm 1** **SP**arsity-**I**nformed Priors for Bayesian **N**eural Networks (SPIN)

1: **Input:** Neural network architecture $f_\theta$, training dataset $\mathcal{D}$
2: **Output:** Hybrid prior $p(\mathbf{w})$ for a BNN
    /* Train and prune until the NLL stops recovering */
3: Train a deterministic DNN $f_{\theta_0}$ on $\mathcal{D}$ and record its validation NLL $L_0$.
4: Iteratively prune and fine-tune $\theta_0$ until NLL exceeds $L_0$; let $\theta^\star$ be the last sparse weights.
    /* Extract important and redundant weights from final model */
5: Let $\mathcal{W}_{\text{important}} = \{\, w_j^\star \neq 0 : w_j^\star \in \theta^\star \,\}$ and $\mathcal{W}_{\text{redundant}} = \theta_0 \setminus \mathcal{W}_{\text{important}}$.
    /* Turn this mask into a hybrid prior over all weights */
6: For each $w_j \in \theta_0$, set $p(w_j) = \mathcal{N}(w_j \mid w_j^\star, \sigma^2)$ if $w_j^\star \in \mathcal{W}_{\text{important}}$, and $p(w_j) = \mathcal{N}(w_j \mid 0, 1)$ if $w_j^\star \in \mathcal{W}_{\text{redundant}}$.
7: **return** $p(\mathbf{w}) = \prod_j p(w_j)$

---

prior. We then define a *hybrid prior*:

$$
p(w_j) = \begin{cases} \mathcal{N}(w_j \mid w_j^*, \sigma^2), & \text{if } w_j \in \mathcal{W}_{\text{important}} \\ \mathcal{N}(w_j \mid 0, 1), & \text{if } w_j \in \mathcal{W}_{\text{redundant}} \end{cases}.
$$

Here, the first component encodes a data-driven informative prior centered at the pretrained value $w_j^*$, with a small variance $\sigma^2 \ll 1$ to reflect strong prior belief. The second component is a standard Gaussian prior to softly regularizing less critical parameters. The design principle of the SPIN prior aligns with *soft inductive bias* (Wilson, 2025), which do not restrict the hypothesis space but rather exploit the full parameter space to improve generalization in overparameterized deep neural networks. A brief overview of the algorithm is shown in Algorithm 1, and the full procedure is described in Algorithm 2.

This framework, SPIN, can be categorized as an *empirical Bayes* strategy, where prior belief is *informed by evidence* (i.e., weight importance revealed via pruning). To integrate this prior, we use VI to approximate the BNN's intractable posterior. This approach allows the model to anchor its beliefs on reliable knowledge derived from a pretrained model, while the VI process fine-tunes these beliefs based on data-driven evidence. Specifically, we employ a fully-factorized Gaussian variational posterior, $q_\phi(\mathbf{w}) = \prod_{j=1}^d \mathcal{N}(w_j \mid \mu_j, \sigma_j^2)$, adhering to the mean-field assumption. The parameters $\phi = \{(\mu_j, \sigma_j^2)\}_{j=1}^d$ of this variational distribution are optimized by maximizing the ELBO, $\mathcal{L}_{ELBO}(\phi)$. This objective function creates a lower bound on the log marginal likelihood and is composed of two principal terms: an expected log-likelihood term that measures data fidelity and a KL divergence term that acts as a regularizer. The general form of the ELBO is:

$$
\mathcal{L}_{\text{ELBO}}(\phi) = \mathbb{E}_{q_\phi(\mathbf{w})}\big[\log p(\mathcal{D} \mid \mathbf{w})\big] - D_{\text{KL}}\big(q_\phi(\mathbf{w}) \,\|\, p(\mathbf{w})\big).
$$

The crucial part of this objective, where our SPIN prior exerts its influence, is the KL divergence term. This term decomposes into a sum over the individual weights, which we can split according to our partitioning of the weight space:

$$
D_{KL}(q_\phi(\mathbf{w}) \| p(\mathbf{w})) = \sum_{w_j \in \mathcal{W}_{\text{important}}} D_{KL}(q_\phi(w_j) \| p(w_j)) + \sum_{w_j \in \mathcal{W}_{\text{redundant}}} D_{KL}(q_\phi(w_j) \| p(w_j)).
$$

For the **important weights** ($w_j \in \mathcal{W}_{\text{important}}$), the KL term penalizes deviations from their prior means $w_j^*$:

$$
D_{KL}(q_\phi(w_j) \| p(w_j)) = \log \frac{\sigma}{\sigma_j} + \frac{\sigma_j^2 + (\mu_j - w_j^*)^2}{2\sigma^2} - \frac{1}{2}.
$$

For the **redundant weights** ($w_j \in \mathcal{W}_{\text{redundant}}$), the KL term acts as a standard regularizer, encouraging them towards zero:

$$
D_{KL}(q_\phi(w_j) \| p(w_j)) = \log \frac{1}{\sigma_j} + \frac{\sigma_j^2 + \mu_j^2}{2} - \frac{1}{2}.
$$

Combining these elements gives us the final objective function. Maximizing this ELBO encourages the BNN to preserve the knowledge encoded in $\mathcal{W}_{\text{important}}$ while allowing for uncertainty and adaptation, primarily through the less critical weights in $\mathcal{W}_{\text{redundant}}$.

## 5 EXPERIMENTS

We evaluate our method on **CIFAR-10**, **CIFAR-100** (Krizhevsky et al., 2009), and **TinyImageNet** (Wu et al., 2017) using several architectures, including **ResNet-18, ResNet-20** (He et al., 2016), **DenseNet-30** (Huang et al., 2017), **MobileNet-v2** (Sandler et al., 2018), and a lightweight **ViT** (Dosovitskiy et al., 2020). For out-of-distribution detection, models trained on **CIFAR** datasets are tested against **TinyImageNet**, and vice versa. Further implementation details, including the specific **ViT** architecture and training procedure, are provided in Appendix A. Our approach is compared against several baseline priors: *Isotropic Gaussian*, *Laplace*, *Student-t*, *Spike-and-slab*, *Horseshoe* and *MOPED* (Krishnan et al., 2020). The SPIN used in the experiment, the `sparsity` is determined for each model via an iterative global magnitude-based pruning that finds *the highest* `sparsity` that preserves the original validation performance as described in Algorithm 1. BNNs are trained using the Bayes by Backprop (Blundell et al., 2015). For the prior variance of the unpruned weights, we set $\sigma = 0.001$ across all experiments. We also present a detailed ablation study on the effect of different $\sigma$ values under various `sparsity` levels in Appendix B. Additionally, Section 6.2 provides intuition and empirical evidence for why $\sigma$ should be chosen small, while Appendix E offers a complementary theoretical justification using a simplified linear–Gaussian model, showing that a smaller $\sigma$ leads to lower expected excess test error. To further demonstrate the broad applicability of our approach, Appendix C presents additional experiments on classification, regression, and semantic segmentation. For clarity, the main text reports results on **CIFAR-10** and **CIFAR-100** using **ResNet-20** and **ViT**; due to page limitations, the full experimental results—including those for **TinyImageNet** and other model architectures (e.g., **ResNet-18**, **MobileNet-v2**, and **DenseNet-30**)—are summarized in Appendix B. All results reported in the main text are averaged over 10 independent random seeds. The best result in each column is highlighted in **bold**, and the second-best is underlined.

**Results** The comprehensive performance evaluation of our proposed prior against several baselines is presented in Table 1 for **CIFAR-10** and **CIFAR-100**. We assess performance across three key areas: predictive performance (Accuracy and NLL), model calibration (expected calibration error, ECE), and **out-of-distribution (OOD) detection** entropy (ENT). The value of the maximum `sparsity` level that our method SPIN uses is noted in Appendix B.

**Overall Performance** Across both datasets and architectures, SPIN achieves consistently strong predictive performance, particularly on convolutional models. For instance, SPIN improves ResNet-20 accuracy to 90.43% on CIFAR-10 and 66.96% on CIFAR-100, outperforming existing priors in both accuracy and NLL. Similar benefits are observed for ViT, including on **CIFAR-10**, where SPIN reaches 74.48%. These results highlight the value of leveraging pruning-derived structural information in prior construction. Furthermore, the performance advantage extends to the larger-scale **TinyImageNet** dataset (Appendix B), and also translates to stronger OOD detection, where SPIN generally achieves the highest AUROC OOD score.

**Comparison With Baseline Priors** Standard priors like the *Isotropic Gaussian*, and heavy-tailed distributions such as *Laplace* and *Student-t*, provide stable baseline performances but are fundamentally limited as they treat all weights uniformly, ignoring any underlying importance structure. Furthermore, although sparsity-inducing priors such as *Spike-and-Slab* and *Horseshoe* theoretically allow for heterogeneous weight treatment, they often underperform compared to SPIN in our experiments. In particular, the Horseshoe prior exhibits training instability, leading to high variance across runs and degraded predictive performance, whereas SPIN yields consistently stable and reliable results. We attribute this performance gap to the optimization challenges inherent in jointly learning weight importance and data fit. In contrast, SPIN effectively circumvents this issue by explicitly decoupling structure identification via pruning from the BNN training process.

**Calibration and the NLL/ECE Trade-off** Although our method excels in both accuracy and NLL, *it does not yield the best ECE*. This appears to stem from a known tendency that achieving lower NLL often makes more confident predictions, which can slightly harm calibration. This behavior is not unusual; fortunately, this calibration issue can be effectively mitigated through *temperature scaling* (Guo et al., 2017), which adjusts the confidence without degrading accuracy and NLL as shown in Appendix D.5.

Table 1: Evaluation of Prior Distributions for BNNs on CIFAR-10 (C-10) and CIFAR-100 (C-100)

| Model | Prior | Acc (↑) C-10 / C-100 | NLL (↓) C-10 / C-100 | ECE (↓) C-10 / C-100 | OOD (↑) C-10 / C-100 |
|---|---|---|---|---|---|
| ResNet 20 | Iso. Gaussian | 0.8922 / 0.6341 | 0.3402 / 1.3600 | 0.0275 / 0.0275 | 0.8564 / 0.7365 |
| | Laplace | 0.8931 / 0.6320 | 0.3374 / 1.3616 | 0.0321 / 0.0258 | 0.8561 / 0.7383 |
| | Student-t | 0.8932 / 0.6299 | 0.3400 / 1.3700 | 0.0282 / 0.0277 | 0.8578 / 0.7357 |
| | Spike-and-Slab | 0.8917 / 0.6326 | 0.3461 / 1.3626 | 0.0243 / 0.0250 | 0.8556 / 0.7357 |
| | Horseshoe | 0.8916 / 0.6292 | 0.3248 / 1.3556 | 0.0610 / 0.0428 | 0.8602 / 0.7363 |
| | MOPED | 0.8907 / 0.6298 | 0.3442 / 1.3668 | 0.0283 / 0.0269 | 0.8556 / 0.7340 |
| | SPIN$_{@maximum}$ | 0.9043 / 0.6696 | 0.3099 / 1.3534 | 0.0305 / 0.0203 | 0.8641 / 0.7569 |
| ViT | Iso. Gaussian | 0.6456 / 0.4997 | 0.9881 / 1.9502 | 0.0219 / 0.0273 | 0.7032 / 0.6733 |
| | Laplace | 0.6438 / 0.4987 | 0.9936 / 1.9517 | 0.0211 / 0.0248 | 0.7024 / 0.6727 |
| | Student-t | 0.6425 / 0.4884 | 0.9983 / 1.9872 | 0.0252 / 0.0246 | 0.7019 / 0.6670 |
| | Spike-and-Slab | 0.6444 / 0.4983 | 0.9958 / 1.9495 | 0.0243 / 0.0262 | 0.7021 / 0.6716 |
| | Horseshoe | 0.5926 / 0.4995 | 1.1060 / 1.9374 | 0.0278 / 0.0217 | 0.6850 / 0.6700 |
| | MOPED | 0.5936 / 0.5008 | 1.1271 / 1.9348 | 0.0251 / 0.0272 | 0.6772 / 0.6707 |
| | SPIN$_{@maximum}$ | 0.7448 / 0.5022 | 0.7197 / 1.9384 | 0.0189 / 0.0364 | 0.7485 / 0.6730 |

Table 2: Performance evaluation across pruning criteria on CIFAR-10

| Model | Criterion | Acc(↑) | NLL(↓) | ECE(↓) | OOD(↑) |
|---|---|---|---|---|---|
| ResNet 20 | Global L1 | 0.9043 ± 0.0031 | 0.3099 ± 0.0075 | 0.0305 ± 0.0022 | 0.8641 ± 0.0033 |
| | Gradient ($|w \odot g|$) | 0.9035 ± 0.0020 | 0.3170 ± 0.0079 | 0.0235 ± 0.0037 | 0.8647 ± 0.0018 |
| | Random | 0.8894 ± 0.0031 | 0.3499 ± 0.0083 | 0.0308 ± 0.0022 | 0.8561 ± 0.0038 |
| | Structured L2 | 0.8702 ± 0.0042 | 0.3976 ± 0.0071 | 0.0408 ± 0.0084 | 0.8380 ± 0.0035 |
| ViT | Global L1 | 0.7448 ± 0.0117 | 0.7197 ± 0.0297 | 0.0189 ± 0.0019 | 0.7485 ± 0.0069 |
| | Gradient ($|w \odot g|$) | 0.7453 ± 0.0054 | 0.7188 ± 0.0094 | 0.0232 ± 0.0024 | 0.7507 ± 0.0021 |
| | Random | 0.7381 ± 0.0059 | 0.7382 ± 0.0172 | 0.0193 ± 0.0026 | 0.7495 ± 0.0032 |
| | Structured L2 | 0.7190 ± 0.0063 | 0.7822 ± 0.0141 | 0.0228 ± 0.0018 | 0.7381 ± 0.0025 |

**The Case of ViT and the Importance of DNN Quality**   The results on **ViT** highlight the critical role of pretrained DNN quality in designing effective priors. Unlike CNNs, ViTs lack strong inductive biases, which makes their performance highly sensitive to prior choice, as reflected in the large variation across baselines in Table 1. On **CIFAR-10**, where the pretrained model is reasonably strong, SPIN yields moderate but consistent improvements over standard priors, demonstrating that pruning-derived structure can serve as a useful inductive bias. However, when the base DNN is weak or underfitting, such as on **CIFAR-100** and **TinyImageNet**, the sparse subnetworks extracted by pruning may fail to retain essential representation capacity, resulting in limited benefit from SPIN. This effect is clearly observed for **ViT**, where all priors—including SPIN—achieve similar performance on these more challenging datasets (see Table B-3). These observations reveal an important practical limitation: *the effectiveness of SPIN depends strongly on the quality of the pretrained DNN prior to pruning*. To mitigate this issue in capacity-limited regimes (e.g., **ViT/TinyImageNet**), we explore lightweight architectural adjustments and training refinements, with results presented in Appendix F.2. Additional discussion on generalization behavior and robustness trade-offs appears in Appendix D-7 and D-8.

## 6  ANALYSIS

In this section, we provide ablation studies on pruning sensitivity, posterior behavior, computational cost, and slab variants, demonstrating that heavy-tailed alternatives offer limited gains once salient weights are identified.

### 6.1  SENSITIVITY TO PRUNING CRITERIA

We conducted an ablation study to examine the sensitivity of SPIN to different pruning criteria, and the results are summarized in Table 2. Across both ResNet-20 and ViT, we consistently observe the ordering *Gradient ≈ Global L1 > Random > Structured L2*. High-quality pruning criteria (i.e.,*Gradient* and *Global L1*) achieve the strongest results across accuracy, NLL, ECE, and OOD AUROC, indicating that SPIN effectively leverages structurally meaningful masks that accurately isolate salient weights. *Random* pruning performs worse but remains superior to *Structured L2*, as

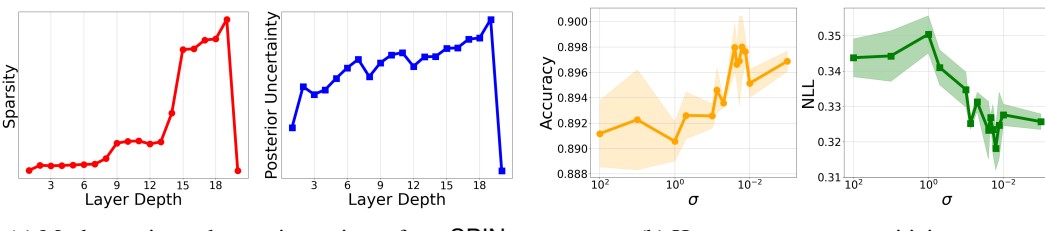

(a) Mask sparsity and posterior variance from SPIN      (b) Hyperparameter $\sigma$ sensitivity

Figure 2: **(a)** Layer-wise sparsity of the mask from **ResNet-20** (left; sparsity = number of zero weights per layer) and the corresponding posterior uncertainty induced from SPIN (right). **(b)** Performance of SPIN as the hyperparameter $\sigma$ varies.

Table 3: Representation quality of **ResNet-20** on **CIFAR-10**, evaluated using the *Davies–Bouldin Index* (lower is better). **Bold** indicates the best and underline denotes the second-best.

| isotropic Gaussian | Laplace | student-t | MOPED | spike-and-slab | Ours (0%) | Ours (50%) | Ours (70%) |
|---|---|---|---|---|---|---|---|
| 1.26 | 0.97 | 1.33 | 1.64 | 1.21 | 2.09 | 1.03 | **0.90** |

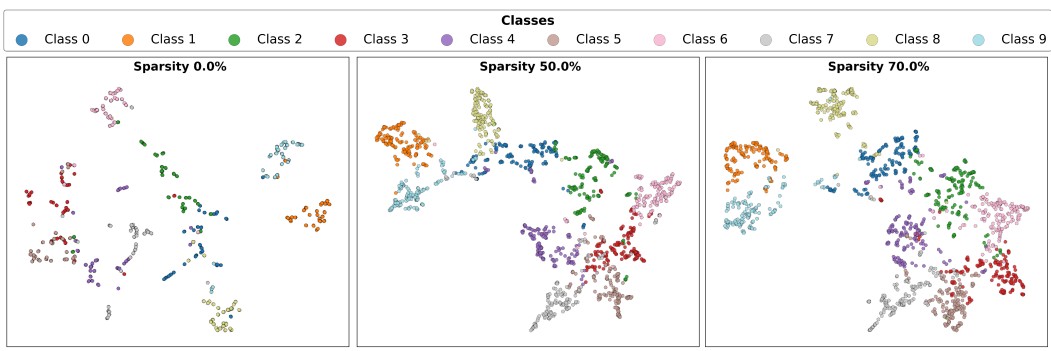

Figure 3: *UMAP* (McInnes et al., 2018) visualization of **CIFAR-10** features extracted by **ResNet-20** under varying `sparsity` levels. The left panel corresponds to **0%** sparsity, the middle panel to **50%** sparsity, and the right panel to **70%** sparsity.

it preserves distributed connectivity and functional pathways despite lacking principled ranking. In contrast, *Structured L2* pruning exhibits the most severe degradation due to coarse removal of entire channels or filters, which can overshoot and eliminate critical computation paths. *Overall, these findings demonstrate that mask correctness and pruning granularity are central determinants of SPIN performance, supporting the use of fine-grained pruning when constructing the sparse network used for prior modeling.*

## 6.2 UNDERSTANDING THE MECHANISMS BEHIND SPIN

**Layer-Wise Structural Patterns in Masks from Pruning** We investigate why SPIN performs well in practice. Our analysis highlights an important property of pruning, which we call the **layer-wise structural patterns in pruning masks**. In **ResNet-20**, pruning consistently produces a *Dense → Sparse → Dense* pattern across depth: early layers remain dense (not many pruned weights), middle layers become highly sparse (many weights are pruned), and the final layers near the classifier again retain relatively high density (shown in Fig. 2 (a)). When this mask is used in SPIN with a small variance parameter (e.g., $\sigma = 0.001$), the prior naturally induces a *Low Uncertainty → High Uncertainty → Low Uncertainty hierarchy*: dense layers receive narrow priors, sparse layers receive wide priors, and the final dense layers return to narrow priors. As a result, the variational posterior inherits a similar hierarchical uncertainty structure, as illustrated in Fig. 2 (a). This behavior aligns with empirical observations from expensive MCMC sampling (Sommer et al., 2024) (see Fig. F-12) and empirical posterior weight distributions (Fortuin et al., 2021), suggesting that SPIN recovers a well-known uncertainty profile in a principled variational form utilizing the mask. More

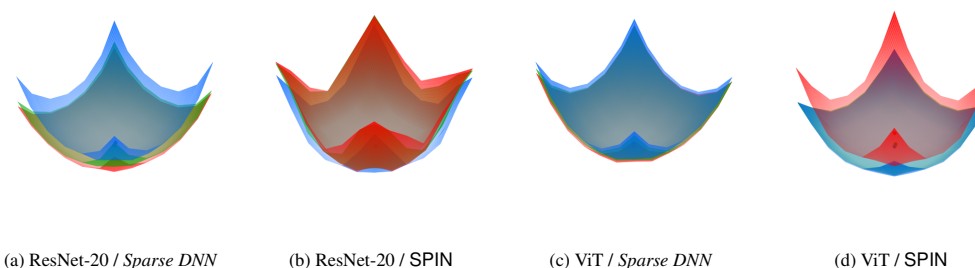

(a) ResNet-20 / *Sparse DNN*     (b) ResNet-20 / SPIN     (c) ViT / *Sparse DNN*     (d) ViT / SPIN

Figure 4: 3D plots of the NLL loss landscape. (a) and (c) correspond to the *Sparse DNN*, and (b) and (d) show the BNN from SPIN. Colors indicate different `sparsity` levels: red = 30%, green = 60%, and blue = 80% for (a) and (b); red = 0%, green = 20%, and blue = 40% for (c) and (d).

detailed explanations are reported in Appendix F. Again, SPIN mask is not arbitrary; *it captures the similar underlying structural importance that drives from high-cost MCMC exploration, but encodes it explicitly and more efficiently into the prior of BNN to enable scalable VI.*

**Choice of the Hyperparameter $\sigma$**  In SPIN, the most critical hyperparameter is $\sigma$. To understand its effect, we conducted an ablation study on **ResNet-20** trained on **CIFAR-10** while varying $\sigma$ over a wide range. We observed that, as long as $\sigma$ remains within a reasonably small range, the performance of SPIN is strong. In contrast, as $\sigma$ becomes larger, the performance of SPIN consistently degrades (illustrated in Fig. 2b). This behavior is well aligned with our intuition: assigning a small variance to important weights is desirable, whereas making this variance too large weakens the effect of the prior. Also, we analyze in a simplified linear model, further supporting this observation in Appenidx D.2 and E, showing that constructing the SPIN prior with $\sigma$ close to zero is preferable to using a very large one. Consequently, *the choice of $\sigma = 0.001$ used in the main experiments is not arbitrary; it is guided by both theoretical considerations and empirical evidence, and corresponds to the smallest value that still yields numerically stable gradient propagation during training.*

**Sparsity Regularizes the Representation of BNN**  We further analyze SPIN with varying `sparsity` on **CIFAR-10** with **ResNet-20**. For each class, we randomly sample 30 data points, perform inference with 5 *Monte Carlo* samples per point, and visualize the learned feature vectors using *UMAP* (McInnes et al., 2018). At 0% `sparsity`, all weights are treated as important, so the model behaves almost deterministically and yields low-diversity, nearly point-like feature clusters (Fig. 3 (left)). As `sparsity` increases, more of the weight space is freed by assigning large-variance priors to pruned weights, leading to richer and more diverse feature representations; this effect is most pronounced at 70% `sparsity` (Fig. 3 (right)), with 50% `sparsity` lying in between (Fig. 3 (middle)). *These results indicate that `sparsity` in SPIN effectively regularizes the BNN ensemble.* A quantitative comparison with other priors is given in Table 3, with additional results and experiment setting are in Appendix D.1.

**Comparison With Sparse DNN**  Previous work has suggested that pruning may improve generalization through an *implicit regularization effect* (Bartoldson et al., 2020). From this perspective, one might expect that a *Sparse DNN* already benefits from such regularization, and that using it as a prior in a BNN would automatically yield better performance. However, our results show that the performance of a BNN trained with SPIN does not exhibit a consistent correlation with the accuracy of the corresponding *Sparse DNN* (see Appendix D.4). Moreover, as `sparsity` increases, the loss landscape of the *sparse DNN* becomes noticeably sharper, as shown in Fig. 4(a) and (c). In contrast, when the same sparse structure is encoded into the BNN prior through SPIN, the resulting loss landscape becomes progressively flatter with higher `sparsity`, as seen in Fig. 4(b) and (d). *This indicates that the improvements observed in the BNNs from SPIN are not simply due to the sparse DNN already performing well.* Additional reports are included in Appendix D.3 and D.4.

**Comparison With Sub-Networks**  It is natural to question why SPIN assigns a high-variance prior to redundant weights instead of hard-pruning them to zero (effectively treating them as a Dirac delta distribution). To investigate this, we compare SPIN against a *Pruned BNN* baseline, where

Table 4: Pruned BNN vs. SPIN across sparsity levels

| Sparsity | 0% | 10% | 20% | 30% | 40% | 50% | 60% | 70% | 80% | 90% |
|---|---|---|---|---|---|---|---|---|---|---|
| Pruned BNN (Accuracy) | **0.8908** | **0.8890** | **0.8907** | 0.8897 | 0.8886 | 0.8853 | 0.8815 | 0.8758 | 0.8674 | 0.8430 |
| SPIN (Accuracy) | 0.8792 | 0.8815 | 0.8867 | **0.8928** | **0.8944** | **0.8988** | **0.8987** | **0.8994** | **0.8994** | **0.8970** |

Table 5: Training cost comparison across Bayesian priors measured in GPU-hours (RTX 3090×1)

| Metric | Gaussian | Laplace | Student-t | Spike-and-Slab | Horseshoe | SPIN |
|---|---|---|---|---|---|---|
| GPU-hours | **1.93** | 6.04 | 8.93 | 3.26 | 35.56 | 2.84 |

Table 6: Performance comparison of SPIN instantiated with alternative slab priors

| SPIN Variant | Acc($\uparrow$) | NLL($\downarrow$) | ECE($\downarrow$) | AUROC (ENT)($\uparrow$) |
|---|---|---|---|---|
| Gaussian (default) | 0.9035 ± 0.0020 | 0.3170 ± 0.0079 | **0.0235 ± 0.0037** | 0.8647 ± 0.0018 |
| Laplace | **0.9061 ± 0.0020** | **0.2995 ± 0.0056** | 0.0317 ± 0.0016 | **0.8655 ± 0.0022** |
| Student-t | 0.8867 ± 0.0013 | 0.3493 ± 0.0038 | 0.0333 ± 0.0036 | 0.8552 ± 0.0040 |
| Cauchy | 0.8891 ± 0.0020 | 0.3475 ± 0.0060 | 0.0338 ± 0.0043 | 0.8552 ± 0.0025 |
| Horseshoe | 0.8919 ± 0.0029 | 0.3253 ± 0.0060 | 0.0682 ± 0.0115 | 0.8612 ± 0.0020 |

redundant weights are strictly removed from the inference process. As shown in Table 4, SPIN consistently outperforms the Pruned BNN baseline, particularly in high-sparsity regimes ($\geq 30\%$). *This finding supports the advantage of **soft inductive biases (Wilson, 2025)**: retaining stochastic flexibility in redundant parameters allows for better generalization compared to the hard restriction bias of pruning.* Further details and results are provided in Appendix F.1.

## 6.3 COMPUTATIONAL COST ANALYSIS

We quantify computational burden in GPU-hours, as summarized in Table 5. All models were trained on a single NVIDIA RTX 3090 GPU under identical configurations. SPIN requires additional compute relative to a Gaussian-prior BNN due to dense pretraining and iterative pruning (10%→20%→30% sparsity), increasing total training cost from 1.93 to 2.84 GPU-hours ($\approx 1.5\times$). *Despite this overhead, SPIN remains substantially more efficient than heavy-tailed priors* such as Laplace (6.04), Student-t (8.93), and Horseshoe (35.56) GPU-hours.

## 6.4 ABLATION ON HEAVY-TAILED SLABS

We evaluate SPIN with alternative heavy-tailed slabs, including Laplace, Student-t, Cauchy, and Horseshoe priors. As shown in Table 6, the Gaussian and Laplace variants provide the best overall trade-off across Accuracy, NLL, ECE, and OOD AUROC, while the heavier-tailed options generally underperform. This is natural in our setting: since pruning has already produced a reliable split between important and redundant weights, we do not need additional heavy-tailed flexibility. Heavier-tailed slabs mainly introduce extra variance and optimization difficulty without clear benefits, while also incurring higher computational cost. *A simple Gaussian or Laplace slab, therefore, suffices for stable posterior refinement, and SPIN gains most of its advantage from the pruning-based structural information rather than from heavy-tailed Bayesian modeling.*

## 7 CONCLUSION

In this work, we proposed SPIN, a pruning-informed prior framework for BNN that turns a well-trained sparse DNN into an informative, data-driven prior. Our analysis shows that pruning naturally induces a layer-wise structure, which SPIN translates into a meaningful hierarchy of posterior uncertainty. We further demonstrated that increasing `sparsity` regularizes the BNN, yielding richer feature representations and consistent gains over subnetwork-based baselines. Although SPIN introduces modest additional computational cost compared to Gaussian-prior BNN, it remains far more efficient than heavy-tailed priors. Overall, these results indicate that leveraging pruning-based structural information offers a principled and scalable way to construct informative priors for BNN.

[†]This paper used assistance from large language models (Gemini, ChatGPT) for code, math, and writing. All content was reviewed and validated by the authors.

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
