# A IMPLEMENTATION DETAILS

In this section, we report our implementation details.

## A.1 DETERMINISTIC DEEP NEURAL NETWORKS AND PRUNING

Our implementation for training and pruning deterministic models is based on PyTorch. The specific architectures and data augmentation schemes varied by dataset as follows:

For the **CIFAR-10** and **CIFAR-100** datasets, we employed **ResNet-20**, **DenseNet-30**, and a vision transformer (**ViT**). The **ViT** architecture is adapted from the Hugging Face transformers library and configured with the following parameters: a patch size of 4, an embedding dimension of 192, 6 transformer layers, 3 attention heads, and a multi-layer perceptron (**MLP**) ratio of 2.0. For all models on these datasets, we applied standard data augmentation, including random cropping, random horizontal flipping, and normalization.

For the more complex **TinyImageNet** dataset, we found the models used for the **CIFAR** datasets had limited learning capacity. Consequently, we utilized larger architectures: **ResNet-18**, **MobileNet-v2**, and a larger **ViT**. For **ResNet-18** and **MobileNet-v2**, input images were upscaled to $224 \times 224$ pixels, after which the aforementioned augmentations (random crop, random horizontal flip, and normalization) were applied. For the ViT, we adopted the model architecture, training hyperparameters, and data augmentation directly from (Gani et al., 2022), configuring it with a patch size of 8, an embedding dimension of 192, 12 transformer layers, 3 attention heads, and an MLP ratio of 2.0.

With the exception of the **ViT** on **TinyImageNet**, all deterministic models are trained for 90 epochs using a batch size of 128. We use the SGD optimizer with a momentum of 0.9, a weight decay of 0.0001, and Nesterov momentum disabled. The learning rate is managed by a cosine annealing scheduler, starting from an initial value of 0.1 and gradually decaying to 0.001.

For the pruning process, we employ global magnitude-based unstructured pruning. The subsequent fine-tuning phase for the pruned models followed the same configuration as the initial DNN training (i.e., optimizer, epochs, and batch size). The only adjustment is to the learning rate scheduler, which was modified to start at a lower initial rate of 0.001 instead of 0.1. However, for the **ViT** on **Tiny-ImageNet**, the learning rate schedule for fine-tuning remained identical to its initial training phase as proposed (Gani et al., 2022).

## A.2 IMPLEMENTATION OF BAYESIAN NEURAL NETWORKS

The majority of our implementation is built upon an open-source library[3]. We evaluated several prior distributions for the weights and biases. For MOPED (Krishnan et al., 2020) and the *isotropic Gaussian prior*, we utilized the implementations provided by the referenced library. We additionally implemented and evaluated the *Laplace*, *Student-t*, *Spike-and-Slab* priors, and SPIN to extend the scope of our analysis. Variational inference (**VI**) is applied exclusively to the convolutional and fully-connected layers. Other components of the network, such as activation functions and batch normalization layers, remained deterministic. For training BNN, we disable weight decay since it conflicts with the probabilistic interpretation of the prior.

### A.2.1 KULLBACK-LEIBLER DIVERGENCE FORMULATIONS FOR THE ELBO

The evidence lower bound (ELBO) is the Kullback-Leibler (KL) divergence between the approximate posterior $q(w)$ and the prior $p(w)$. We employ a mean-field VI approach, where the posterior for the model's weights is approximated by a factorized distribution. Specifically, the posterior for each individual weight $w$ is modeled as an independent diagonal Gaussian distribution, $q(w) \sim \mathcal{N}(\mu_q, \sigma_q^2)$. Here, the mean $\mu_q$ and standard deviation $\sigma_q$ are the trainable variational parameters for each weight. The specific formulation of the KL divergence, $D_{\mathrm{KL}}(q(w) \,\|\, p(w))$, varies depending on the chosen prior as detailed below.

---

[3] https://github.com/IntelLabs/bayesian-torch

**Isotropic Gaussian Prior:** For a Gaussian prior $p(w) \sim \mathcal{N}(\mu_p, \sigma_p^2)$, the KL divergence has a well-known closed-form analytical solution:

$$D_{\text{KL}}(q(w) \,\|\, p(w)) = \log\left(\frac{\sigma_p}{\sigma_q}\right) + \frac{\sigma_q^2 + (\mu_q - \mu_p)^2}{2\sigma_p^2} - \frac{1}{2}.$$

**Laplace Prior:** For a Laplace prior $p(w) \sim \text{Laplace}(\mu_p, b_p)$, the KL divergence is also computed analytically. In our experiments, we used a standard Laplace prior, setting the location $\mu_p = 0$ and the scale $b_p = 1$. The general formula is:

$$D_{\text{KL}}(q(w) \,\|\, p(w)) = \log(2b_p) - \frac{1}{2}\log(2\pi\sigma_q^2) - \frac{1}{2} + \frac{1}{b_p}\mathbb{E}_{q(w)}[|w - \mu_p|],$$

where the expectation term $\mathbb{E}_{q(w)}[|w - \mu_p|]$ is calculated as:

$$\sigma_q\sqrt{\frac{2}{\pi}}\exp\left(-\frac{(\mu_q - \mu_p)^2}{2\sigma_q^2}\right) + (\mu_q - \mu_p)\left(1 - 2\Phi\left(-\frac{\mu_q - \mu_p}{\sigma_q}\right)\right).$$

Here, $\Phi(\cdot)$ denotes the cumulative distribution function (CDF) of the standard Normal distribution.

**Student-t Prior:** For the Student-t prior, the analytical KL divergence between the Gaussian posterior and the Student-t prior is intractable. Therefore, we estimate it using a Monte Carlo approximation.

First, the probability density function (PDF) of a general (non-standardized) Student's t-distribution is defined as:

$$p(w|\nu, \mu_p, \sigma_p) = \frac{\Gamma\left(\frac{\nu+1}{2}\right)}{\Gamma\left(\frac{\nu}{2}\right)\sqrt{\pi\nu}\sigma_p}\left(1 + \frac{1}{\nu}\left(\frac{w - \mu_p}{\sigma_p}\right)^2\right)^{-\frac{\nu+1}{2}},$$

where $\nu$ is the degrees of freedom, $\mu_p$ is the location, and $\sigma_p$ is the scale. In our experiments, the prior was configured as a standard Student's t-distribution with degrees of freedom $\nu = 1.0$ (i.e., a Cauchy distribution), location $\mu_p = 0$, and scale $\sigma_p = 1$.

The KL divergence is then approximated by drawing samples from the variational posterior $q(w)$:

$$D_{\text{KL}}(q(w) \,\|\, p(w)) \approx \frac{1}{L}\sum_{i=1}^{L}\left[\log q(w^{(i)}) - \log p(w^{(i)})\right] \quad \text{where} \quad w^{(i)} \sim \mathcal{N}(\mu_q, \sigma_q^2).$$

During training, this expectation was approximated using $L = 10$ samples. The term $\log p(w^{(i)})$ is calculated using the PDF of the Student's t-distribution defined above with the specified parameters.

**Spike-and-Slab Prior:** The Spike-and-Slab prior is a mixture model designed to encourage sparsity by explicitly modeling whether a weight is zero or non-zero. It is defined using a binary latent variable $z \in \{0, 1\}$.

The prior is specified hierarchically:

1. A Bernoulli distribution governs the selection of the spike or the slab:

$$p(z) = \text{Bernoulli}(z|\pi_p).$$

   where $\pi_p$ is a fixed hyperparameter representing the prior probability of a weight being non-zero (i.e., coming from the "slab").

2. The weight $w$ is drawn from a conditional distribution depending on $z$:
   - **Slab:** If $z = 1$, the weight is drawn from a Gaussian distribution: $p(w|z = 1) = \mathcal{N}(w|\mu_p, \sigma_p^2)$.
   - **Spike:** If $z = 0$, the weight is exactly zero, represented by a Dirac delta function: $p(w|z = 0) = \delta_0(w)$.

For inference, we use a structured variational approximation for the posterior that mirrors the prior's structure, $q(w, z) = q(z)q(w|z = 1)$. The posterior for the selection variable is $q(z) = \text{Bernoulli}(z|\alpha_q)$, where $\alpha_q$ is a learnable parameter representing the posterior inclusion probability. The posterior for the slab component is $q(w|z = 1) = \mathcal{N}(w|\mu_q, \sigma_q^2)$.

Under this formulation, the KL divergence between the joint posterior $q(w, z)$ and the joint prior $p(w, z)$ decomposes as follows:

$$D_{\text{KL}}(q(w, z) \,\|\, p(w, z)) = D_{\text{KL}}(\text{Bernoulli}(\alpha_q) \,\|\, \text{Bernoulli}(\pi_p)) + \alpha_q \cdot D_{\text{KL}}(\mathcal{N}(\mu_q, \sigma_q^2) \,\|\, \mathcal{N}(\mu_p, \sigma_p^2)).$$

The first term is the KL divergence between the Bernoulli distributions for the selection variables. The second term is the KL divergence for the slab components, weighted by the posterior inclusion probability $\alpha_q$. In our implementation, we encourage approximately 50% `sparsity` by setting the prior inclusion probability to $\pi_p = 0.5$ and using a standard normal distribution for the slab, i.e., $p(w|z = 1) = \mathcal{N}(w|0, 1)$.

**Horseshoe Prior:** The Horseshoe prior is a global–local shrinkage prior designed to strongly shrink small weights toward zero while allowing large weights to escape shrinkage through heavy-tailed behavior. It is defined as a Gaussian scale mixture:

$$w_j \mid \lambda_j, \tau \sim \mathcal{N}(0, \tau^2 \lambda_j^2),$$
$$\lambda_j \sim \mathcal{C}^+(0, 1),$$
$$\tau \sim \mathcal{C}^+(0, 1).$$

Here, $\tau$ controls global sparsity and $\lambda_j$ provides per-weight adaptive shrinkage.

The implied shrinkage factor under a quadratic approximation of the likelihood with curvature $\kappa_j$ is

$$\gamma_j = \frac{\kappa_j}{\kappa_j + \tau^{-2}\lambda_j^{-2}} \in (0, 1),$$

which interpolates between strong shrinkage ($\gamma_j \approx 0$ for small $\tau^2 \lambda_j^2$) and weak shrinkage ($\gamma_j \approx 1$ for large $\tau^2 \lambda_j^2$), enabling selective sparsity.

For variational inference with a mean-field approximation

$$q(w, \lambda, \tau) = \prod_j q(w_j)q(\lambda_j)q(\tau),$$

and estimate $D_{\text{KL}}(q \,\|\, p)$ via Monte Carlo due to the lack of closed-form expressions.

## B  ADDITIONAL PERFORMANCE COMPARISON

First, we report the results of applying SPIN to the **CIFAR-10**, **CIFAR-100**, and **TinyImageNet** datasets using **ResNet-20**, **ViT**, and additionally **DenseNet-30**. These results are obtained under the same experimental setting ($\sigma = 0.001$ with maximum sparsity level, i.e. SPIN$_{@\mathrm{maximum}}$) described in Section 5 and are summarized in Tables B-1, B-2, and B-3. We additionally report experimental results across all `sparsity` levels at which network pruning is feasible. Experiments are conducted on **ResNet-20**, **DenseNet-30**, and **ViT** using the **CIFAR-10**, **CIFAR-100**, and **TinyImageNet** datasets. Each experiment[4] is performed with standard deviations $\sigma \in \{0.1, 0.01, 0.001\}$. Results for $\sigma = 0.1$ are shown in Table B-4, Table B-5, and Table B-6. Results for $\sigma = 0.01$ are shown in Table B-7, Table B-8, and Table B-9. Results for $\sigma = 0.001$ are shown in Table B-10, Table B-11, and Table B-12. To better understand SPIN, we summarize several key observations below:

- For $\sigma = 0.1$, increasing `sparsity` in the proposed prior does not lead to significant performance gains (Table B-4, **ResNet-20, DenseNet-30**), and in some cases performs worse than existing priors (Table B-4, **DenseNet-30**; Table B-6, **DenseNet-30**). This suggests that when $\sigma \not\ll 1$, pruning-based priors do not provide substantial benefits.

- In contrast, for $\sigma = 0.01$ and $0.001$ (i.e., $\sigma \ll 1$), priors constructed from pruned networks (with `sparsity` $> 0.0$) yield performance improvements across most metrics, including Accuracy, NLL, MSP, Entropy, and MI (Table B-7–B-12). However, improvements in calibration (ECE) are less consistent, with some cases showing limited or no gains (Table B-7–B-9).

- For $\sigma = 0.01$ and $0.001$, higher `sparsity` often correlates with stronger performance (e.g., Table B-9, **ResNet-20**; Table B-10 and Table B-11, **ResNet-20, DenseNet-30**). Nevertheless, the trend is not strictly monotonic: moderate `sparsity` levels sometimes outperform extremely sparse configurations (e.g., Table B-7, **ViT**; Table B-8 and Table B-11, **ViT**; Table B-12, **ResNet-20, DenseNet-30**).

Table B-1: Performance evaluation of various priors on **CIFAR-10**. These results are obtained under the same experimental setting as in the main experiments ($\sigma = 0.001$ with the maximum sparsity level, i.e., SPIN$_{@\mathrm{maximum}}$). The value in parentheses next to SPIN indicates the maximum sparsity used for each model.

| Model | Prior | Acc(↑) | NLL(↓) | ECE(↓) | MSP(↑) | ENT(↑) | MI(↑) |
|---|---|---|---|---|---|---|---|
| ResNet 20 | *Iso. Gaussian* | 0.8922 | 0.3402 | 0.0275 | 0.8417 | 0.8564 | 0.8404 |
| | *Laplace* | 0.8931 | 0.3374 | 0.0321 | 0.8406 | 0.8561 | 0.8415 |
| | *Student-t* | 0.8932 | 0.3399 | 0.0282 | 0.8421 | 0.8578 | 0.8418 |
| | *Spike-and-Slab* | 0.8917 | 0.3461 | 0.0243 | 0.8404 | 0.8556 | 0.8411 |
| | *Horseshoe* | 0.8916 | 0.3248 | 0.0610 | 0.8460 | 0.8602 | 0.8409 |
| | MOPED | 0.8907 | 0.3442 | 0.0283 | 0.8403 | 0.8556 | 0.8408 |
| | SPIN (`0.8`) | **0.9043** | **0.3099** | 0.0305 | **0.8501** | **0.8641** | **0.8488** |
| DenseNet 30 | *Iso. Gaussian* | 0.8971 | 0.3322 | 0.0415 | 0.8415 | 0.8561 | 0.8337 |
| | *Laplace* | 0.8946 | 0.3313 | 0.0523 | 0.8344 | 0.8496 | 0.8314 |
| | *Student-t* | 0.8983 | 0.3242 | 0.0450 | 0.8422 | 0.8583 | 0.8433 |
| | *Spike-and-Slab* | 0.8971 | 0.3413 | **0.0320** | 0.8429 | 0.8581 | 0.8435 |
| | *Horseshoe* | 0.8618 | 0.4056 | 0.2141 | 0.8162 | 0.8213 | 0.7705 |
| | MOPED | 0.8969 | 0.3425 | 0.0462 | 0.8397 | 0.8541 | 0.8384 |
| | SPIN (`0.5`) | **0.9131** | **0.2829** | 0.0459 | **0.8578** | **0.8712** | **0.8568** |
| ViT | *Iso. Gaussian* | 0.6456 | 0.9881 | 0.0219 | 0.6832 | 0.7032 | 0.6495 |
| | *Laplace* | 0.6438 | 0.9936 | 0.0211 | 0.6824 | 0.7024 | 0.6484 |
| | *Student-t* | 0.6425 | 0.9983 | 0.0252 | 0.6826 | 0.7019 | 0.6499 |
| | *Spike-and-Slab* | 0.6444 | 0.9958 | 0.0243 | 0.6823 | 0.7021 | 0.6471 |
| | *Horseshoe* | 0.5926 | 1.1060 | 0.2778 | 0.6661 | 0.6850 | 0.6458 |
| | MOPED | 0.5936 | 1.1271 | 0.0251 | 0.6600 | 0.6772 | 0.6267 |
| | SPIN (`0.4`) | **0.7448** | **0.7197** | **0.0189** | **0.7238** | **0.7485** | **0.7169** |

---

[4] Note that SPIN($\cdot$) indicates the `sparsity`.

Table B-2: Performance evaluation of various priors on **CIFAR-100**. These results are obtained under the same experimental setting as in the main experiments ($\sigma = 0.001$ with the maximum sparsity level, i.e., SPIN$_{@\text{maximum}}$). The value in parentheses next to SPIN indicates the maximum sparsity used for each model.

| Model | Prior | Acc(↑) | NLL(↓) | ECE(↓) | MSP(↑) | ENT(↑) | MI(↑) |
|---|---|---|---|---|---|---|---|
| ResNet 20 | *Iso. Gaussian* | 0.6341 | 1.3596 | **0.0275** | 0.7189 | 0.7365 | 0.7231 |
| | *Laplace* | 0.6320 | 1.3616 | 0.0258 | 0.7201 | 0.7383 | 0.7254 |
| | *Student-t* | 0.6299 | 1.3700 | 0.0277 | 0.7174 | 0.7357 | 0.7229 |
| | *Spike-and-Slab* | 0.6326 | 1.3626 | 0.0250 | 0.7179 | 0.7357 | 0.7249 |
| | *Horseshoe* | 0.6292 | 1.3556 | 0.0428 | 0.7193 | 0.7363 | 0.7262 |
| | MOPED | 0.6298 | 1.3668 | 0.0269 | 0.7156 | 0.7340 | 0.7208 |
| | SPIN (0.2) | **0.6696** | **1.3534** | 0.0203 | **0.7363** | **0.7569** | **0.7454** |
| DenseNet 30 | *Iso. Gaussian* | 0.6556 | 1.2860 | 0.0374 | 0.7294 | 0.7462 | 0.7425 |
| | *Laplace* | 0.6516 | 1.3180 | 0.0384 | 0.7237 | 0.7408 | 0.7243 |
| | *Student-t* | 0.6461 | 1.3184 | 0.0393 | 0.7253 | 0.7393 | 0.7267 |
| | *Spike-and-Slab* | 0.3220 | 2.6106 | 0.1208 | 0.5585 | 0.5613 | 0.5382 |
| | *Horseshoe* | 0.6180 | 1.3903 | 0.1379 | 0.7113 | 0.7185 | 0.6937 |
| | MOPED | 0.6483 | 1.3220 | **0.0361** | 0.7276 | 0.7440 | 0.7248 |
| | SPIN (0.7) | **0.6837** | **1.2025** | 0.0537 | **0.7527** | **0.7699** | **0.7485** |
| ViT | *Iso. Gaussian* | 0.4997 | 1.9502 | 0.0273 | 0.6574 | 0.6733 | 0.6488 |
| | *Laplace* | 0.4987 | 1.9517 | 0.0248 | **0.6572** | 0.6727 | 0.6508 |
| | *Student-t* | 0.4884 | 1.9872 | **0.0246** | 0.6524 | 0.6670 | 0.6472 |
| | *Spike-and-Slab* | 0.4983 | 1.9495 | 0.0262 | 0.6564 | 0.6716 | **0.6485** |
| | *Horseshoe* | 0.4995 | 1.9374 | 0.0217 | 0.6562 | 0.6700 | 0.6512 |
| | MOPED | 0.5008 | 1.9348 | 0.0272 | 0.6557 | 0.6707 | 0.6510 |
| | SPIN (0.8) | **0.5022** | **1.9384** | 0.0364 | 0.6561 | **0.6730** | 0.6434 |

Table B-3: Performance evaluation of various priors on **TinyImageNet**. These results are obtained under the same experimental setting as in the main experiments ($\sigma = 0.001$ with the maximum sparsity level, i.e., SPIN$_{@\text{maximum}}$). The value in parentheses next to SPIN indicates the maximum sparsity used for each model.

| Model | Prior | Acc(↑) | NLL(↓) | ECE(↓) | MSP(↑) | ENT(↑) | MI(↑) |
|---|---|---|---|---|---|---|---|
| ResNet 18 | *Iso. Gaussian* | 0.5072 | 2.1029 | 0.0863 | 0.7865 | 0.8230 | 0.5028 |
| | *Laplace* | 0.5200 | 2.0847 | 0.1050 | 0.7968 | 0.8332 | 0.5160 |
| | *Student-t* | 0.5156 | 2.1025 | 0.0906 | 0.7937 | 0.8291 | 0.4968 |
| | *Spike-and-Slab* | 0.5076 | 2.1171 | 0.0858 | 0.7815 | 0.8162 | 0.5209 |
| | *Horseshoe* | 0.5124 | 2.0855 | 0.0794 | 0.7936 | 0.8260 | 0.5714 |
| | MOPED | 0.5122 | 2.1041 | **0.0851** | 0.7676 | 0.7996 | 0.5439 |
| | SPIN (0.3) | **0.5436** | **2.0596** | 0.1279 | **0.8068** | **0.8402** | **0.5545** |
| MobileNet v2 | *Iso. Gaussian* | 0.5675 | 1.7557 | 0.0795 | 0.7753 | 0.8109 | 0.6820 |
| | *Laplace* | 0.5729 | 1.7374 | 0.0748 | 0.7846 | 0.8232 | 0.6647 |
| | *Student-t* | 0.5859 | 1.6876 | 0.0788 | 0.7915 | 0.8281 | 0.6639 |
| | *Spike-and-Slab* | 0.5785 | 1.7062 | **0.0608** | 0.7893 | 0.8260 | 0.6644 |
| | *Horseshoe* | 0.5415 | 1.8492 | 0.1430 | 0.5888 | 0.5104 | 0.9634 |
| | MOPED | 0.5714 | 1.7343 | 0.0821 | 0.7771 | 0.8137 | 0.7039 |
| | SPIN (0.3) | **0.6315** | **1.4864** | 0.0796 | **0.8051** | **0.8367** | **0.7120** |
| ViT | *Iso. Gaussian* | 0.4516 | 2.3898 | 0.1302 | 0.6670 | 0.6898 | 0.8204 |
| | *Laplace* | 0.4544 | 2.3801 | 0.1327 | 0.6682 | 0.6790 | 0.8069 |
| | *Student-t* | 0.4637 | 2.3266 | 0.1188 | 0.6908 | 0.7075 | 0.7973 |
| | *Spike-and-Slab* | 0.4691 | **2.3208** | **0.1083** | 0.6927 | 0.7030 | **0.8633** |
| | *Horseshoe* | 0.5022 | 1.3984 | 0.0364 | 0.6561 | 0.6730 | 0.6434 |
| | MOPED | 0.4512 | 2.3976 | 0.1282 | 0.6681 | 0.6923 | 0.7760 |
| | SPIN (0.5) | **0.4706** | 2.3441 | 0.1108 | **0.6997** | **0.7187** | 0.6725 |

Table B-4: Performance evaluation of various priors on **CIFAR-10** ($\sigma = 0.1$). **Bold** entries indicate the best performance (per model, per column; Accuracy/MSP/Entropy/MI ↑, NLL/ECE ↓).

| Model | Prior | Acc (↑) | NLL(↓) | ECE(↓) | MSP(↑) | Entropy(↑) | MI(↑) |
|---|---|---|---|---|---|---|---|
| ResNet 20 | *Iso. Gaussian* | 0.8922 | 0.3402 | 0.0275 | 0.8417 | 0.8564 | 0.8404 |
| | *Laplace* | 0.8931 | 0.3374 | 0.0321 | 0.8406 | 0.8561 | 0.8415 |
| | *Student-t* | 0.8931 | 0.3399 | 0.0282 | 0.8421 | 0.8578 | 0.8418 |
| | *Spike-and-Slab* | 0.8917 | 0.3461 | **0.0243** | 0.8404 | 0.8556 | 0.8411 |
| | *Horseshoe* | 0.8916 | 0.3248 | 0.0610 | 0.8460 | 0.8602 | 0.8409 |
| | MOPED | 0.8907 | 0.3442 | 0.0283 | 0.8403 | 0.8556 | 0.8408 |
| | SPIN (0.0) | **0.8964** | **0.3068** | 0.0897 | **0.8560** | **0.8669** | **0.8506** |
| | SPIN (0.1) | 0.8925 | 0.3250 | 0.0804 | 0.8449 | 0.8592 | 0.8385 |
| | SPIN (0.2) | 0.8912 | 0.3251 | 0.0643 | 0.8455 | 0.8613 | 0.8400 |
| | SPIN (0.3) | 0.8958 | 0.3189 | 0.0731 | 0.8455 | 0.8581 | 0.8356 |
| | SPIN (0.4) | 0.8923 | 0.3217 | 0.0679 | 0.8512 | 0.8653 | 0.8456 |
| | SPIN (0.5) | 0.8928 | 0.3311 | 0.0444 | 0.8470 | 0.8608 | 0.8458 |
| | SPIN (0.6) | 0.8918 | 0.3266 | 0.0575 | 0.8450 | 0.8621 | 0.8478 |
| | SPIN (0.7) | 0.8943 | 0.3318 | 0.0456 | 0.8462 | 0.8613 | 0.8496 |
| | SPIN (0.8) | 0.8937 | 0.3217 | 0.0462 | 0.8435 | 0.8584 | 0.8415 |
| DenseNet 30 | *Iso. Gaussian* | 0.8971 | 0.3322 | 0.0415 | 0.8415 | 0.8561 | 0.8337 |
| | *Laplace* | 0.8946 | 0.3313 | 0.0523 | 0.8344 | 0.8496 | 0.8314 |
| | *Student-t* | **0.8983** | 0.3242 | 0.0450 | 0.8422 | 0.8583 | 0.8433 |
| | *Spike-and-Slab* | 0.8971 | 0.3413 | **0.0320** | 0.8429 | 0.8581 | **0.8435** |
| | *Horseshoe* | 0.8618 | 0.4056 | 0.2141 | 0.8162 | 0.8213 | 0.7705 |
| | MOPED | 0.8969 | 0.3425 | 0.0462 | 0.8397 | 0.8541 | 0.8384 |
| | SPIN (0.0) | 0.8921 | 0.3135 | 0.1501 | 0.8402 | 0.8468 | 0.8207 |
| | SPIN (0.1) | 0.8951 | **0.3074** | 0.1342 | **0.8446** | 0.8532 | 0.8186 |
| | SPIN (0.2) | 0.8962 | 0.3127 | 0.1212 | 0.8402 | 0.8501 | 0.8266 |
| | SPIN (0.3) | 0.8965 | 0.3082 | 0.1037 | 0.8414 | 0.8537 | 0.8274 |
| | SPIN (0.4) | 0.8962 | 0.3122 | 0.0924 | 0.8438 | **0.8593** | 0.8354 |
| | SPIN (0.5) | 0.8958 | 0.3143 | 0.0795 | 0.8429 | 0.8568 | 0.8358 |
| ViT | *Iso. Gaussian* | 0.6179 | 1.0780 | 0.0266 | 0.6682 | 0.6862 | 0.6286 |
| | *Laplace* | 0.6834 | 0.8862 | **0.0146** | 0.7060 | 0.7281 | 0.6817 |
| | *Student-t* | 0.6639 | 0.9472 | 0.0226 | 0.6911 | 0.7107 | 0.6502 |
| | *Spike-and-Slab* | 0.3321 | 1.7888 | 0.0314 | 0.5396 | 0.5443 | 0.5116 |
| | *Horseshoe* | 0.5926 | 1.1060 | 0.2778 | 0.6661 | 0.6850 | 0.6458 |
| | MOPED | 0.5493 | 1.2523 | 0.0266 | 0.6439 | 0.6556 | 0.5843 |
| | SPIN (0.0) | 0.6357 | 1.0221 | 0.0296 | 0.6809 | 0.6997 | 0.6440 |
| | SPIN (0.1) | 0.7031 | 0.8404 | 0.0294 | 0.7134 | 0.7370 | 0.6880 |
| | SPIN (0.2) | **0.7191** | **0.7955** | 0.0329 | **0.7157** | **0.7397** | **0.6910** |
| | SPIN (0.3) | 0.6509 | 0.9752 | 0.0274 | 0.6827 | 0.7040 | 0.6420 |
| | SPIN (0.4) | 0.6670 | 0.9316 | 0.0230 | 0.6979 | 0.7181 | 0.6604 |

Table B-5: Performance evaluation of various priors on **CIFAR-100** ($\sigma = 0.1$). **Bold** entries indicate the best performance (per model, per column; Accuracy/MSP/Entropy/MI $\uparrow$, NLL/ECE $\downarrow$).

| Model | Prior | Acc($\uparrow$) | NLL($\downarrow$) | ECE($\downarrow$) | MSP($\uparrow$) | Entropy($\uparrow$) | MI($\uparrow$) |
|---|---|---|---|---|---|---|---|
| ResNet 20 | *Iso. Gaussian* | 0.6339 | 1.3668 | **0.0226** | 0.7190 | 0.7366 | 0.7325 |
| | *Laplace* | 0.6316 | 1.3614 | 0.0260 | 0.7218 | 0.7425 | 0.7272 |
| | *Student-t* | 0.6325 | 1.3699 | 0.0257 | 0.7200 | 0.7370 | 0.7281 |
| | *Spike-and-Slab* | 0.6307 | 1.3660 | 0.0264 | 0.7245 | **0.7436** | **0.7370** |
| | *Horseshoe* | 0.6292 | 1.3556 | 0.0428 | 0.7193 | 0.7363 | 0.7262 |
| | MOPED | 0.6396 | **1.3522** | 0.0343 | 0.7262 | 0.7426 | 0.7328 |
| | SPIN(0.0) | 0.6462 | 1.2743 | 0.1032 | 0.7261 | 0.7391 | 0.7286 |
| | SPIN(0.1) | 0.6438 | 1.2642 | 0.0682 | 0.7302 | 0.7472 | 0.7313 |
| | SPIN(0.2) | **0.6499** | 1.2857 | 0.0786 | **0.7277** | 0.7439 | 0.7244 |
| DenseNet 30 | *Iso. Gaussian* | 0.6556 | 1.2860 | 0.0374 | 0.7294 | 0.7462 | 0.7425 |
| | *Laplace* | 0.6516 | 1.3180 | 0.0384 | 0.7237 | 0.7408 | 0.7243 |
| | *Student-t* | 0.6461 | 1.3184 | 0.0393 | 0.7253 | 0.7393 | 0.7267 |
| | *Spike-and-Slab* | 0.6528 | 1.3128 | 0.0238 | 0.7336 | 0.7520 | 0.7399 |
| | *Horseshoe* | 0.6180 | 1.3903 | 0.1379 | 0.7113 | 0.7185 | 0.6937 |
| | MOPED | 0.6483 | 1.3220 | **0.0361** | 0.7276 | 0.7440 | 0.7248 |
| | SPIN(0.0) | 0.6575 | **1.2239** | 0.1722 | 0.7318 | 0.7339 | 0.6963 |
| | SPIN(0.1) | 0.6517 | 1.2459 | 0.1502 | 0.7337 | 0.7417 | 0.7085 |
| | SPIN(0.2) | 0.6542 | 1.2253 | 0.1322 | **0.7394** | 0.7505 | 0.7127 |
| | SPIN(0.3) | 0.6521 | 1.2608 | 0.1176 | 0.7281 | 0.7416 | 0.7095 |
| | SPIN(0.4) | 0.6555 | 1.2757 | 0.1205 | 0.7333 | 0.7444 | 0.7101 |
| | SPIN(0.5) | 0.6505 | 1.2740 | 0.0862 | 0.7357 | 0.7482 | 0.7300 |
| | SPIN(0.6) | **0.6586** | 1.2502 | 0.0766 | 0.7377 | **0.7523** | 0.7272 |
| | SPIN(0.7) | 0.6548 | 1.3026 | 0.0726 | 0.7326 | 0.7481 | **0.7297** |
| ViT | *Iso. Gaussian* | 0.4884 | 1.9897 | 0.0251 | 0.6491 | 0.6646 | 0.6470 |
| | *Laplace* | 0.4926 | 1.9797 | 0.0314 | 0.6604 | 0.6715 | 0.6483 |
| | *Student-t* | 0.4778 | 2.0178 | 0.0238 | 0.6469 | 0.6611 | 0.6515 |
| | *Spike-and-Slab* | 0.4936 | 1.9555 | 0.0254 | 0.6563 | 0.6723 | 0.6544 |
| | *Horseshoe* | 0.4995 | 1.9374 | 0.0217 | 0.6562 | 0.6700 | 0.6512 |
| | MOPED | 0.4939 | 1.9669 | 0.0268 | 0.6551 | 0.6691 | 0.6440 |
| | SPIN(0.0) | 0.5153 | 1.8729 | 0.0185 | 0.6641 | 0.6812 | 0.6678 |
| | SPIN(0.1) | 0.4838 | 2.0181 | 0.0255 | 0.6516 | 0.6641 | 0.6458 |
| | SPIN(0.2) | 0.5003 | 1.9334 | 0.0198 | 0.6558 | 0.6719 | 0.6546 |
| | SPIN(0.3) | 0.5100 | 1.8809 | 0.0201 | 0.6576 | 0.6745 | 0.6489 |
| | SPIN(0.4) | 0.5127 | 1.8788 | 0.0198 | 0.6608 | 0.6767 | 0.6633 |
| | SPIN(0.5) | 0.5013 | 1.9204 | **0.0166** | 0.6565 | 0.6707 | 0.6614 |
| | SPIN(0.6) | 0.4950 | 1.9711 | 0.0277 | 0.6527 | 0.6679 | 0.6503 |
| | SPIN(0.7) | 0.4943 | 1.9721 | 0.0228 | 0.6543 | 0.6659 | 0.6486 |
| | SPIN(0.8) | **0.5290** | **1.7941** | 0.0217 | **0.6709** | **0.6879** | **0.6679** |

Table B-6: Performance evaluation of various priors on **TinyImageNet** ($\sigma = 0.1$). **Bold** entries indicate the best performance (per model, per column; Accuracy/MSP/Entropy/MI ↑, NLL/ECE ↓).

| Model | Prior | Acc(↑) | NLL(↓) | ECE(↓) | MSP(↑) | Entropy(↑) | MI(↑) |
|---|---|---|---|---|---|---|---|
| ResNet 18 | *Iso. Gaussian* | 0.5072 | 2.1029 | 0.0863 | 0.7865 | 0.8230 | 0.5028 |
| | *Laplace* | 0.5200 | 2.0847 | 0.1050 | **0.7968** | **0.8332** | 0.5160 |
| | *Student-t* | 0.5156 | 2.1025 | 0.0906 | 0.7937 | 0.8291 | 0.4968 |
| | *Spike-and-Slab* | 0.5076 | 2.1171 | 0.0858 | 0.7815 | 0.8162 | 0.5209 |
| | *Horseshoe* | 0.5124 | 2.0855 | 0.0794 | 0.7936 | 0.8260 | 0.5714 |
| | MOPED | 0.5122 | 2.1041 | 0.0851 | 0.7676 | 0.7996 | 0.5439 |
| | SPIN(0.0) | 0.5120 | 2.0970 | **0.0762** | 0.7917 | 0.8283 | 0.5104 |
| | SPIN(0.1) | 0.5173 | 2.1093 | 0.1090 | 0.7691 | 0.8004 | 0.5008 |
| | SPIN(0.2) | 0.5084 | 2.1127 | 0.0773 | 0.7363 | 0.7664 | **0.5485** |
| | SPIN(0.3) | **0.5203** | **2.0590** | 0.0847 | 0.7804 | 0.8131 | 0.5304 |
| MobileNet v2 | *Iso. Gaussian* | 0.5675 | 1.7557 | 0.0795 | 0.7753 | 0.8109 | 0.6820 |
| | *Laplace* | 0.5729 | 1.7374 | **0.0748** | 0.7846 | 0.8232 | 0.6647 |
| | *Student-t* | **0.5859** | **1.6876** | 0.0788 | **0.7915** | **0.8281** | 0.6639 |
| | *Spike-and-Slab* | 0.5785 | 1.7062 | 0.0608 | 0.7893 | 0.8260 | 0.6644 |
| | *Horseshoe* | 0.5415 | 1.8492 | 0.1430 | 0.5888 | 0.5104 | 0.9634 |
| | MOPED | 0.5714 | 1.7343 | 0.0821 | 0.7771 | 0.8137 | **0.7039** |
| | SPIN(0.0) | 0.5109 | 2.0526 | 0.1960 | 0.6781 | 0.7153 | 0.5978 |
| | SPIN(0.1) | 0.5509 | 1.8491 | 0.1858 | 0.7080 | 0.7396 | 0.6333 |
| | SPIN(0.2) | 0.5486 | 1.8435 | 0.1491 | 0.7291 | 0.7708 | 0.6882 |
| | SPIN(0.3) | 0.5591 | 1.7992 | 0.1418 | 0.7587 | 0.7996 | 0.6723 |
| ViT | *Iso. Gaussian* | 0.4516 | 2.3898 | 0.1302 | 0.6670 | 0.6898 | 0.8204 |
| | *Laplace* | 0.4544 | 2.3801 | 0.1327 | 0.6682 | 0.6790 | 0.8069 |
| | *Student-t* | 0.4637 | 2.3266 | 0.1188 | 0.6908 | 0.7075 | 0.7973 |
| | *Spike-and-Slab* | **0.4691** | **2.3208** | **0.1083** | **0.6927** | **0.7030** | **0.8633** |
| | *Horseshoe* | 0.5022 | 1.3984 | 0.0364 | 0.6561 | 0.6730 | 0.6434 |
| | MOPED | 0.4512 | 2.3976 | 0.1282 | 0.6681 | 0.6923 | 0.7760 |
| | SPIN(0.0) | 0.4499 | 2.3690 | 0.1434 | 0.6912 | 0.7087 | 0.7781 |
| | SPIN(0.1) | 0.4478 | 2.3797 | 0.1445 | 0.6669 | 0.6757 | 0.8077 |
| | SPIN(0.2) | 0.4460 | 2.3937 | 0.1361 | 0.6785 | 0.6924 | 0.8045 |
| | SPIN(0.3) | 0.4535 | 2.3759 | 0.1362 | 0.6697 | 0.6904 | 0.8194 |
| | SPIN(0.4) | 0.4456 | 2.4058 | 0.1398 | 0.6786 | 0.7011 | 0.7848 |
| | SPIN(0.5) | 0.4482 | 2.3845 | 0.1454 | 0.6581 | 0.6716 | 0.8116 |

Table B-7: Performance evaluation of various priors on **CIFAR-10** ($\sigma = 0.01$). **Bold** entries indicate the best performance (per model, per column; Accuracy/MSP/Entropy/MI ↑, NLL/ECE ↓).

| Model | Prior | Acc(↑) | NLL(↓) | ECE(↓) | MSP(↑) | Entropy(↑) | MI(↑) |
|---|---|---|---|---|---|---|---|
| ResNet 20 | *Iso. Gaussian* | 0.8922 | 0.3402 | 0.0275 | 0.8417 | 0.8564 | 0.8404 |
| | *Laplace* | 0.8931 | 0.3374 | 0.0321 | 0.8406 | 0.8561 | 0.8415 |
| | *Student-t* | 0.8931 | 0.3399 | 0.0282 | 0.8421 | 0.8578 | 0.8418 |
| | *Spike-and-Slab* | 0.8917 | 0.3461 | 0.0243 | 0.8404 | 0.8556 | 0.8411 |
| | *Horseshoe* | 0.8916 | 0.3248 | 0.0610 | 0.8460 | 0.8602 | 0.8409 |
| | MOPED | 0.8907 | 0.3442 | 0.0283 | 0.8403 | 0.8556 | 0.8408 |
| | SPIN (0.0) | 0.9061 | 0.2930 | 0.0647 | 0.8580 | 0.8734 | 0.8550 |
| | SPIN (0.1) | 0.9086 | 0.2853 | 0.0568 | 0.8621 | 0.8757 | 0.8560 |
| | SPIN (0.2) | 0.9069 | 0.2824 | 0.0562 | 0.8601 | 0.8735 | 0.8555 |
| | SPIN (0.3) | 0.9112 | 0.2820 | 0.0547 | 0.8632 | 0.8769 | 0.8610 |
| | SPIN (0.4) | 0.9074 | 0.2847 | 0.0491 | 0.8631 | 0.8767 | 0.8594 |
| | SPIN (0.5) | 0.9097 | **0.2796** | 0.0476 | **0.8636** | 0.8773 | 0.8620 |
| | SPIN (0.6) | 0.9098 | 0.2847 | 0.0416 | 0.8607 | 0.8732 | 0.8590 |
| | SPIN (0.7) | **0.9120** | 0.2859 | 0.0420 | 0.8630 | **0.8773** | **0.8627** |
| | SPIN (0.8) | 0.9063 | 0.3045 | 0.0384 | 0.8548 | 0.8711 | 0.8530 |
| DenseNet 30 | *Iso. Gaussian* | 0.8971 | 0.3322 | 0.0415 | 0.8415 | 0.8561 | 0.8337 |
| | *Laplace* | 0.8946 | 0.3313 | 0.0523 | 0.8344 | 0.8496 | 0.8314 |
| | *Student-t* | 0.8983 | 0.3242 | 0.0450 | 0.8422 | 0.8583 | 0.8433 |
| | *Spike-and-Slab* | 0.8971 | 0.3413 | **0.0320** | 0.8429 | 0.8581 | **0.8435** |
| | *Horseshoe* | 0.8618 | 0.4056 | 0.2141 | 0.8162 | 0.8213 | 0.7705 |
| | MOPED | 0.8969 | 0.3425 | 0.0462 | 0.8397 | 0.8541 | 0.8384 |
| | SPIN (0.0) | 0.9152 | 0.2776 | 0.0569 | 0.8572 | 0.8703 | 0.8562 |
| | SPIN (0.1) | **0.9167** | 0.2667 | 0.0646 | 0.8578 | 0.8712 | 0.8538 |
| | SPIN (0.2) | 0.9148 | 0.2720 | 0.0682 | 0.8599 | 0.8725 | 0.8567 |
| | SPIN (0.3) | 0.9147 | **0.2624** | 0.0711 | 0.8606 | 0.8737 | 0.8534 |
| | SPIN (0.4) | 0.9156 | 0.2658 | 0.0660 | 0.8605 | 0.8731 | 0.8523 |
| | SPIN (0.5) | 0.9149 | 0.2680 | 0.0631 | **0.8616** | **0.8724** | 0.8502 |
| ViT | *Iso. Gaussian* | 0.6179 | 1.0780 | 0.0266 | 0.6682 | 0.6862 | 0.6286 |
| | *Laplace* | 0.6834 | 0.8862 | **0.0146** | 0.7060 | 0.7281 | 0.6817 |
| | *Student-t* | 0.6639 | 0.9472 | 0.0226 | 0.6911 | 0.7107 | 0.6502 |
| | *Spike-and-Slab* | 0.3321 | 1.7888 | 0.0314 | 0.5396 | 0.5443 | 0.5116 |
| | *Horseshoe* | 0.5926 | 1.1060 | 0.2778 | 0.6661 | 0.6850 | 0.6458 |
| | MOPED | 0.5493 | 1.2523 | 0.0266 | 0.6439 | 0.6556 | 0.5843 |
| | SPIN (0.0) | **0.7698** | **0.6643** | 0.0699 | **0.7482** | **0.7693** | 0.7337 |
| | SPIN (0.1) | 0.7411 | 0.7287 | 0.0576 | 0.7332 | 0.7558 | 0.7226 |
| | SPIN (0.2) | 0.7591 | 0.6875 | 0.0500 | 0.7440 | 0.7658 | **0.7369** |
| | SPIN (0.3) | 0.7621 | 0.6708 | 0.0443 | 0.7405 | 0.7646 | 0.7320 |
| | SPIN (0.4) | 0.7514 | 0.6913 | 0.0396 | 0.7384 | 0.7616 | 0.7182 |

Table B-8: Performance evaluation of various priors on **CIFAR-100** ($\sigma = 0.01$). **Bold** entries indicate the best performance (per model, per column; Accuracy/MSP/Entropy/MI ↑, NLL/ECE ↓).

| Model | Prior | Acc(↑) | NLL(↓) | ECE(↓) | MSP(↑) | Entropy(↑) | MI(↑) |
|---|---|---|---|---|---|---|---|
| ResNet 20 | *Iso. Gaussian* | 0.6339 | 1.3668 | **0.0226** | 0.7190 | 0.7366 | 0.7325 |
| | *Laplace* | 0.6316 | 1.3614 | 0.0260 | 0.7218 | 0.7425 | 0.7272 |
| | *Student-t* | 0.6325 | 1.3699 | 0.0257 | 0.7200 | 0.7370 | 0.7281 |
| | *Spike-and-Slab* | 0.6307 | 1.3660 | 0.0264 | 0.7245 | 0.7436 | **0.7370** |
| | *Horseshoe* | 0.6292 | 1.3556 | 0.0428 | 0.7193 | 0.7363 | 0.7262 |
| | MOPED | 0.6396 | 1.3522 | 0.0343 | 0.7262 | 0.7426 | 0.7328 |
| | SPIN(0.0) | 0.6462 | 1.2743 | 0.1032 | 0.7261 | 0.7391 | 0.7286 |
| | SPIN(0.1) | 0.6438 | **1.2642** | 0.0682 | **0.7302** | **0.7472** | 0.7313 |
| | SPIN(0.2) | **0.6499** | 1.2857 | 0.0786 | 0.7277 | 0.7439 | 0.7244 |
| DenseNet 30 | *Iso. Gaussian* | 0.6556 | 1.2860 | 0.0374 | 0.7294 | 0.7462 | 0.7425 |
| | *Laplace* | 0.6516 | 1.3180 | 0.0384 | 0.7237 | 0.7408 | 0.7243 |
| | *Student-t* | 0.6461 | 1.3184 | 0.0393 | 0.7253 | 0.7393 | 0.7267 |
| | *Spike-and-Slab* | 0.6528 | 1.3128 | 0.0238 | 0.7336 | 0.7520 | 0.7399 |
| | *Horseshoe* | 0.6180 | 1.3903 | 0.1379 | 0.7113 | 0.7185 | 0.6937 |
| | MOPED | 0.6483 | 1.3220 | **0.0361** | 0.7276 | 0.7440 | 0.7248 |
| | SPIN(0.0) | 0.6824 | 1.2039 | 0.0574 | 0.7553 | **0.7725** | **0.7462** |
| | SPIN(0.1) | 0.6825 | 1.2185 | 0.0727 | 0.7497 | 0.7656 | 0.7446 |
| | SPIN(0.2) | 0.6830 | 1.1844 | 0.0882 | 0.7524 | 0.7678 | 0.7447 |
| | SPIN(0.3) | 0.6806 | 1.1791 | 0.0925 | 0.7557 | 0.7688 | 0.7425 |
| | SPIN(0.4) | 0.6789 | 1.1780 | 0.0938 | 0.7523 | 0.7668 | 0.7372 |
| | SPIN(0.5) | 0.6836 | 1.1614 | 0.0958 | **0.7577** | 0.7710 | 0.7441 |
| | SPIN(0.6) | **0.6890** | **1.1359** | 0.0836 | 0.7529 | 0.7677 | 0.7431 |
| | SPIN(0.7) | 0.6825 | 1.1713 | 0.0712 | 0.7530 | 0.7700 | 0.7446 |
| ViT | *Iso. Gaussian* | 0.4884 | 1.9897 | 0.0251 | 0.6491 | 0.6646 | 0.6470 |
| | *Laplace* | 0.4926 | 1.9797 | 0.0314 | 0.6604 | 0.6715 | 0.6483 |
| | *Student-t* | 0.4778 | 2.0178 | 0.0238 | 0.6469 | 0.6611 | 0.6515 |
| | *Spike-and-Slab* | 0.4936 | 1.9555 | 0.0254 | 0.6563 | 0.6723 | 0.6544 |
| | *Horseshoe* | 0.4995 | 1.9374 | 0.0217 | 0.6562 | 0.6700 | 0.6512 |
| | MOPED | 0.4939 | 1.9669 | 0.0268 | 0.6551 | 0.6691 | 0.6440 |
| | SPIN(0.0) | 0.5656 | 1.6199 | 0.0314 | 0.6882 | 0.7041 | 0.6907 |
| | SPIN(0.1) | 0.5685 | **1.6160** | 0.0340 | 0.6873 | 0.7041 | 0.6936 |
| | SPIN(0.2) | 0.5719 | 1.6162 | 0.0240 | **0.6936** | 0.7089 | **0.6960** |
| | SPIN(0.3) | **0.5732** | 1.6514 | 0.0218 | 0.6930 | **0.7107** | 0.6909 |
| | SPIN(0.4) | 0.5657 | 1.6656 | 0.0211 | 0.6867 | 0.7034 | 0.6873 |
| | SPIN(0.5) | 0.5538 | 1.7103 | 0.0233 | 0.6800 | 0.6989 | 0.6897 |
| | SPIN(0.6) | 0.5039 | 1.9216 | 0.0238 | 0.6607 | 0.6754 | 0.6635 |
| | SPIN(0.7) | 0.4841 | 2.0072 | 0.0261 | 0.6457 | 0.6605 | 0.6423 |
| | SPIN(0.8) | 0.4923 | 1.9704 | **0.0192** | 0.6486 | 0.6612 | 0.6469 |

Table B-9: Performance evaluation of various priors on **TinyImageNet** ($\sigma = 0.01$). **Bold** entries indicate the best performance (per model, per column; Accuracy/MSP/Entropy/MI ↑, NLL/ECE ↓).

| Model | Prior | Acc(↑) | NLL(↓) | ECE(↓) | MSP(↑) | Entropy(↑) | MI(↑) |
|---|---|---|---|---|---|---|---|
| ResNet 18 | *Iso. Gaussian* | 0.5072 | 2.1029 | 0.0863 | 0.7865 | 0.8230 | 0.5028 |
| | *Laplace* | 0.5200 | 2.0847 | 0.1050 | 0.7968 | 0.8332 | 0.5160 |
| | *Student-t* | 0.5156 | 2.1025 | 0.0906 | 0.7937 | 0.8291 | 0.4968 |
| | *Spike-and-Slab* | 0.5076 | 2.1171 | 0.0858 | 0.7815 | 0.8162 | 0.5209 |
| | *Horseshoe* | 0.5124 | 2.0855 | 0.0794 | 0.7936 | 0.8260 | 0.5714 |
| | MOPED | 0.5122 | 2.1041 | **0.0851** | 0.7676 | 0.7996 | 0.5439 |
| | SPIN(0.0) | **0.5332** | **2.0382** | 0.1198 | 0.7920 | 0.8248 | **0.5689** |
| | SPIN(0.1) | 0.5283 | 2.0552 | 0.0965 | 0.7730 | 0.8049 | 0.5447 |
| | SPIN(0.2) | 0.5301 | 2.0667 | 0.1260 | 0.7813 | 0.8199 | 0.5511 |
| | SPIN(0.3) | 0.5263 | 2.0550 | 0.1206 | **0.7986** | **0.8370** | 0.5307 |
| MobileNet v2 | *Iso. Gaussian* | 0.5675 | 1.7557 | 0.0795 | 0.7753 | 0.8109 | 0.6820 |
| | *Laplace* | 0.5729 | 1.7374 | 0.0748 | 0.7846 | 0.8232 | 0.6647 |
| | *Student-t* | 0.5859 | 1.6876 | 0.0788 | 0.7915 | 0.8281 | 0.6639 |
| | *Spike-and-Slab* | 0.5785 | 1.7062 | **0.0608** | 0.7893 | 0.8260 | 0.6644 |
| | *Horseshoe* | 0.5415 | 1.8492 | 0.1430 | 0.5888 | 0.5104 | 0.9634 |
| | MOPED | 0.5714 | 1.7343 | 0.0821 | 0.7771 | 0.8137 | **0.7039** |
| | SPIN(0.0) | 0.5109 | 2.0526 | 0.1960 | 0.6781 | 0.7153 | 0.5978 |
| | SPIN(0.1) | 0.6057 | 1.6242 | 0.2138 | 0.7900 | 0.8221 | 0.6199 |
| | SPIN(0.2) | **0.6079** | **1.6095** | 0.1883 | **0.7990** | **0.8311** | 0.6319 |
| | SPIN(0.3) | 0.5824 | 1.7058 | 0.1609 | 0.7740 | 0.8079 | 0.6611 |
| ViT | *Iso. Gaussian* | 0.4516 | 2.3898 | 0.1302 | 0.6670 | 0.6898 | 0.8204 |
| | *Laplace* | 0.4544 | 2.3801 | 0.1327 | 0.6682 | 0.6790 | 0.8069 |
| | *Student-t* | 0.4637 | 2.3266 | 0.1188 | 0.6908 | 0.7075 | 0.7973 |
| | *Spike-and-Slab* | **0.4691** | **2.3208** | **0.1083** | 0.6927 | 0.7030 | **0.8633** |
| | *Horseshoe* | 0.5022 | 1.3984 | 0.0364 | 0.6561 | 0.6730 | 0.6434 |
| | MOPED | 0.4512 | 2.3976 | 0.1282 | 0.6681 | 0.6923 | 0.7760 |
| | SPIN(0.0) | 0.4556 | 2.3914 | 0.1677 | 0.6798 | 0.6902 | 0.7251 |
| | SPIN(0.1) | 0.4528 | 2.3733 | 0.1543 | 0.6918 | 0.7035 | 0.7457 |
| | SPIN(0.2) | 0.4593 | 2.3639 | 0.1445 | **0.6996** | **0.7159** | 0.7165 |
| | SPIN(0.3) | 0.4519 | 2.3866 | 0.1466 | 0.6728 | 0.6774 | 0.7715 |
| | SPIN(0.4) | 0.4529 | 2.3903 | 0.1382 | 0.6602 | 0.6692 | 0.7854 |
| | SPIN(0.5) | 0.4353 | 2.4589 | 0.1428 | 0.6520 | 0.6547 | 0.7735 |

Table B-10: Performance evaluation of various priors on **CIFAR-10** ($\sigma = 0.001$). **Bold** entries indicate the best performance (per model, per column; Accuracy/MSP/Entropy/MI ↑, NLL/ECE ↓).

| Model | Prior | Acc(↑) | NLL(↓) | ECE(↓) | MSP(↑) | Entropy(↑) | MI(↑) |
|---|---|---|---|---|---|---|---|
| ResNet 20 | *Iso. Gaussian* | 0.8922 | 0.3402 | 0.0275 | 0.8417 | 0.8564 | 0.8404 |
| | *Laplace* | 0.8931 | 0.3374 | 0.0321 | 0.8406 | 0.8561 | 0.8415 |
| | *Student-t* | 0.8931 | 0.3399 | 0.0282 | 0.8421 | 0.8578 | 0.8418 |
| | *Spike-and-Slab* | 0.8917 | 0.3461 | 0.0243 | 0.8404 | 0.8556 | 0.8411 |
| | *Horseshoe* | 0.8916 | 0.3248 | 0.0610 | 0.8460 | 0.8602 | 0.8409 |
| | MOPED | 0.8907 | 0.3442 | 0.0283 | 0.8403 | 0.8556 | 0.8408 |
| | SPIN (0.0) | 0.9009 | 0.3679 | 0.0444 | 0.8399 | 0.8489 | 0.8390 |
| | SPIN (0.1) | 0.8941 | 0.3553 | 0.0109 | 0.8386 | 0.8527 | 0.8367 |
| | SPIN (0.2) | 0.9099 | 0.3470 | 0.0134 | 0.8556 | 0.8672 | 0.8594 |
| | SPIN (0.3) | 0.9114 | 0.3262 | **0.0087** | 0.8577 | 0.8709 | 0.8602 |
| | SPIN (0.4) | 0.9112 | 0.3254 | 0.0157 | 0.8564 | 0.8713 | 0.8601 |
| | SPIN (0.5) | **0.9114** | 0.3103 | 0.0194 | 0.8563 | 0.8705 | **0.8606** |
| | SPIN (0.6) | 0.9096 | 0.3052 | 0.0217 | 0.8565 | 0.8696 | 0.8546 |
| | SPIN (0.7) | 0.9094 | 0.3026 | 0.0273 | 0.8564 | 0.8701 | 0.8556 |
| | SPIN (0.8) | 0.9107 | **0.3002** | 0.0272 | **0.8581** | **0.8725** | 0.8598 |
| DenseNet 30 | *Iso. Gaussian* | 0.8971 | 0.3322 | 0.0415 | 0.8415 | 0.8561 | 0.8337 |
| | *Laplace* | 0.8946 | 0.3313 | 0.0523 | 0.8344 | 0.8496 | 0.8314 |
| | *Student-t* | 0.8983 | 0.3242 | 0.0450 | 0.8422 | 0.8583 | 0.8433 |
| | *Spike-and-Slab* | 0.8971 | 0.3413 | 0.0320 | 0.8429 | 0.8581 | 0.8435 |
| | *Horseshoe* | 0.8618 | 0.4056 | 0.2141 | 0.8162 | 0.8213 | 0.7705 |
| | MOPED | 0.8969 | 0.3425 | 0.0462 | 0.8397 | 0.8541 | 0.8384 |
| | SPIN (0.0) | 0.9076 | 0.3155 | **0.0148** | 0.8419 | 0.8549 | 0.8436 |
| | SPIN (0.1) | 0.8964 | 0.3265 | 0.0223 | 0.8334 | 0.8489 | 0.8301 |
| | SPIN (0.2) | 0.9020 | 0.3090 | 0.0289 | 0.8397 | 0.8553 | 0.8356 |
| | SPIN (0.3) | **0.9145** | 0.3060 | 0.0259 | 0.8497 | 0.8634 | 0.8500 |
| | SPIN (0.4) | 0.9143 | 0.2901 | 0.0382 | 0.8529 | 0.8666 | 0.8501 |
| | SPIN (0.5) | 0.9131 | **0.2829** | 0.0459 | **0.8578** | **0.8712** | **0.8568** |
| ViT | *Iso. Gaussian* | 0.6179 | 1.0780 | 0.0266 | 0.6682 | 0.6862 | 0.6286 |
| | *Laplace* | 0.6834 | 0.8862 | 0.0146 | 0.7060 | 0.7281 | 0.6817 |
| | *Student-t* | 0.6639 | 0.9472 | 0.0226 | 0.6911 | 0.7107 | 0.6502 |
| | *Spike-and-Slab* | 0.3321 | 1.7888 | 0.0314 | 0.5396 | 0.5443 | 0.5116 |
| | *Horseshoe* | 0.5926 | 1.1060 | 0.2778 | 0.6661 | 0.6850 | 0.6458 |
| | MOPED | 0.5493 | 1.2523 | 0.0266 | 0.6439 | 0.6556 | 0.5843 |
| | SPIN (0.0) | 0.5731 | 1.1826 | 0.0324 | 0.6522 | 0.6707 | 0.6516 |
| | SPIN (0.1) | 0.6786 | 0.8945 | **0.0145** | 0.7034 | 0.7249 | 0.6929 |
| | SPIN (0.2) | 0.7236 | 0.7794 | 0.0203 | 0.7148 | 0.7396 | 0.7020 |
| | SPIN (0.3) | 0.7042 | 0.8315 | 0.0282 | 0.7060 | 0.7290 | 0.6920 |
| | SPIN (0.4) | **0.7467** | **0.7212** | 0.0175 | **0.7238** | **0.7486** | **0.7176** |

Table B-11: Performance evaluation of various priors on **CIFAR-100** ($\sigma = 0.001$). **Bold** entries indicate the best performance (per model, per column; Accuracy/MSP/Entropy/MI ↑, NLL/ECE ↓).

| Model | Prior | Acc(↑) | NLL(↓) | ECE(↓) | MSP(↑) | Entropy(↑) | MI(↑) |
|---|---|---|---|---|---|---|---|
| ResNet 20 | *Iso. Gaussian* | 0.6339 | 1.3668 | **0.0226** | 0.7190 | 0.7366 | 0.7325 |
| | *Laplace* | 0.6316 | 1.3614 | 0.0260 | 0.7218 | 0.7425 | 0.7272 |
| | *Student-t* | 0.6325 | 1.3699 | 0.0257 | 0.7200 | 0.7370 | 0.7281 |
| | *Spike-and-Slab* | 0.6307 | 1.3660 | 0.0264 | 0.7245 | 0.7436 | 0.7370 |
| | *Horseshoe* | 0.6292 | 1.3556 | 0.0428 | 0.7193 | 0.7363 | 0.7262 |
| | MOPED | 0.6396 | 1.3522 | 0.0343 | 0.7262 | 0.7426 | 0.7328 |
| | SPIN (0.0) | 0.6596 | 1.5365 | 0.1310 | 0.7204 | 0.7418 | 0.7216 |
| | SPIN (0.1) | 0.6615 | 1.4174 | 0.0713 | 0.7328 | 0.7556 | 0.7382 |
| | SPIN (0.2) | **0.6639** | **1.3506** | 0.0249 | **0.7373** | **0.7585** | **0.7389** |
| DenseNet 30 | *Iso. Gaussian* | 0.6556 | 1.2860 | 0.0374 | 0.7294 | 0.7462 | 0.7425 |
| | *Laplace* | 0.6516 | 1.3180 | 0.0384 | 0.7237 | 0.7408 | 0.7243 |
| | *Student-t* | 0.6461 | 1.3184 | 0.0393 | 0.7253 | 0.7393 | 0.7267 |
| | *Spike-and-Slab* | 0.6528 | 1.3128 | 0.0238 | 0.7336 | 0.7520 | 0.7399 |
| | *Horseshoe* | 0.6180 | 1.3903 | 0.1379 | 0.7113 | 0.7185 | 0.6937 |
| | MOPED | 0.6483 | 1.3220 | 0.0361 | 0.7276 | 0.7440 | 0.7248 |
| | SPIN (0.0) | 0.6452 | 1.3480 | 0.0389 | 0.7373 | 0.7629 | 0.7259 |
| | SPIN (0.1) | 0.6375 | 1.3617 | **0.0230** | 0.7345 | 0.7569 | 0.7303 |
| | SPIN (0.2) | 0.6773 | 1.3462 | 0.0300 | 0.7428 | 0.7649 | 0.7481 |
| | SPIN (0.3) | 0.6840 | 1.2965 | 0.0266 | 0.7444 | 0.7642 | 0.7454 |
| | SPIN (0.4) | 0.6803 | 1.2521 | 0.0395 | 0.7507 | 0.7686 | 0.7474 |
| | SPIN (0.5) | 0.6832 | 1.2300 | 0.0536 | 0.7534 | 0.7710 | 0.7474 |
| | SPIN (0.6) | 0.6800 | 1.2100 | 0.0560 | 0.7517 | 0.7658 | 0.7428 |
| | SPIN (0.7) | **0.6837** | **1.2025** | 0.0537 | **0.7527** | **0.7699** | **0.7485** |
| ViT | *Iso. Gaussian* | 0.4884 | 1.9897 | 0.0251 | 0.6491 | 0.6646 | 0.6470 |
| | *Laplace* | 0.4926 | 1.9797 | 0.0314 | 0.6604 | 0.6715 | 0.6483 |
| | *Student-t* | 0.4778 | 2.0178 | 0.0238 | 0.6469 | 0.6611 | 0.6515 |
| | *Spike-and-Slab* | 0.4936 | 1.9555 | 0.0254 | 0.6563 | 0.6723 | 0.6544 |
| | *Horseshoe* | 0.4995 | 1.9374 | 0.0217 | 0.6562 | 0.6700 | 0.6512 |
| | MOPED | 0.4939 | 1.9669 | 0.0268 | 0.6551 | 0.6691 | 0.6440 |
| | SPIN (0.0) | **0.5434** | **1.8223** | 0.0868 | 0.6653 | 0.6846 | 0.6721 |
| | SPIN (0.1) | 0.5168 | 1.8533 | 0.0467 | 0.6598 | 0.6774 | 0.6615 |
| | SPIN (0.2) | 0.5161 | 1.8457 | 0.0542 | 0.6664 | 0.6848 | 0.6693 |
| | SPIN (0.3) | 0.5277 | 1.8457 | 0.0646 | 0.6665 | 0.6836 | **0.6719** |
| | SPIN (0.4) | 0.5272 | 1.8527 | **0.0448** | 0.6657 | 0.6830 | 0.6702 |
| | SPIN (0.5) | 0.5265 | 1.8226 | 0.0308 | **0.6689** | **0.6881** | 0.6630 |
| | SPIN (0.6) | 0.5156 | 1.8617 | 0.0510 | 0.6642 | 0.6810 | 0.6645 |
| | SPIN (0.8) | 0.4611 | 2.1428 | 0.0445 | 0.6411 | 0.6541 | 0.6390 |
| | SPIN (0.9) | 0.5022 | 1.9384 | 0.0364 | 0.6561 | 0.6730 | 0.6434 |

Table B-12: Performance evaluation of various priors on **TinyImageNet** ($\sigma = 0.001$). **Bold** entries indicate the best performance (per model, per column; Accuracy/MSP/Entropy/MI ↑, NLL/ECE ↓).

| Model | Prior | Acc(↑) | NLL(↓) | ECE(↓) | MSP(↑) | Entropy(↑) | MI(↑) |
|---|---|---|---|---|---|---|---|
| ResNet 18 | *Iso. Gaussian* | 0.5072 | 2.1029 | 0.0863 | 0.7865 | 0.8230 | 0.5028 |
| | *Laplace* | 0.5200 | 2.0847 | 0.1050 | 0.7968 | 0.8332 | 0.5160 |
| | *Student-t* | 0.5156 | 2.1025 | 0.0906 | 0.7937 | 0.8291 | 0.4968 |
| | *Spike-and-Slab* | 0.5076 | 2.1171 | 0.0858 | 0.7815 | 0.8162 | 0.5209 |
| | *Horseshoe* | 0.5124 | 2.0855 | 0.0794 | 0.7936 | 0.8260 | 0.5714 |
| | MOPED | 0.5122 | 2.1041 | **0.0851** | 0.7676 | 0.7996 | 0.5439 |
| | SPIN (0.0) | 0.6096 | 2.0770 | 0.1831 | 0.7923 | 0.8222 | 0.6534 |
| | SPIN (0.1) | **0.6123** | 2.0632 | 0.1798 | 0.8036 | 0.8359 | **0.6482** |
| | SPIN (0.2) | 0.6062 | 2.0763 | 0.1839 | **0.8071** | 0.8387 | 0.6477 |
| | SPIN (0.3) | 0.5436 | **2.0596** | 0.1279 | 0.8068 | **0.8402** | 0.5545 |
| MobileNet v2 | *Iso. Gaussian* | 0.5675 | 1.7557 | 0.0795 | 0.7753 | 0.8109 | 0.6820 |
| | *Laplace* | 0.5729 | 1.7374 | 0.0748 | 0.7846 | 0.8232 | 0.6647 |
| | *Student-t* | 0.5859 | 1.6876 | 0.0788 | 0.7915 | 0.8281 | 0.6639 |
| | *Spike-and-Slab* | 0.5785 | 1.7062 | 0.0608 | 0.7893 | 0.8260 | 0.6644 |
| | *Horseshoe* | 0.5415 | 1.8492 | 0.1430 | 0.5888 | 0.5104 | 0.9634 |
| | MOPED | 0.5714 | 1.7343 | 0.0821 | 0.7771 | 0.8137 | 0.7039 |
| | SPIN (0.0) | 0.6424 | 1.5021 | **0.0130** | 0.8099 | 0.8431 | 0.6946 |
| | SPIN (0.1) | **0.6450** | 1.4651 | 0.0514 | **0.8185** | **0.8530** | 0.6990 |
| | SPIN (0.2) | 0.6424 | **1.4528** | 0.0638 | 0.8174 | 0.8505 | 0.6944 |
| | SPIN (0.3) | 0.6315 | 1.4864 | 0.0796 | 0.8051 | 0.8367 | **0.7120** |
| ViT | *Iso. Gaussian* | 0.4516 | 2.3898 | 0.1302 | 0.6670 | 0.6898 | 0.8204 |
| | *Laplace* | 0.4544 | 2.3801 | 0.1327 | 0.6682 | 0.6790 | 0.8069 |
| | *Student-t* | 0.4637 | 2.3266 | 0.1188 | 0.6908 | 0.7075 | 0.7973 |
| | *Spike-and-Slab* | 0.4691 | **2.3208** | 0.1083 | 0.6927 | 0.7030 | **0.8633** |
| | *Horseshoe* | 0.5022 | 1.3984 | 0.0364 | 0.6561 | 0.6730 | 0.6434 |
| | MOPED | 0.4512 | 2.3976 | 0.1282 | 0.6681 | 0.6923 | 0.7760 |
| | SPIN (0.0) | 0.0926 | 4.3723 | **0.0579** | 0.3991 | 0.3816 | 0.4848 |
| | SPIN (0.1) | 0.2888 | 3.1729 | 0.1141 | 0.4150 | 0.3753 | 0.7133 |
| | SPIN (0.2) | 0.4359 | 2.4588 | 0.1198 | 0.6394 | 0.6463 | 0.7266 |
| | SPIN (0.3) | 0.4647 | 2.3756 | 0.1052 | 0.6835 | 0.6976 | 0.7542 |
| | SPIN (0.4) | 0.4508 | 2.4089 | 0.1248 | 0.6577 | 0.6659 | 0.7526 |
| | SPIN (0.5) | **0.4706** | 2.3441 | 0.1108 | **0.6997** | **0.7187** | 0.6725 |

## C FURTHER EXPERIMENTS ON DIVERSE TASKS

To thoroughly assess the versatility and effectiveness of our proposed prior, we conduct a comprehensive evaluation across a diverse spectrum of tasks. Our experiments are designed to demonstrate the prior's applicability, starting from fundamental neural network architectures like MLPs on classification and regression tasks, and extending to complex, high-level computer vision challenges such as semantic segmentation. This series of experiments validates the robustness and generalizability of our approach in various settings.

### C.1 MULTI LAYER PERCEPTRON PERFORMANCE

To demonstrate the broad applicability of our method, we first conduct experiments on a multi layer perceptron (MLP), one of the most fundamental neural network architectures.

#### C.1.1 MNIST CLASSIFICATION

We additionally evaluate our proposed prior on the widely-used **MNIST** dataset for image classification. The model is a **MLP** with an input dimension of 784 ($28 \times 28$), two hidden layers with 200 and 100 units respectively, and a final output layer with 10 units for classification. Each hidden layer is followed by batch normalization and a ReLU activation function. Similar to our other experiments, we apply variational weight reparameterization to all fully connected layers and approximate the posterior predictive distribution with 30 *Monte Carlo* forward passes during inference. For OOD detection, we use the **Fashion-MNIST** dataset.

Table C-1: Performance evaluation of various priors on **MNIST** ($\sigma = 0.001$). **Bold** entries indicate the best performance (per model, per column; Accuracy/MSP/Entropy/MI $\uparrow$, NLL/ECE $\downarrow$). Note that ($\cdot$) indicates the `sparsity`.

| Model | Prior | Acc($\uparrow$) | NLL($\downarrow$) | ECE($\downarrow$) | MSP($\uparrow$) | Entropy($\uparrow$) | MI($\uparrow$) |
|---|---|---|---|---|---|---|---|
| | *Iso. Gaussian* | 0.9683 | 0.0892 | **0.0091** | 0.7973 | 0.7950 | 0.8078 |
| | *Laplace* | 0.9741 | 0.0795 | 0.0131 | 0.8186 | 0.8165 | 0.8308 |
| | *Student-t* | 0.9695 | 0.0852 | 0.0075 | 0.8019 | 0.8011 | 0.8147 |
| | *Spike-and-Slab* | 0.9693 | 0.0863 | 0.0103 | 0.7903 | 0.7876 | 0.8005 |
| | MOPED | 0.9699 | 0.0864 | 0.0096 | 0.8244 | 0.8227 | 0.8325 |
| | SPIN (`0.0`) | 0.9778 | 0.0658 | 0.0119 | 0.8061 | 0.8042 | 0.8039 |
| MLP | SPIN (`0.1`) | **0.9781** | **0.0662** | 0.0143 | 0.8078 | 0.8052 | 0.8076 |
| | SPIN (`0.2`) | 0.9767 | 0.0659 | 0.0150 | 0.8411 | 0.8393 | 0.8410 |
| | SPIN (`0.3`) | 0.9761 | 0.0683 | 0.0171 | 0.8317 | 0.8278 | 0.8307 |
| | SPIN (`0.4`) | 0.9758 | 0.0688 | 0.0203 | 0.8329 | 0.8296 | 0.8325 |
| | SPIN (`0.5`) | 0.9762 | 0.0712 | 0.0203 | 0.8470 | 0.8440 | 0.8454 |
| | SPIN (`0.6`) | 0.9723 | 0.0789 | 0.0195 | 0.8492 | 0.8467 | 0.8468 |
| | SPIN (`0.7`) | 0.9724 | 0.0811 | 0.0182 | **0.8508** | **0.8473** | **0.8510** |

#### C.1.2 REGRESSION TASKS

Table C-2: Performance comparison of different priors on a regression task. **Bold** indicates the best result, and underlined indicates the second best. Note that ($\cdot$) indicates the `sparsity`.

| Model | Prior | RMSE($\downarrow$) |
|---|---|---|
| | *Iso. Gaussian* | 0.7357 |
| | MOPED | 0.7384 |
| MLP | SPIN (`0.0`) | 0.7139 |
| | SPIN (`0.3`) | **0.6914** |
| | SPIN (`0.6`) | 0.7038 |
| | SPIN (`0.9`) | 0.8030 |

To evaluate the generality of our prior-setting approach beyond classification, we further test it on a regression task using the **California Housing** dataset. We employ a **MLP** comprising three

fully connected layers, each equipped with variational weight reparameterization. *Batch Normalization* (Ioffe & Szegedy, 2015) is applied after the first and second layers to stabilize training and improve convergence. Training is performed with *Monte Carlo* sampling of 30 forward passes to approximate the posterior predictive distribution. SPIN parameter $\sigma$ is set as 0.001.

## C.2 SEMANTIC SEGMENTATION

We evaluated the performance of BNNs with the proposed method and other baseline priors for the task of image segmentation. We employed a U-Net architecture with a MobileNetv2 encoder backbone and trained the network on the Cityscapes dataset (Cordts et al., 2016). We used simple data augmentation (i.e., random horizontal flip and color jittering). All models are trained for 50 epochs using the Adam (Kingma & Ba, 2014) optimizer with a learning rate of 0.0001 and a batch size of 4. During inference, we perform 30 Monte Carlo forward passes to approximate the predictive distribution.

Table C-3: Performance of different priors and `sparsity` levels on Cityscapes (Cordts et al., 2016) on **U-Net** with encoder **MobileNet-v2**. Pixel Acc measures the ratio of correctly classified pixels, mIoU (mean Intersection over Union) is the average class-wise IoU, and FWIoU (frequency-weighted IoU) weights IoU by class frequency. **Bold** entries denote the best results, and underlined entries denote the second best.

| Model | Pixel Acc (↑) | mIoU (↑) | FWIoU (↑) |
|---|---|---|---|
| *Iso. Gaussian* | 0.6108 | 0.1190 | 0.4531 |
| *Laplace* | 0.6413 | 0.1280 | 0.4896 |
| *Student-t* | 0.7290 | 0.1588 | 0.5848 |
| *Spike-and-Slab* | 0.6763 | 0.1386 | 0.5272 |
| MOPED | 0.6689 | 0.1396 | 0.5246 |
| SPIN (0.0) | 0.4936 | 0.0754 | 0.3389 |
| SPIN (0.1) | 0.7413 | 0.1593 | 0.5962 |
| SPIN (0.2) | **0.7575** | **0.1838** | **0.6193** |
| SPIN (0.3) | 0.6566 | 0.1162 | 0.5022 |
| SPIN (0.4) | 0.5810 | 0.1332 | 0.4341 |
| SPIN (0.5) | 0.5877 | 0.0990 | 0.4305 |
| SPIN (0.6) | 0.6088 | 0.1026 | 0.4633 |

Table C-4: Qualitative results of image segmentation with different priors

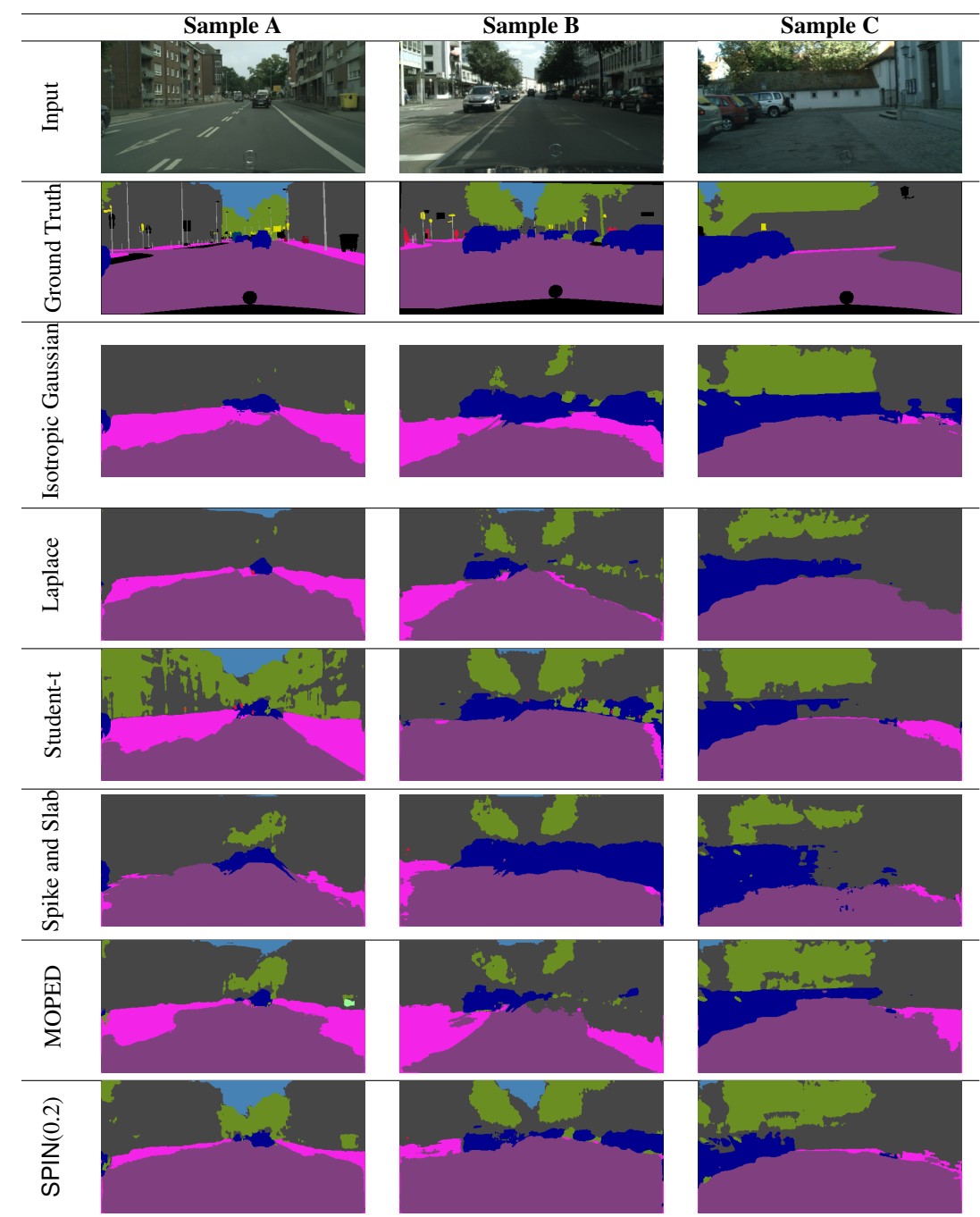

# D ADDITIONAL ANALYSIS RESULTS

## D.1 ADDITIONAL RESULTS WITH UMAP

**Implementation Detail** We evaluate the quality of unsupervised representations by extracting features from trained BNN models from SPIN (with $\sigma = 0.001$) and projecting them into two dimensions using *UMAP* embeddings (McInnes et al., 2018) for visualization on **CIFAR-10** (Fig. D-1) and **SVHN** (Fig. D-2). Comparisons with different prior settings are provided in Table D-5 for **CIFAR-10** and Table D-6 for **SVHN**.

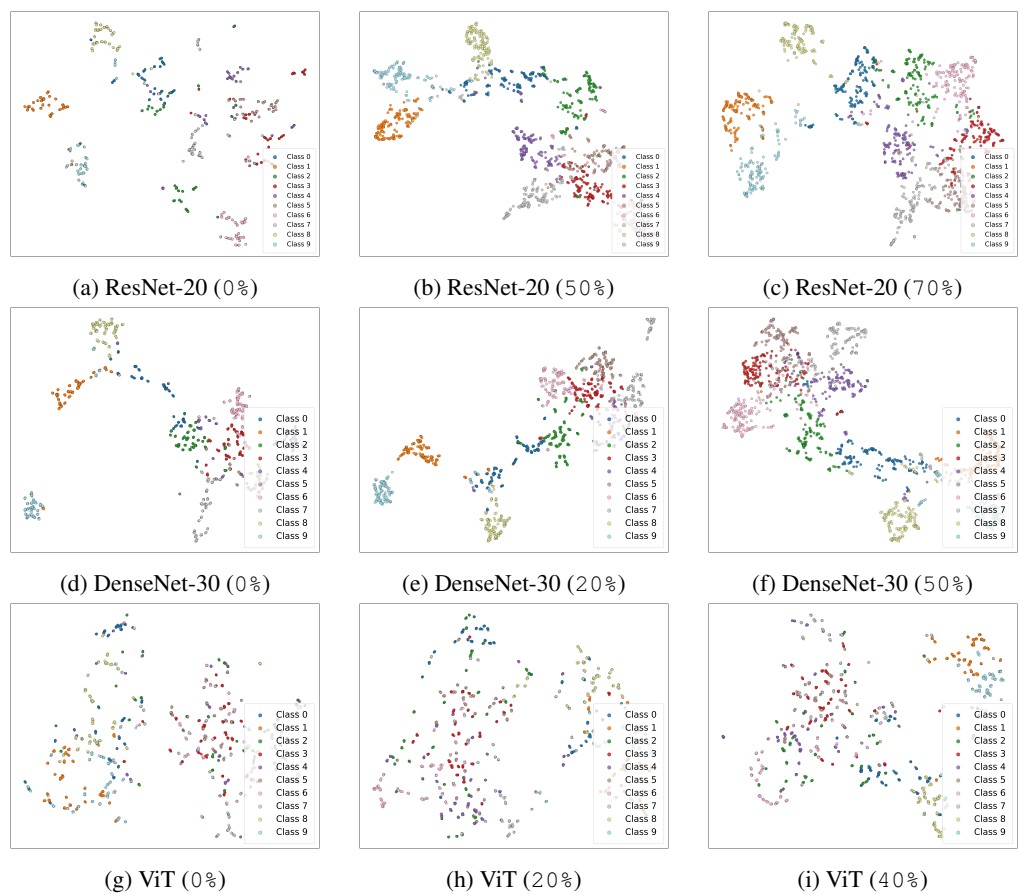

(a) ResNet-20 (`0%`)  (b) ResNet-20 (`50%`)  (c) ResNet-20 (`70%`)

(d) DenseNet-30 (`0%`)  (e) DenseNet-30 (`20%`)  (f) DenseNet-30 (`50%`)

(g) ViT (`0%`)  (h) ViT (`20%`)  (i) ViT (`40%`)

Figure D-1: *UMAP* McInnes et al. (2018) visualization of **CIFAR-10** features extracted by three architectures under varying `sparsity`: **ResNet-20** (top row: 0%, 50%, 70%), **DenseNet-30** (middle row: 0%, 20%, 50%), and **ViT** (bottom row: 0%, 20%, 40%).

Table D-5: **ResNet-20**, **DenseNet-30**, and **ViT** on **CIFAR-10** with *UMAP* embeddings, evaluated by *Davies–Bouldin Index* (lower is better). **Bold** marks the best and underline the second-best prior for each model. Note that (·) indicates the `sparsity`.

| Prior | ResNet-20 | DenseNet-30 | ViT |
|---|---|---|---|
| *Isotropic Gaussian* | 1.26 | 1.14 | 8.90 |
| *Laplace* | 0.97 | 1.28 | 11.10 |
| *Student-t* | 1.33 | 1.24 | 5.54 |
| MOPED | 1.64 | 1.51 | 13.14 |
| *Spike-and-Slab* | 1.21 | 1.25 | 11.49 |
| SPIN (low) | 2.09 (`0`) | 1.30 (`0`) | 10.04 (`0`) |
| SPIN (mid) | 1.03 (`50`) | **0.97** (`20`) | 7.02 (`20`) |
| SPIN (high) | **0.90** (`70`) | 1.22 (`50`) | **3.31** (`40`) |

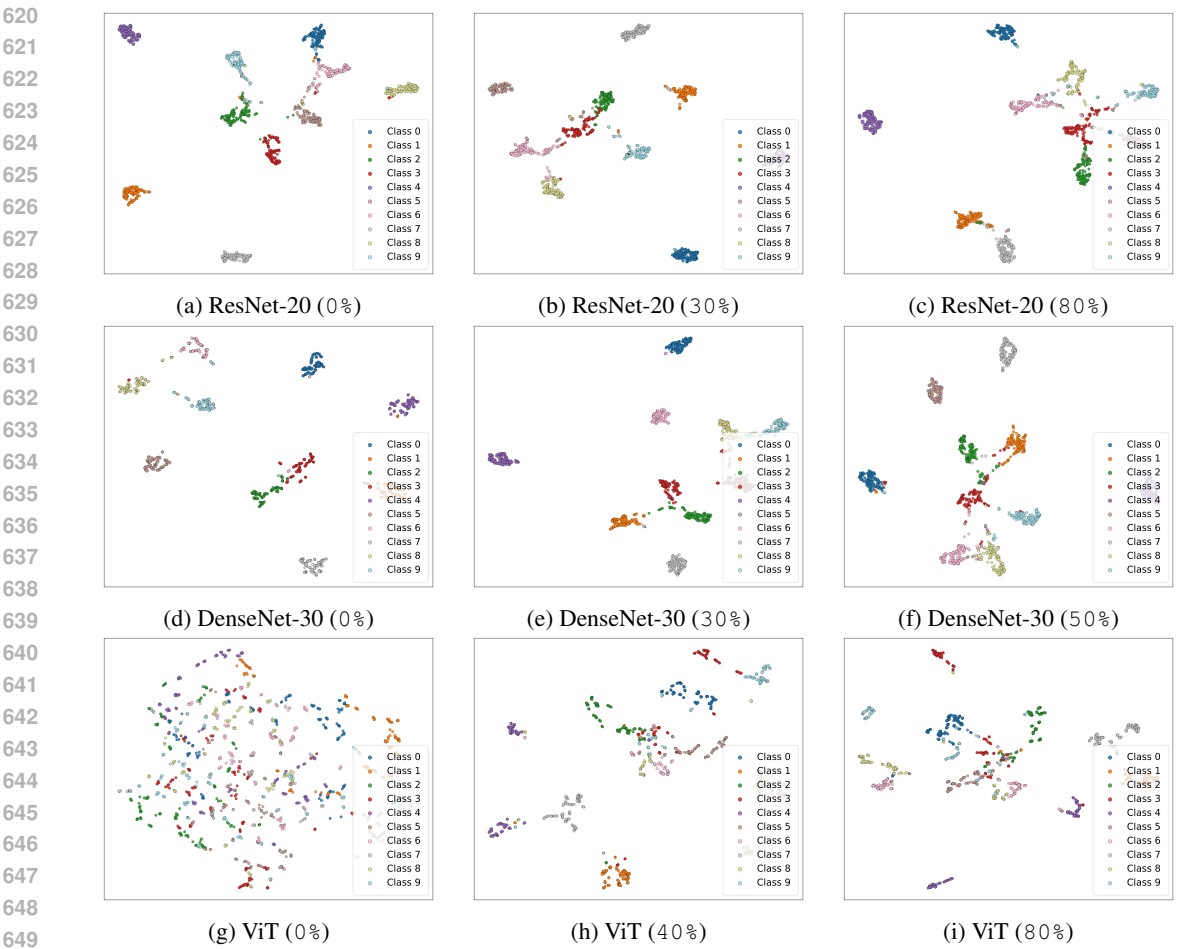

Figure D-2: *UMAP* McInnes et al. (2018) visualization of **SVHN** features across three architectures under varying `sparsity` levels. Top row: **ResNet-20** (0%, 30%, 80%), middle row: **DenseNet-30** (0%, 30%, 50%), bottom row: **ViT** (0%, 40%, 80%).

Table D-6: **ResNet-20**, **DenseNet-30**, and **ViT** on **SVHN** with *UMAP* embeddings, evaluated by *Davies–Bouldin Index* (lower is better). **Bold** marks the best and underline the second-best prior for each model. Note that (·) indicates the `sparsity`.

| Prior | ResNet-20 | DenseNet-30 | ViT |
|---|---|---|---|
| *Isotropic Gaussian* | 0.659 | 0.633 | 13.329 |
| *Laplace* | 0.491 | 0.886 | 9.489 |
| *Student-t* | 0.529 | 1.232 | 4.317 |
| MOPED | 0.574 | 0.719 | 13.188 |
| *Spike-and-Slab* | 0.472 | 0.746 | 12.972 |
| SPIN (low) | 0.515 (0) | 0.471 (0) | 4.203 (0) |
| SPIN (mid) | **0.394** (30) | **0.441** (30) | 4.046 (40) |
| SPIN (high) | 0.610 (80) | 0.588 (50) | **3.047** (80) |

### D.2 OTHER STANDARD DEVIATION SETTINGS OF SPIN

One might question the validity of assigning a high standard deviation to pruned weights (treating them as *less important*) while assigning a low standard deviation to the remaining weights (treating them as *more important*). To test this, we conducted experiments with the *opposite configuration*, where $\sigma \geq 1$, using **ResNet-20**, **DenseNet-30** and **ViT** on **CIFAR-10** (Fig. D-3). While our proposed setting with $\sigma \ll 1$ (i.e., $\sigma = 0.001$) led to performance improvements as `sparsity` changes, the reverse setting (i.e., $\sigma = 1, 10, 100$) failed to outperform the low-variance baseline ($\sigma = 0.001$). These results confirm that the proposed setting ($\sigma \ll 1$) and the underlying assumption—that *pruned weights are less important while the remaining weights are critical*—are *indeed valid*.

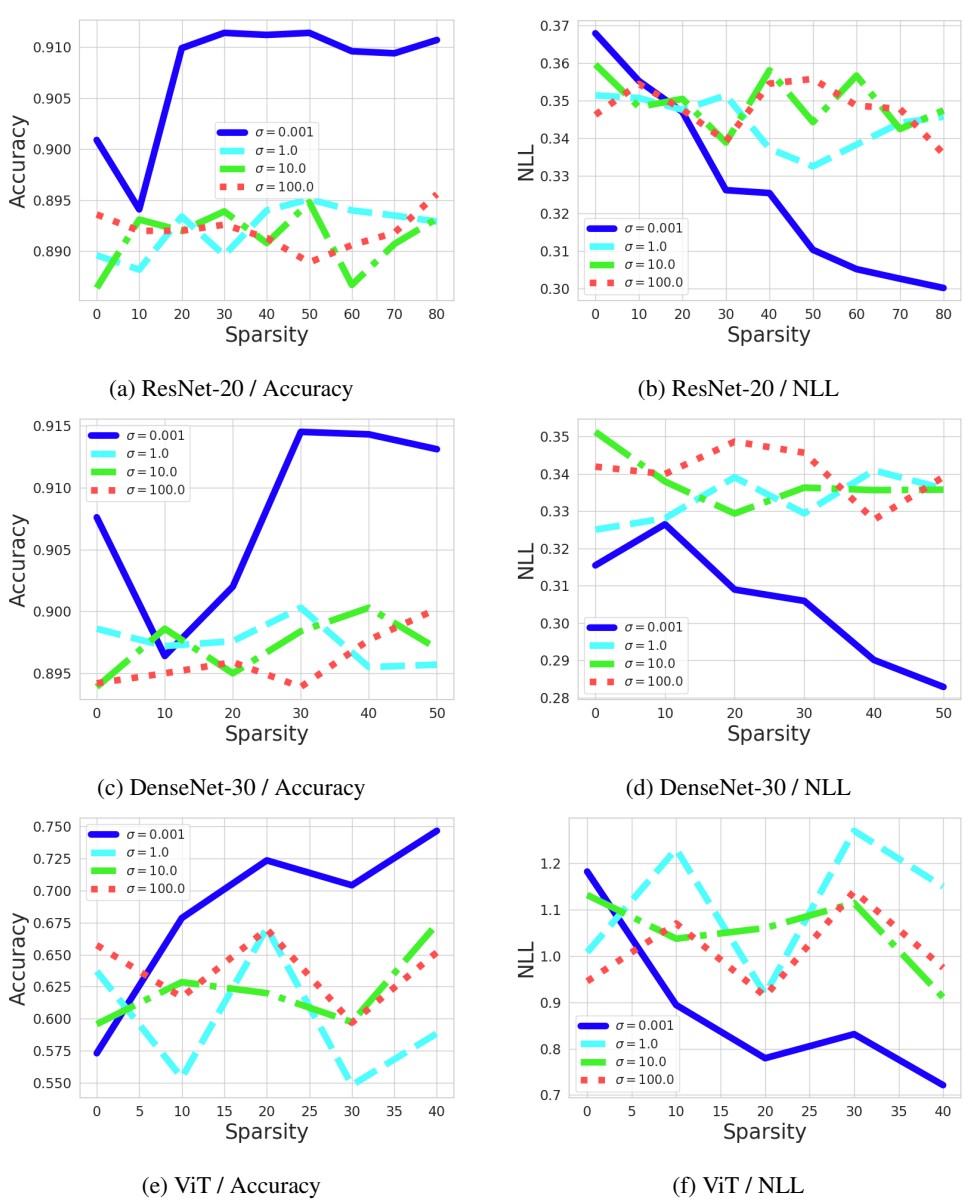

Figure D-3: Effect of different $\sigma$ values assigned to pruned weights under varying `sparsity` on **CIFAR-10**. Rows correspond to models (**ResNet-20**, **DenseNet-30**, **ViT**), and columns show Accuracy (left) and NLL (right).

### D.3 Loss landscape of BNN and adversarial perturbations

**Implementation Detail**    To visualize the loss landscape of BNN, we separately plot the NLL term of the loss function. Following the *filter normalization* technique proposed by Li et al. (2018), we normalize only the convolutional and linear filters while excluding biases and batch normalization layers, referring to the `--xignore biasbn` option in the open-source library[5]. The loss surface is computed over 5 randomly sampled basis planes, with each landscape obtained via *Monte Carlo* sampling using 20 forward passes (see Fig. D-4 for 2D plots and Fig. D-5 for 3D plots).

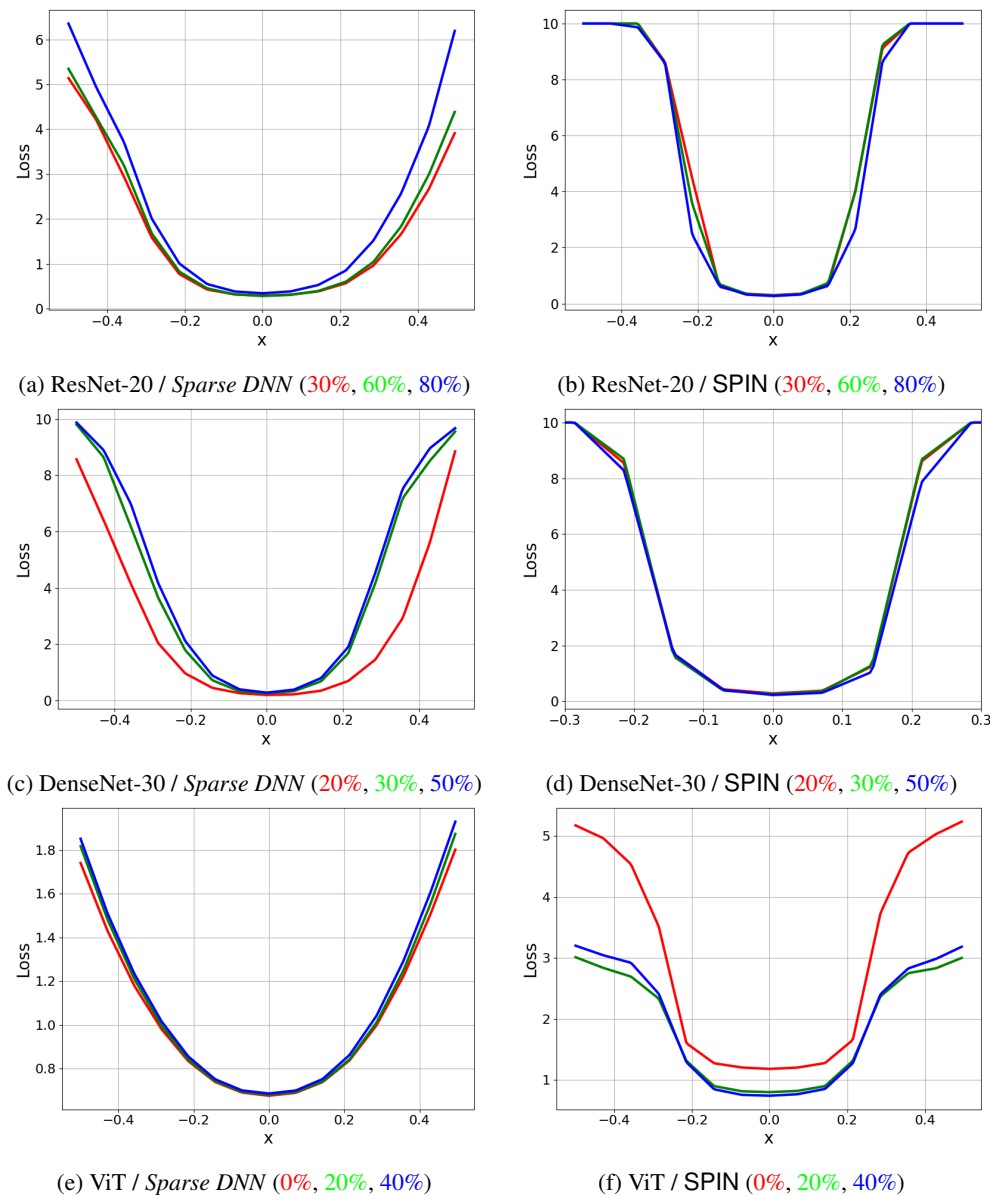

(a) ResNet-20 / *Sparse DNN* (30%, 60%, 80%)

(b) ResNet-20 / SPIN (30%, 60%, 80%)

(c) DenseNet-30 / *Sparse DNN* (20%, 30%, 50%)

(d) DenseNet-30 / SPIN (20%, 30%, 50%)

(e) ViT / *Sparse DNN* (0%, 20%, 40%)

(f) ViT / SPIN (0%, 20%, 40%)

Figure D-4: 2D plots of the NLL loss landscape on **CIFAR-10**. Each row corresponds to a model (**ResNet-20**, **DenseNet-30**, **ViT**), comparing the *Sparse DNN* (left) and the BNN from SPIN (right). The colors in each subfigure caption indicate the corresponding `sparsity`.

---

[5] https://github.com/tomgoldstein/loss-landscape

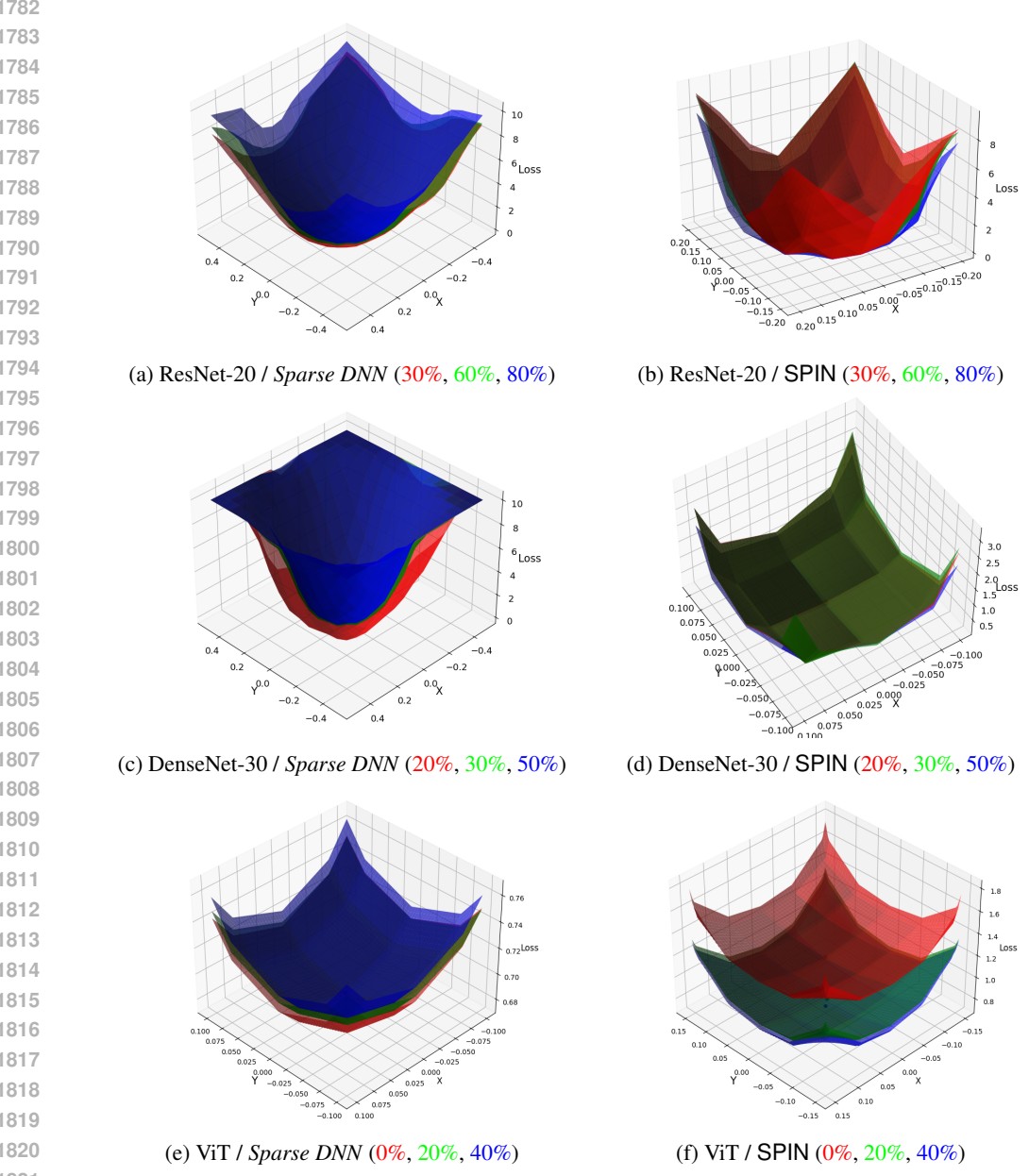

(a) ResNet-20 / *Sparse DNN* (30%, 60%, 80%)

(b) ResNet-20 / SPIN (30%, 60%, 80%)

(c) DenseNet-30 / *Sparse DNN* (20%, 30%, 50%)

(d) DenseNet-30 / SPIN (20%, 30%, 50%)

(e) ViT / *Sparse DNN* (0%, 20%, 40%)

(f) ViT / SPIN (0%, 20%, 40%)

Figure D-5: 3D plots of the NLL loss landscape on **CIFAR-10**. Each row corresponds to a model (**ResNet-20**, **DenseNet-30**, **ViT**), comparing the *Sparse DNN* (left) and the BNN from SPIN (right). The colors in each subfigure caption indicate the corresponding sparsity.

## D.4 PERFORMANCE GAP BETWEEN SPARSE DNN AND BNN FROM SPIN

We conduct additional experiments to further investigate the performance gap between the *sparse DNN* and the BNN from SPIN. Specifically, we evaluate **ResNet-20**, **DenseNet-30**, and **ViT** on **CIFAR-10** (Fig. D-6), **CIFAR-100** (Fig. D-7), and **TinyImageNet**(Fig. D-8). Note that $\sigma$ is fixed at 0.001 throughout the experiments.

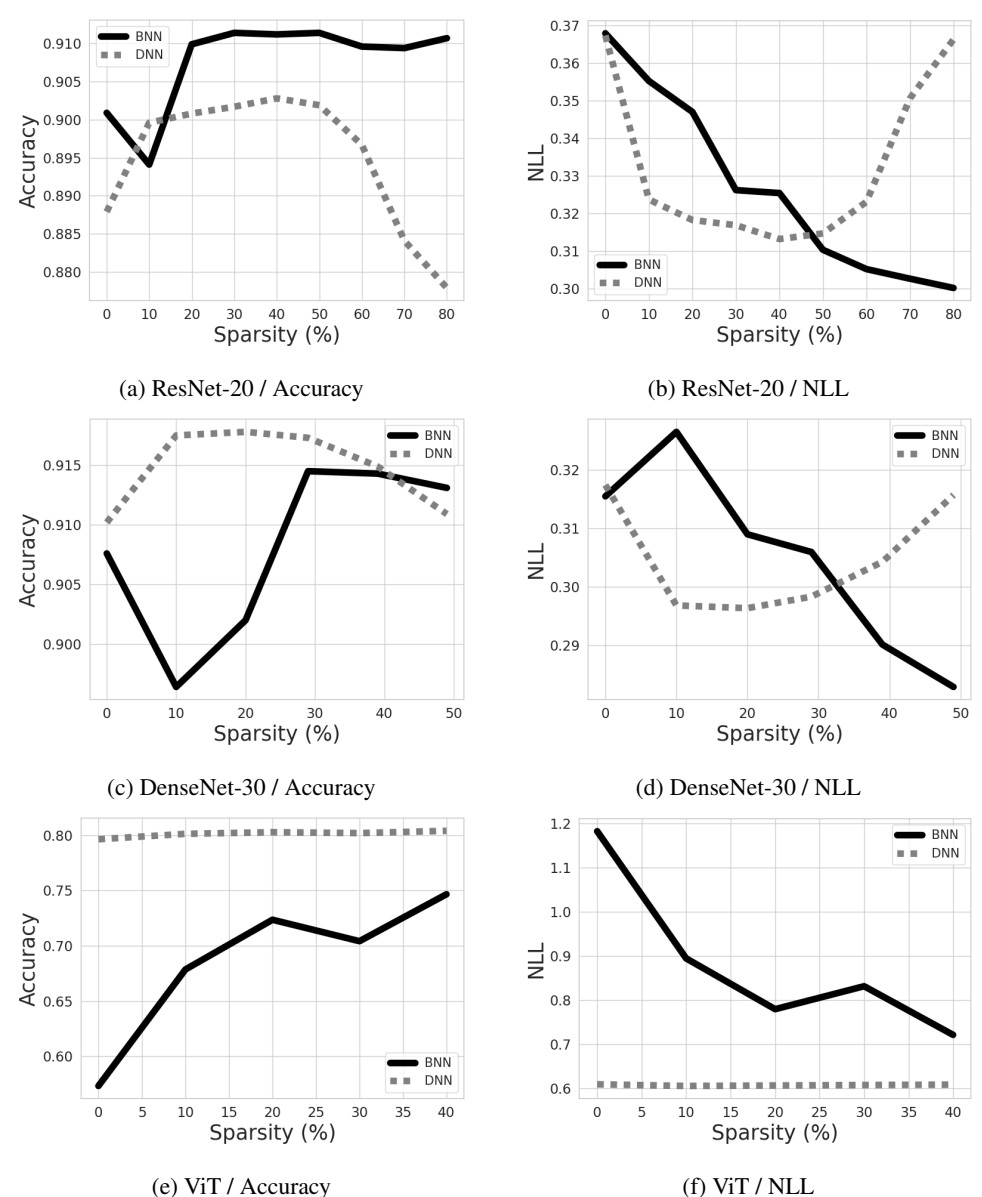

(a) ResNet-20 / Accuracy

(b) ResNet-20 / NLL

(c) DenseNet-30 / Accuracy

(d) DenseNet-30 / NLL

(e) ViT / Accuracy

(f) ViT / NLL

Figure D-6: Performance comparison between *sparse DNN* and BNN from SPIN across varying `sparsity` on **CIFAR-10**. Rows correspond to different architectures: **ResNet-20** (top), **DenseNet-30** (middle), and **ViT** (bottom). Columns show (left) accuracy and (right) NLL.

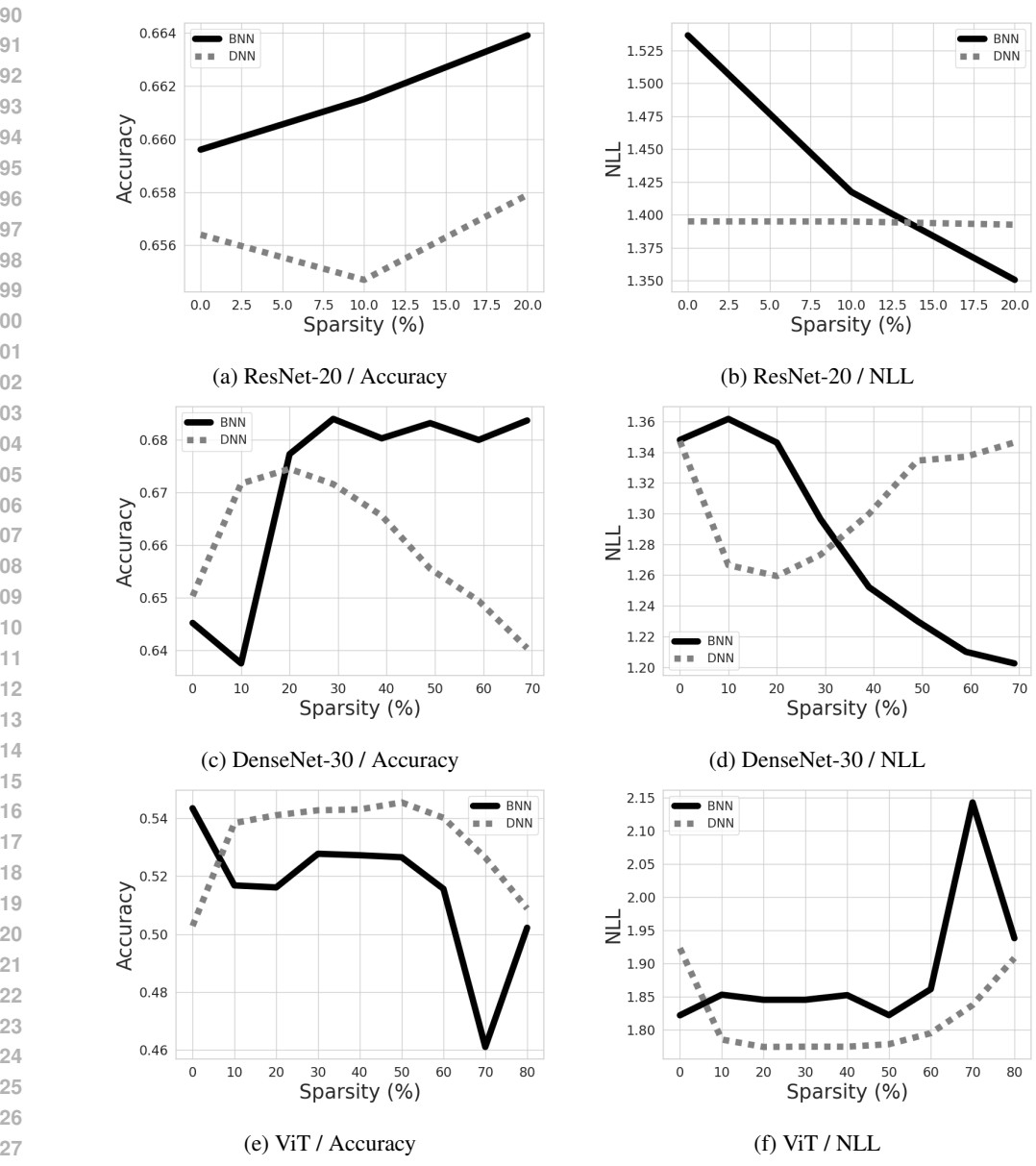

(a) ResNet-20 / Accuracy

(b) ResNet-20 / NLL

(c) DenseNet-30 / Accuracy

(d) DenseNet-30 / NLL

(e) ViT / Accuracy

(f) ViT / NLL

Figure D-7: Performance comparison between *sparse DNN* and BNN from SPIN across varying `sparsity` on **CIFAR-100**. Rows correspond to different architectures: **ResNet-20** (top), **DenseNet-30** (middle), and **ViT** (bottom). Columns show (left) accuracy and (right) NLL.

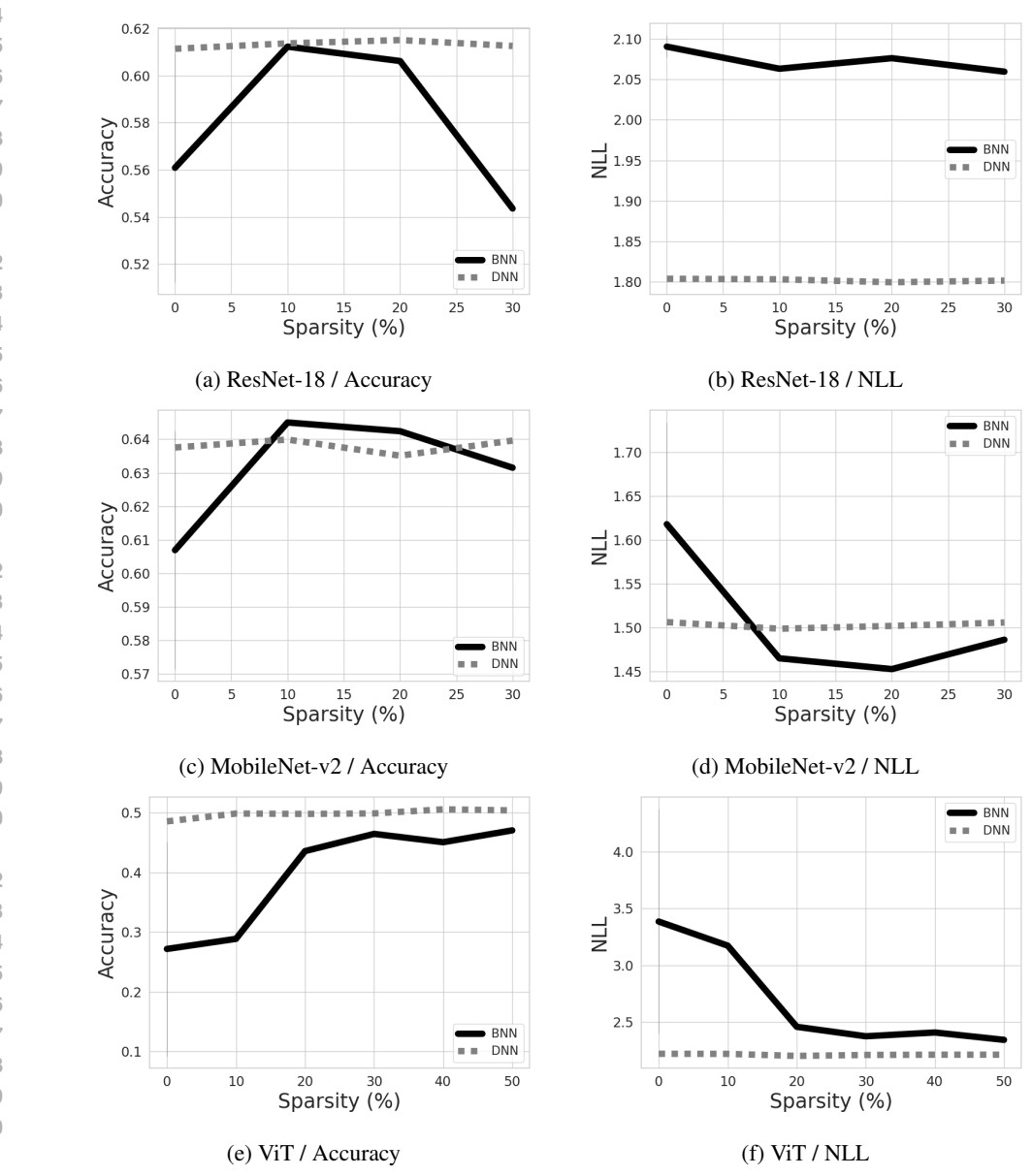

(a) ResNet-18 / Accuracy

(b) ResNet-18 / NLL

(c) MobileNet-v2 / Accuracy

(d) MobileNet-v2 / NLL

(e) ViT / Accuracy

(f) ViT / NLL

Figure D-8: Performance comparison between *sparse DNN* and BNN from SPIN across varying `sparsity` on **TinyImageNet**. Rows correspond to different architectures: **ResNet-18** (top), **MobileNet-v2** (middle), and **ViT** (bottom). Columns show (left) accuracy and (right) NLL.

## D.5 CALIBRATION ISSUE AND THE EFFECT OF TEMPERATURE SCALING

Although SPIN achieves strong accuracy and NLL, its ECE is relatively high due to overconfident predictions. Applying post-hoc temperature scaling (Guo et al., 2017) with $T = 1.5$ substantially reduces ECE ($0.0272 \rightarrow 0.0088$) while leaving accuracy and NLL nearly unchanged (see Table D-7).

Formally, given $S$ Monte Carlo samples $\{\mathbf{w}^{(s)}\}_{s=1}^{S}$ from the variational posterior $q_\phi(\mathbf{w})$, we compute the averaged logit

$$\bar{\mathbf{z}}(\mathbf{x}) = \frac{1}{S} \sum_{s=1}^{S} f_{\mathbf{w}^{(s)}}(\mathbf{x}),$$

where $f_{\mathbf{w}^{(s)}}(\mathbf{x})$ denotes the logit vector under parameters $\mathbf{w}^{(s)}$. Temperature scaling then modifies the predictive distribution as

$$p_T(y \mid \mathbf{x}) = \mathrm{softmax}\left(\frac{\bar{\mathbf{z}}(\mathbf{x})}{T}\right)_y = \frac{\exp\big(\bar{z}_y(\mathbf{x})/T\big)}{\sum_j \exp\big(\bar{z}_j(\mathbf{x})/T\big)}.$$

Here, $T > 1$ produces a smoother distribution with reduced confidence, which improves calibration without altering predictive accuracy or NLL.

Table D-7: Performance of various priors on **CIFAR-10** ($\sigma = 0.001$, **ResNet-20**). **Bold** entries denote the best results, and underlined entries denote the second best. $+T$ indicates temperature scaling ($T = 1.5$).

| Model | Prior | Acc(↑) | NLL(↓) | ECE(↓) | MSP(↑) | ENT(↑) | MI(↑) |
|---|---|---|---|---|---|---|---|
| ResNet 20 | *Iso. Gaussian* | 0.8922 | 0.3402 | 0.0275 | 0.8417 | 0.8564 | 0.8404 |
| | *Laplace* | 0.8931 | 0.3374 | 0.0321 | 0.8406 | 0.8561 | 0.8415 |
| | *Student-t* | 0.8931 | 0.3399 | 0.0282 | 0.8421 | 0.8578 | 0.8418 |
| | *Spike-and-Slab* | 0.8917 | 0.3461 | 0.0243 | 0.8404 | 0.8556 | 0.8411 |
| | MOPED | 0.8907 | 0.3442 | 0.0283 | 0.8403 | 0.8556 | 0.8408 |
| | SPIN (0.8) | **0.9107** | 0.3001 | 0.0272 | **0.8581** | **0.8725** | **0.8598** |
| | SPIN (0.8 $+T$) | 0.9095 | **0.2986** | **0.0088** | 0.8572 | 0.8717 | 0.8590 |

# E  THEORETICAL ANALYSIS OF HYPERPARAMETER $\sigma$ IN LINEAR MODEL

In this section, we develop a tractable linear-model analysis of the SPIN prior hyperparameter $\sigma$ and its effect on test-time performance. Directly analyzing the full nonlinear deep network is intractable, so we instead consider an overparameterized linear regression model that captures the local quadratic behavior of the loss around the dense solution and its pruned counterpart. Within this setting, we obtain closed-form expressions for the posterior mean and covariance under the SPIN prior, derive a per-coordinate bias–variance decomposition of the expected excess test error, and compare the strong- and weak-prior limits as $\sigma^2 \to 0$ and $\sigma^2 \to \infty$. Finally, we connect these expressions to the validation-loss constraint used by SPIN and show that, under mild conditions, a smaller $\sigma$ is theoretically justified whenever pruning preserves validation performance.

## E.1  PRELIMINARY

**Flattened linear model.**  We consider a standard linear regression model obtained by flattening the weight matrix $W$ into a vector. Let
$$w \in \mathbb{R}^p$$
denote the (vectorized) parameter, and let the dataset be given by
$$X \in \mathbb{R}^{N \times p}, \qquad y \in \mathbb{R}^N.$$
The data are generated according to
$$y = Xw^\star + \varepsilon, \qquad \varepsilon \sim \mathcal{N}(0, \tau^2 I_N), \tag{1}$$
where
$$w^\star \in \mathbb{R}^p$$
is the *test-optimal* parameter and $\tau^2 > 0$ is the noise variance.

**Expected test error.**  We define the per-example negative log-likelihood as
$$\ell(w; x, y) := \frac{1}{2\tau^2}\left(y - x^\top w\right)^2.$$
The *expected test error* is
$$L(w) := \mathbb{E}_{(x,y) \sim P_{\text{data}}}\left[\ell(w; x, y)\right]. \tag{2}$$
By definition,
$$w^\star \in \arg\min_w L(w).$$

**Quadratic approximation of expected test error.**  We assume that $L(w)$ is twice differentiable and that around $w^\star$ it admits a quadratic approximation
$$L(w) \approx L(w^\star) + \frac{1}{2}(w - w^\star)^\top H(w - w^\star), \tag{3}$$
where $H \in \mathbb{R}^{p \times p}$ is the Hessian of $L$ at $w^\star$:
$$H := \nabla^2 L(w^\star),$$
and we assume $H$ is symmetric positive semi-definite.

**Overparameterization and DNN solution.**  We consider the overparameterized regime
$$p > N,$$
in which the least-squares problem (Schaeffer et al., 2023)
$$\min_w \frac{1}{2\tau^2}\|Xw - y\|_2^2$$
has infinitely many global minimizers. Gradient descent starting from $w = 0$ converges to the minimum-norm solution given by the Moore–Penrose pseudoinverse:
$$w_{\text{DNN}} := X^+ y = X^\top (XX^\top)^{-1} y, \tag{4}$$
which we interpret as the trained dense DNN parameter vector.

**Generalization gap of the dense solution.** We define the generalization gap of the dense DNN solution as

$$\delta := w_{\text{DNN}} - w^\star. \tag{5}$$

**SPIN pruning and sparse solution.** Starting from the dense solution $w_{\text{DNN}}$, SPIN performs pruning under a validation-loss constraint and produces a sparse solution

$$w_{\text{sparse}} \in \mathbb{R}^p.$$

We assume that the validation loss is nearly preserved:

$$\ell_{\text{val}}(w_{\text{sparse}}) - \ell_{\text{val}}(w_{\text{DNN}}) = \varepsilon, \qquad \varepsilon \approx 0. \tag{6}$$

We define the *pruning displacement* as

$$\Delta := w_{\text{sparse}} - w_{\text{DNN}}. \tag{7}$$

**Important vs. redundant coordinates.** SPIN induces a partition of coordinates into important and redundant sets:

$$I := \{j : \text{weight } j \text{ kept (important)}\}, \qquad R := \{1, \ldots, p\} \setminus I,$$

and we decompose

$$w = \begin{bmatrix} w_I \\ w_R \end{bmatrix}, \quad w_{\text{DNN}} = \begin{bmatrix} (w_{\text{DNN}})_I \\ (w_{\text{DNN}})_R \end{bmatrix}, \quad w_{\text{sparse}} = \begin{bmatrix} (w_{\text{sparse}})_I \\ 0_R \end{bmatrix}.$$

**SPIN prior.** SPIN uses $w_{\text{sparse}}$ as the prior mean. Along important coordinates $j \in I$, the prior variance is a hyperparameter $\sigma^2$; along redundant coordinates $j \in R$, the prior variance is fixed to 1. Thus

$$p(w) = \prod_{j \in I} \mathcal{N}\big(w_j \mid (w_{\text{sparse}})_j, \sigma^2\big) \prod_{j \in R} \mathcal{N}(w_j \mid 0, 1). \tag{8}$$

In vector form,

$$p(w) = \mathcal{N}(w \mid \mu_{\text{prior}}, \Sigma_{\text{prior}}), \qquad \mu_{\text{prior}} := w_{\text{sparse}}, \tag{9}$$

with covariance

$$\Sigma_{\text{prior}} := \begin{bmatrix} \sigma^2 I_{|I|} & 0 \\ 0 & I_{|R|} \end{bmatrix},$$

and precision

$$\Lambda_{\text{prior}} := \Sigma_{\text{prior}}^{-1} = \begin{bmatrix} \frac{1}{\sigma^2} I_{|I|} & 0 \\ 0 & I_{|R|} \end{bmatrix}. \tag{10}$$

**Local quadratic likelihood around $w_{\text{DNN}}$.** The negative log-likelihood for the linear-Gaussian model is

$$\ell_N(w) := \frac{1}{2\tau^2} \|Xw - y\|_2^2.$$

Around the dense solution $w_{\text{DNN}}$, we approximate

$$\ell_N(w) \approx \ell_N(w_{\text{DNN}}) + \frac{1}{2}(w - w_{\text{DNN}})^\top H_{\text{DNN}}(w - w_{\text{DNN}}), \tag{11}$$

where

$$H_{\text{DNN}} := \nabla^2 \ell_N(w_{\text{DNN}}) = \frac{1}{\tau^2} X^\top X.$$

Under equation 11, the likelihood has the Gaussian form

$$p(y \mid w) \propto \exp\left(-\frac{1}{2}(w - w_{\text{DNN}})^\top H_{\text{DNN}}(w - w_{\text{DNN}})\right). \tag{12}$$

**Proposition E.1: Posterior under SPIN prior**

Under the quadratic likelihood approximation equation 12 and the SPIN prior equation 9, the posterior $p(w \mid y)$ is Gaussian:

$$p(w \mid y) = \mathcal{N}\big(w \mid m(\sigma^2), \Sigma_{\text{post}}(\sigma^2)\big),$$

with posterior precision, covariance, and mean given by

$$A(\sigma^2) := H_{\text{DNN}} + \Lambda_{\text{prior}}, \tag{13}$$

$$\Sigma_{\text{post}}(\sigma^2) = A(\sigma^2)^{-1}, \tag{14}$$

$$m(\sigma^2) = A(\sigma^2)^{-1}\big(H_{\text{DNN}}w_{\text{DNN}} + \Lambda_{\text{prior}}\mu_{\text{prior}}\big). \tag{15}$$

We denote the posterior mean by

$$w_{\text{post}}(\sigma^2) := m(\sigma^2).$$

*Proof.*  By Bayes' rule, the posterior density is proportional to the product of likelihood and prior:

$$p(w \mid y) \; \propto \; p(y \mid w)\, p(w).$$

Under the quadratic likelihood approximation equation 12, we have

$$p(y \mid w) \propto \exp\left(-\frac{1}{2}(w - w_{\text{DNN}})^\top H_{\text{DNN}}(w - w_{\text{DNN}})\right). \tag{16}$$

The SPIN prior equation 9 is Gaussian,

$$p(w) = \mathcal{N}(w \mid \mu_{\text{prior}}, \Sigma_{\text{prior}}),$$

with precision $\Lambda_{\text{prior}} := \Sigma_{\text{prior}}^{-1}$, so its density is

$$p(w) \propto \exp\left(-\frac{1}{2}(w - \mu_{\text{prior}})^\top \Lambda_{\text{prior}}(w - \mu_{\text{prior}})\right). \tag{17}$$

Multiplying equation 16 and equation 17, we obtain the unnormalized posterior:

$$p(w \mid y) \; \propto \; \exp\left(-\frac{1}{2}(w - w_{\text{DNN}})^\top H_{\text{DNN}}(w - w_{\text{DNN}}) - \frac{1}{2}(w - \mu_{\text{prior}})^\top \Lambda_{\text{prior}}(w - \mu_{\text{prior}})\right).$$

We now expand the quadratic forms in the exponent. First,

$$(w - w_{\text{DNN}})^\top H_{\text{DNN}}(w - w_{\text{DNN}}) = w^\top H_{\text{DNN}}w - 2w^\top H_{\text{DNN}}w_{\text{DNN}} + w_{\text{DNN}}^\top H_{\text{DNN}}w_{\text{DNN}}.$$

Similarly,

$$(w - \mu_{\text{prior}})^\top \Lambda_{\text{prior}}(w - \mu_{\text{prior}}) = w^\top \Lambda_{\text{prior}}w - 2w^\top \Lambda_{\text{prior}}\mu_{\text{prior}} + \mu_{\text{prior}}^\top \Lambda_{\text{prior}}\mu_{\text{prior}}.$$

Therefore the total exponent can be written as

$$-\frac{1}{2}(w - w_{\text{DNN}})^\top H_{\text{DNN}}(w - w_{\text{DNN}}) \; - \; \frac{1}{2}(w - \mu_{\text{prior}})^\top \Lambda_{\text{prior}}(w - \mu_{\text{prior}})$$

$$= -\frac{1}{2}\Big[w^\top H_{\text{DNN}}w - 2w^\top H_{\text{DNN}}w_{\text{DNN}} + w_{\text{DNN}}^\top H_{\text{DNN}}w_{\text{DNN}}$$

$$+ w^\top \Lambda_{\text{prior}}w - 2w^\top \Lambda_{\text{prior}}\mu_{\text{prior}} + \mu_{\text{prior}}^\top \Lambda_{\text{prior}}\mu_{\text{prior}}\Big].$$

Collecting the terms involving $w$, we obtain

$$-\frac{1}{2}\Big[w^\top (H_{\text{DNN}} + \Lambda_{\text{prior}})w - 2w^\top (H_{\text{DNN}}w_{\text{DNN}} + \Lambda_{\text{prior}}\mu_{\text{prior}})\Big] + \text{const},$$

where "const" denotes all terms independent of $w$:

$$\text{const} = -\frac{1}{2}\big(w_{\text{DNN}}^\top H w_{\text{DNN}} + \mu_{\text{prior}}^\top \Lambda_{\text{prior}}\mu_{\text{prior}}\big).$$

Define the posterior precision matrix

$$A(\sigma^2) := H_{\text{DNN}} + \Lambda_{\text{prior}}.$$

Then the exponent becomes

$$-\frac{1}{2}\left[w^\top A(\sigma^2)w - 2w^\top\left(H_{\text{DNN}}w_{\text{DNN}} + \Lambda_{\text{prior}}\mu_{\text{prior}}\right)\right] + \text{const.}$$

To identify the mean and covariance of the posterior Gaussian, we complete the square. Let

$$b := H_{\text{DNN}}w_{\text{DNN}} + \Lambda_{\text{prior}}\mu_{\text{prior}}.$$

We seek a vector $m(\sigma^2)$ such that

$$w^\top A(\sigma^2)w - 2w^\top b = (w - m(\sigma^2))^\top A(\sigma^2)(w - m(\sigma^2)) - m(\sigma^2)^\top A(\sigma^2)m(\sigma^2).$$

Expanding the right-hand side gives

$$(w - m)^\top A(w - m) = w^\top Aw - 2w^\top Am + m^\top Am.$$

Matching the linear term in $w$, we require

$$-2A(\sigma^2)m(\sigma^2) = -2b, \qquad \implies \qquad A(\sigma^2)m(\sigma^2) = b.$$

Since $A(\sigma^2)$ is positive definite (sum of two positive semi-definite plus appropriate regularization in SPIN), it is invertible, and we obtain

$$m(\sigma^2) = A(\sigma^2)^{-1}\left(H_{\text{DNN}}w_{\text{DNN}} + \Lambda_{\text{prior}}\mu_{\text{prior}}\right),$$

which is exactly equation 15.

Substituting this back, the exponent in the posterior density can be written as

$$-\frac{1}{2}(w - m(\sigma^2))^\top A(\sigma^2)(w - m(\sigma^2)) + \text{const}',$$

where $\text{const}'$ collects all terms that do not depend on $w$. Thus, up to normalization,

$$p(w \mid y) \propto \exp\left(-\frac{1}{2}(w - m(\sigma^2))^\top A(\sigma^2)(w - m(\sigma^2))\right).$$

This is the density of a multivariate normal distribution with mean $m(\sigma^2)$ and precision $A(\sigma^2)$. Therefore, the posterior covariance is

$$\Sigma_{\text{post}}(\sigma^2) = A(\sigma^2)^{-1},$$

as stated in equation 14, and the posterior is

$$p(w \mid y) = \mathcal{N}\left(w \mid m(\sigma^2), \Sigma_{\text{post}}(\sigma^2)\right).$$

$\square$

---

**Proposition E.2: Expected Error With SPIN Prior**

Assume that the expected test error admits a quadratic approximation around $w^\star$ with Hessian

$$H := \nabla^2 L(w^\star) = \mathrm{diag}(\lambda_1, \ldots, \lambda_p), \qquad \lambda_j > 0, \tag{18}$$

and that, in the same coordinate system, the empirical (training) negative log-likelihood admits a quadratic approximation around $w_{\mathrm{DNN}}$ with Hessian

$$H_{\mathrm{DNN}} := \nabla^2 \ell_N(w_{\mathrm{DNN}}) = \mathrm{diag}(\kappa_1, \ldots, \kappa_p), \qquad \kappa_j > 0. \tag{19}$$

Let the SPIN prior precision be diagonal with the structure

$$\Lambda_{\mathrm{prior}} = \mathrm{diag}(\lambda_{\mathrm{prior},1}, \ldots, \lambda_{\mathrm{prior},p}), \qquad \lambda_{\mathrm{prior},j} = \begin{cases} 1/\sigma^2, & j \in I, \\ 1, & j \in R, \end{cases} \tag{20}$$

and define

$$\delta := w_{\mathrm{DNN}} - w^\star, \qquad \Delta := w_{\mathrm{sparse}} - w_{\mathrm{DNN}},$$

with coordinates $\delta_j$ and $\Delta_j$ in this diagonal basis.

Under the quadratic likelihood approximation with Hessian $H_{\mathrm{DNN}}$ and the SPIN prior, the posterior expected excess test error

$$\mathcal{E}(\sigma^2) := \mathbb{E}_{(x,y)\sim P_{\mathrm{data}}}\big[L(w) - L(w^\star) \mid D\big]$$

decomposes as

$$\mathcal{E}(\sigma^2) = \sum_{j=1}^p E_j(\sigma^2),$$

where each coordinate $j$ contributes

$$E_j(\sigma^2) = \frac{1}{2}\lambda_j\left(\delta_j + \frac{\lambda_{\mathrm{prior},j}}{\kappa_j + \lambda_{\mathrm{prior},j}}\Delta_j\right)^2 + \frac{1}{2}\frac{\lambda_j}{\kappa_j + \lambda_{\mathrm{prior},j}}. \tag{21}$$

---

*Proof.*    By definition of the posterior expected excess test error,

$$\mathcal{E}(\sigma^2) := \mathbb{E}_{(x,y)\sim P_{\mathrm{data}}}\big[L(w) - L(w^\star) \mid D\big].$$

Around $w^\star$, the expected test error admits the quadratic approximation

$$L(w) - L(w^\star) \approx \frac{1}{2}(w - w^\star)^\top H(w - w^\star), \tag{22}$$

with $H$ given by equation 18. In the diagonal basis,

$$H = \mathrm{diag}(\lambda_1, \ldots, \lambda_p), \qquad \lambda_j > 0,$$

so

$$(w - w^\star)^\top H(w - w^\star) = \sum_{j=1}^p \lambda_j(w_j - w_j^\star)^2.$$

The SPIN prior equation 20, together with Proposition E.1, yields

$$w \mid D \sim \mathcal{N}\big(w_{\mathrm{post}}(\sigma^2), \Sigma_{\mathrm{post}}(\sigma^2)\big),$$

with

$$A(\sigma^2) := H_{\mathrm{DNN}} + \Lambda_{\mathrm{prior}} = \mathrm{diag}(\kappa_j + \lambda_{\mathrm{prior},j})_{j=1}^p,$$

$$\Sigma_{\mathrm{post}}(\sigma^2) = A(\sigma^2)^{-1} = \mathrm{diag}\Big(\frac{1}{\kappa_j + \lambda_{\mathrm{prior},j}}\Big)_{j=1}^p,$$

$$w_{\mathrm{post}}(\sigma^2) = A(\sigma^2)^{-1}\big(H_{\mathrm{DNN}}w_{\mathrm{DNN}} + \Lambda_{\mathrm{prior}}\mu_{\mathrm{prior}}\big),$$

where $\mu_{\mathrm{prior}} = w_{\mathrm{sparse}}$.

Substituting equation 22 into the definition of $\mathcal{E}(\sigma^2)$, we obtain

$$\mathcal{E}(\sigma^2) \approx \frac{1}{2}\,\mathbb{E}\big[(w - w^\star)^\top H(w - w^\star) \mid D\big]$$

$$= \frac{1}{2}\,\mathbb{E}\bigg[\sum_{j=1}^{p} \lambda_j (w_j - w_j^\star)^2 \,\bigg|\, D\bigg]$$

$$= \frac{1}{2}\sum_{j=1}^{p} \lambda_j\,\mathbb{E}\big[(w_j - w_j^\star)^2 \mid D\big].$$

For each coordinate $j$, we apply the scalar bias–variance decomposition: for any random variable $Z$ and constant $a$,

$$\mathbb{E}[(Z - a)^2] = (\mathbb{E}[Z] - a)^2 + \mathrm{Var}(Z).$$

Thus,

$$\mathbb{E}\big[(w_j - w_j^\star)^2 \mid D\big] = \big(\mathbb{E}[w_j \mid D] - w_j^\star\big)^2 + \mathrm{Var}(w_j \mid D),$$

and hence

$$\mathcal{E}(\sigma^2) = \frac{1}{2}\sum_{j=1}^{p} \lambda_j\Big(\big(w_{\mathrm{post},j}(\sigma^2) - w_j^\star\big)^2 + \mathrm{Var}(w_j \mid D)\Big), \tag{23}$$

where $w_{\mathrm{post},j}(\sigma^2)$ denotes the $j$-th component of $w_{\mathrm{post}}(\sigma^2)$.

We now compute the posterior mean and variance in this diagonal basis.

**Posterior mean term.** Because $H_{\mathrm{DNN}}$ and $\Lambda_{\mathrm{prior}}$ are diagonal, $A(\sigma^2)$ is diagonal with entries

$$A_{jj}(\sigma^2) = \kappa_j + \lambda_{\mathrm{prior},j},$$

and

$$(H_{\mathrm{DNN}} w_{\mathrm{DNN}} + \Lambda_{\mathrm{prior}} \mu_{\mathrm{prior}})_j = \kappa_j (w_{\mathrm{DNN}})_j + \lambda_{\mathrm{prior},j} (w_{\mathrm{sparse}})_j.$$

Therefore

$$w_{\mathrm{post},j}(\sigma^2) = \frac{\kappa_j (w_{\mathrm{DNN}})_j + \lambda_{\mathrm{prior},j} (w_{\mathrm{sparse}})_j}{\kappa_j + \lambda_{\mathrm{prior},j}}.$$

Introduce

$$\delta_j := (w_{\mathrm{DNN}})_j - w_j^\star, \qquad \Delta_j := (w_{\mathrm{sparse}})_j - (w_{\mathrm{DNN}})_j,$$

so that

$$(w_{\mathrm{DNN}})_j = w_j^\star + \delta_j, \qquad (w_{\mathrm{sparse}})_j = (w_{\mathrm{DNN}})_j + \Delta_j = w_j^\star + \delta_j + \Delta_j.$$

Substituting these into $w_{\mathrm{post},j}(\sigma^2)$,

$$w_{\mathrm{post},j}(\sigma^2) = \frac{\kappa_j(w_j^\star + \delta_j) + \lambda_{\mathrm{prior},j}(w_j^\star + \delta_j + \Delta_j)}{\kappa_j + \lambda_{\mathrm{prior},j}}$$

$$= \frac{(\kappa_j + \lambda_{\mathrm{prior},j})w_j^\star + (\kappa_j + \lambda_{\mathrm{prior},j})\delta_j + \lambda_{\mathrm{prior},j}\Delta_j}{\kappa_j + \lambda_{\mathrm{prior},j}}$$

$$= w_j^\star + \delta_j + \frac{\lambda_{\mathrm{prior},j}}{\kappa_j + \lambda_{\mathrm{prior},j}}\,\Delta_j.$$

Hence

$$w_{\mathrm{post},j}(\sigma^2) - w_j^\star = \delta_j + \frac{\lambda_{\mathrm{prior},j}}{\kappa_j + \lambda_{\mathrm{prior},j}}\,\Delta_j. \tag{24}$$

**Posterior variance term.** Since

$$\Sigma_{\mathrm{post}}(\sigma^2) = A(\sigma^2)^{-1} = \mathrm{diag}\Big(\frac{1}{\kappa_j + \lambda_{\mathrm{prior},j}}\Big)_{j=1}^{p},$$

we have

$$\mathrm{Var}(w_j \mid D) = \big(\Sigma_{\mathrm{post}}(\sigma^2)\big)_{jj} = \frac{1}{\kappa_j + \lambda_{\mathrm{prior},j}}.$$

**Putting everything together.** Substituting equation 24 and the variance expression into equation 23, we obtain

$$\mathcal{E}(\sigma^2) = \frac{1}{2} \sum_{j=1}^{p} \lambda_j \left[ \left( \delta_j + \frac{\lambda_{\text{prior},j}}{\kappa_j + \lambda_{\text{prior},j}} \Delta_j \right)^2 + \frac{1}{\kappa_j + \lambda_{\text{prior},j}} \right].$$

Defining the per-coordinate contribution as

$$E_j(\sigma^2) := \frac{1}{2} \lambda_j \left( \delta_j + \frac{\lambda_{\text{prior},j}}{\kappa_j + \lambda_{\text{prior},j}} \Delta_j \right)^2 + \frac{1}{2} \frac{\lambda_j}{\kappa_j + \lambda_{\text{prior},j}},$$

we obtain

$$\mathcal{E}(\sigma^2) = \sum_{j=1}^{p} E_j(\sigma^2),$$

which is precisely equation 21. □

**Lemma E.1: Strong and weak limits for important**

Let $E_j(\sigma^2)$ be defined as in equation 21 under the diagonal expected test-error Hessian assumption equation 18, the diagonal empirical Hessian assumption equation 19, and the SPIN prior precision equation 20. Then, for each coordinate $j$ if $j \in I$ (important coordinate, SPIN prior variance $\sigma^2$, precision $\lambda_{\text{prior},j} = 1/\sigma^2$), the strong- and weak-prior limits satisfy:

$$E_{j,\text{small}} := \lim_{\sigma^2 \to 0} E_j(\sigma^2) = \frac{1}{2} \lambda_j (\delta_j + \Delta_j)^2, \tag{25}$$

$$E_{j,\text{large}} := \lim_{\sigma^2 \to \infty} E_j(\sigma^2) = \frac{1}{2} \lambda_j \delta_j^2 + \frac{1}{2} \frac{\lambda_j}{\kappa_j}. \tag{26}$$

*Proof.* The per-coordinate contribution to the expected excess test error is, for each $j$,

$$E_j(\sigma^2) = \frac{1}{2} \lambda_j \left( \delta_j + \frac{\lambda_{\text{prior},j}}{\kappa_j + \lambda_{\text{prior},j}} \Delta_j \right)^2 + \frac{1}{2} \frac{\lambda_j}{\kappa_j + \lambda_{\text{prior},j}}, \tag{27}$$

where $\lambda_j$ comes from the Hessian $H$ of the expected test error (cf. equation 18), $\kappa_j$ from the empirical Hessian $H_{\text{DNN}}$ (cf. equation 19), and $\lambda_{\text{prior},j}$ from the SPIN prior precision (cf. equation 20).

We analyze the limits separately for important and redundant coordinates.

**Important coordinates $j \in I$.** For $j \in I$, the SPIN prior precision is

$$\lambda_{\text{prior},j} = \frac{1}{\sigma^2}.$$

Substituting this into equation 27, we obtain

$$E_j(\sigma^2) = \frac{1}{2} \lambda_j \left( \delta_j + \frac{\frac{1}{\sigma^2}}{\kappa_j + \frac{1}{\sigma^2}} \Delta_j \right)^2 + \frac{1}{2} \frac{\lambda_j}{\kappa_j + \frac{1}{\sigma^2}}. \tag{28}$$

We now compute the two limits.

**Strong-prior limit $\sigma^2 \to 0$.** First rewrite the denominators:

$$\kappa_j + \frac{1}{\sigma^2} = \frac{\kappa_j \sigma^2 + 1}{\sigma^2}.$$

Then

$$\frac{\frac{1}{\sigma^2}}{\kappa_j + \frac{1}{\sigma^2}} = \frac{1/\sigma^2}{(\kappa_j \sigma^2 + 1)/\sigma^2} = \frac{1}{\kappa_j \sigma^2 + 1} \xrightarrow[\sigma^2 \to 0]{} 1,$$

and

$$\frac{\lambda_j}{\kappa_j + \frac{1}{\sigma^2}} = \frac{\lambda_j}{(\kappa_j \sigma^2 + 1)/\sigma^2} = \frac{\lambda_j \sigma^2}{\kappa_j \sigma^2 + 1} \xrightarrow[\sigma^2 \to 0]{} 0.$$

Taking the limit $\sigma^2 \to 0$ in equation 28, we obtain

$$E_{j,\text{small}} := \lim_{\sigma^2 \to 0} E_j(\sigma^2) = \frac{1}{2} \lambda_j (\delta_j + \Delta_j)^2,$$

which is exactly equation 25.

**Weak-prior limit $\sigma^2 \to \infty$.** As $\sigma^2 \to \infty$, we have $\lambda_{\text{prior},j} = 1/\sigma^2 \to 0$. Then

$$\frac{\frac{1}{\sigma^2}}{\kappa_j + \frac{1}{\sigma^2}} \leq \frac{1/\sigma^2}{\kappa_j} = \frac{1}{\kappa_j \sigma^2} \xrightarrow[\sigma^2 \to \infty]{} 0,$$

and

$$\frac{\lambda_j}{\kappa_j + \frac{1}{\sigma^2}} \xrightarrow[\sigma^2 \to \infty]{} \frac{\lambda_j}{\kappa_j}.$$

Therefore the limit of equation 28 as $\sigma^2 \to \infty$ is

$$E_{j,\text{large}} := \lim_{\sigma^2 \to \infty} E_j(\sigma^2) = \frac{1}{2} \lambda_j \delta_j^2 + \frac{1}{2} \frac{\lambda_j}{\kappa_j},$$

which is precisely equation 26. $\qquad \square$

**Theorem E.1: Strong SPIN prior under a small validation-loss increase**

Assume the SPIN prior structure equation 20 and the diagonal Hessian assumptions equation 18–equation 19. For each prior variance $\sigma^2$ on important weights, let $\mathcal{E}(\sigma^2)$ denote the posterior expected excess test error defined in Proposition E.1:

$$\mathcal{E}(\sigma^2) := \mathbb{E}_{(x,y)\sim P_{\text{data}}}\big[L(w) - L(w^\star) \mid D\big].$$

Define the strong- and weak-prior limits

$$\mathcal{E}_{\text{small}} := \lim_{\sigma^2 \to 0} \mathcal{E}(\sigma^2), \qquad \mathcal{E}_{\text{large}} := \lim_{\sigma^2 \to \infty} \mathcal{E}(\sigma^2).$$

Suppose that the SPIN pruning step satisfies the validation-loss constraint equation 6:

$$\ell_{\text{val}}(w_{\text{sparse}}) - \ell_{\text{val}}(w_{\text{DNN}}) \leq \varepsilon, \qquad \varepsilon \approx 0, \tag{29}$$

and that $\ell_{\text{val}}(w)$ admits the same quadratic approximation as $L(w)$ in equation 3 with Hessian $H$. Let

$$\delta := w_{\text{DNN}} - w^\star, \qquad \Delta := w_{\text{sparse}} - w_{\text{DNN}},$$

and denote their coordinates in the diagonal basis by $\delta_j$ and $\Delta_j$. Finally, assume that the coordinate-wise quadratic contributions to the validation-loss difference are nonnegative,

$$\lambda_j(\Delta_j^2 + 2\delta_j\Delta_j) \geq 0 \quad \text{for all } j = 1, \ldots, p. \tag{30}$$

If, in addition,

$$\varepsilon < \frac{1}{2}\sum_{j\in I}\frac{\lambda_j}{\kappa_j}, \tag{31}$$

then the strong-prior limit yields strictly smaller expected excess test error than the weak-prior limit:

$$\mathcal{E}_{\text{small}} < \mathcal{E}_{\text{large}}. \tag{32}$$

*Proof.* We recall the definitions

$$\delta := w_{\text{DNN}} - w^\star, \qquad \Delta := w_{\text{sparse}} - w_{\text{DNN}},$$

with coordinates $\delta_j := (w_{\text{DNN}} - w^\star)_j$ and $\Delta_j := (w_{\text{sparse}} - w_{\text{DNN}})_j$. Under the diagonal assumptions and the SPIN prior, Proposition E.1 and the previous lemma give the per-coordinate strong- and weak-prior limits for the expected excess test error.

**Separation into important and redundant coordinates.** For each *important* coordinate $j \in I$, the strong- and weak-prior limits are

$$E_{j,\text{small}} = \frac{1}{2}\lambda_j(\delta_j + \Delta_j)^2, \qquad E_{j,\text{large}} = \frac{1}{2}\lambda_j\delta_j^2 + \frac{1}{2}\frac{\lambda_j}{\kappa_j}.$$

For each *redundant* coordinate $j \in R$, the prior precision $\lambda_{\text{prior},j} = 1$ is independent of $\sigma^2$, so the full expression equation 21 does not depend on $\sigma^2$. In particular,

$$E_{j,\text{small}} = E_{j,\text{large}} \quad \text{for all } j \in R.$$

Therefore the global strong- and weak-prior expected excess test errors

$$\mathcal{E}_{\text{small}} = \sum_{j=1}^{p} E_{j,\text{small}}, \qquad \mathcal{E}_{\text{large}} = \sum_{j=1}^{p} E_{j,\text{large}}$$

differ *only* through the important coordinates:

$$\mathcal{E}_{\text{small}} - \mathcal{E}_{\text{large}} = \sum_{j\in I}(E_{j,\text{small}} - E_{j,\text{large}}),$$

since all redundant-coordinate contributions cancel exactly.

**Difference on important coordinates.** For $j \in I$,

$$E_{j,\text{small}} - E_{j,\text{large}} = \frac{1}{2} \lambda_j (\delta_j + \Delta_j)^2 - \left( \frac{1}{2} \lambda_j \delta_j^2 + \frac{1}{2} \frac{\lambda_j}{\kappa_j} \right)$$

$$= \frac{1}{2} \lambda_j \left[ (\delta_j + \Delta_j)^2 - \delta_j^2 \right] - \frac{1}{2} \frac{\lambda_j}{\kappa_j}$$

$$= \frac{1}{2} \left[ \lambda_j (\Delta_j^2 + 2\delta_j \Delta_j) - \frac{\lambda_j}{\kappa_j} \right].$$

Thus

$$\mathcal{E}_{\text{small}} - \mathcal{E}_{\text{large}} = \frac{1}{2} \sum_{j \in I} \left[ \lambda_j (\Delta_j^2 + 2\delta_j \Delta_j) - \frac{\lambda_j}{\kappa_j} \right]. \tag{33}$$

**Relating the pruning displacement to the validation-loss constraint.** By assumption, the validation loss $\ell_{\text{val}}(w)$ admits the same quadratic approximation as $L(w)$ around $w^\star$ with Hessian $H$:

$$\ell_{\text{val}}(w) \approx \ell_{\text{val}}(w^\star) + \frac{1}{2} (w - w^\star)^\top H (w - w^\star).$$

Using this approximation, the validation-loss difference between the sparse and dense solutions is

$$\ell_{\text{val}}(w_{\text{sparse}}) - \ell_{\text{val}}(w_{\text{DNN}}) \approx \frac{1}{2} \left[ (w_{\text{sparse}} - w^\star)^\top H (w_{\text{sparse}} - w^\star) - (w_{\text{DNN}} - w^\star)^\top H (w_{\text{DNN}} - w^\star) \right]$$

$$= \frac{1}{2} \left[ (\delta + \Delta)^\top H (\delta + \Delta) - \delta^\top H \delta \right]$$

$$= \frac{1}{2} \left[ \Delta^\top H \Delta + 2 \delta^\top H \Delta \right]$$

$$= \frac{1}{2} \sum_{j=1}^{p} \lambda_j (\Delta_j^2 + 2\delta_j \Delta_j).$$

Treating the quadratic approximation as exact along the segment joining $w_{\text{DNN}}$ and $w_{\text{sparse}}$, and combining with the SPIN pruning constraint $\ell_{\text{val}}(w_{\text{sparse}}) - \ell_{\text{val}}(w_{\text{DNN}}) \leq \varepsilon$, we obtain

$$\sum_{j=1}^{p} \lambda_j (\Delta_j^2 + 2\delta_j \Delta_j) \leq 2\varepsilon. \tag{34}$$

By the coordinate-wise nonnegativity assumption equation 30,

$$\lambda_j (\Delta_j^2 + 2\delta_j \Delta_j) \geq 0 \quad \text{for all } j,$$

so every term in the sum equation 34 is nonnegative. Therefore, for any subset of coordinates—in particular, for the important set $I$—we have

$$\sum_{j \in I} \lambda_j (\Delta_j^2 + 2\delta_j \Delta_j) \leq \sum_{j=1}^{p} \lambda_j (\Delta_j^2 + 2\delta_j \Delta_j) \leq 2\varepsilon.^6 \tag{35}$$

---

[6] We briefly justify why the first inequality $\sum_{j \in I} \lambda_j (\Delta_j^2 + 2\delta_j \Delta_j) \leq \sum_{j=1}^{p} \lambda_j (\Delta_j^2 + 2\delta_j \Delta_j)$ is natural under our pruning assumptions. For any coordinate $j$, define

$$a_j := (w_{\text{sparse}})_j - w_j^\star, \qquad b_j := (w_{\text{DNN}})_j - w_j^\star,$$

so that

$$\delta_j = (w_{\text{DNN}})_j - w_j^\star = b_j, \qquad \Delta_j = (w_{\text{sparse}})_j - (w_{\text{DNN}})_j = a_j - b_j.$$

Then we can rewrite the quadratic term as

$$\Delta_j^2 + 2\delta_j \Delta_j = (a_j - b_j)^2 + 2b_j (a_j - b_j)$$

$$= a_j^2 - 2a_j b_j + b_j^2 + 2a_j b_j - 2b_j^2$$

$$= a_j^2 - b_j^2$$

$$= \left( (w_{\text{sparse}})_j - w_j^\star \right)^2 - \left( (w_{\text{DNN}})_j - w_j^\star \right)^2.$$

**Final bound and sufficient condition.** Substituting equation 35 into equation 33, we obtain

$$\mathcal{E}_{\text{small}} - \mathcal{E}_{\text{large}} = \frac{1}{2}\sum_{j\in I}\lambda_j(\Delta_j^2 + 2\delta_j\Delta_j) \; - \; \frac{1}{2}\sum_{j\in I}\frac{\lambda_j}{\kappa_j}$$

$$\leq \varepsilon - \frac{1}{2}\sum_{j\in I}\frac{\lambda_j}{\kappa_j}.$$

Hence a sufficient condition for $\mathcal{E}_{\text{small}} - \mathcal{E}_{\text{large}} < 0$ is exactly

$$\varepsilon < \frac{1}{2}\sum_{j\in I}\frac{\lambda_j}{\kappa_j},$$

which is our assumption Eq. 31 (Assumption Eq. 6 makes assumption Eq. 31 satisfy). Under this condition, $\mathcal{E}_{\text{small}} < \mathcal{E}_{\text{large}}$, and the strong SPIN prior (small prior standard deviation $\sigma$ on important weights) yields strictly smaller expected excess test error than the weak prior (large $\sigma$). $\qquad\square$

**Summary.** *To understand how the $\sigma$ influences performance, we analyzed a simplified linear setting and derived the corresponding expected test error. Under the assumption that pruning identifies an approximately correct subset of important weights, the linear model predicts that smaller values of lead to lower expected error. Although this conclusion is derived in a linearized regime, it is reasonable to conjecture that the same qualitative behavior extends to nonlinear networks. Consistent with this theoretical prediction, our empirical results also show that smaller values yield better performance, as illustrated in Fig. E-9.*

---

Thus

$$\Delta_j^2 + 2\delta_j\Delta_j = \left|(w_{\text{sparse}})_j - w_j^\star\right|^2 - \left|(w_{\text{DNN}})_j - w_j^\star\right|^2,$$

and consequently

$$\lambda_j(\Delta_j^2 + 2\delta_j\Delta_j) = \lambda_j\left(\left|(w_{\text{sparse}})_j - w_j^\star\right|^2 - \left|(w_{\text{DNN}})_j - w_j^\star\right|^2\right).$$

Now consider a non–important coordinate $j \in R$. SPIN performs hard pruning on such coordinates:

$$(w_{\text{sparse}})_j = 0 \quad \text{for all } j \in R.$$

In that case

$$a_j = (w_{\text{sparse}})_j - w_j^\star = -w_j^\star, \qquad b_j = (w_{\text{DNN}})_j - w_j^\star.$$

Moreover, by construction, non–important coordinates are those where the dense solution is already close to zero relative to the population-optimal parameter. A convenient way to formalize this is to assume

$$\left|(w_{\text{DNN}})_j - w_j^\star\right| \; \leq \; \left|0 - w_j^\star\right| = \left|w_j^\star\right| = \left|(w_{\text{sparse}})_j - w_j^\star\right| \quad (j \in R).$$

In terms of $a_j$ and $b_j$, this is

$$|b_j| = \left|(w_{\text{DNN}})_j - w_j^\star\right| \; \leq \; \left|(w_{\text{sparse}})_j - w_j^\star\right| = |a_j| \quad (j \in R),$$

which implies

$$b_j^2 \; \leq \; a_j^2 \quad \Longrightarrow \quad a_j^2 - b_j^2 \; \geq \; 0 \quad (j \in R).$$

Using the identity above,

$$\Delta_j^2 + 2\delta_j\Delta_j = a_j^2 - b_j^2 \; \geq \; 0 \quad (j \in R),$$

and since $\lambda_j > 0$, we obtain

$$\lambda_j(\Delta_j^2 + 2\delta_j\Delta_j) \; \geq \; 0 \quad (j \in R).$$

Hence every non–important coordinate contributes a nonnegative term $\lambda_j(\Delta_j^2 + 2\delta_j\Delta_j)$ to the sum, and adding these terms can only increase the total:

$$\sum_{j\in I}\lambda_j(\Delta_j^2 + 2\delta_j\Delta_j) \; \leq \; \sum_{j\in I}\lambda_j(\Delta_j^2 + 2\delta_j\Delta_j) + \sum_{j\in R}\lambda_j(\Delta_j^2 + 2\delta_j\Delta_j) = \sum_{j=1}^{p}\lambda_j(\Delta_j^2 + 2\delta_j\Delta_j).$$

This is precisely the first inequality in Eq. equation 35.

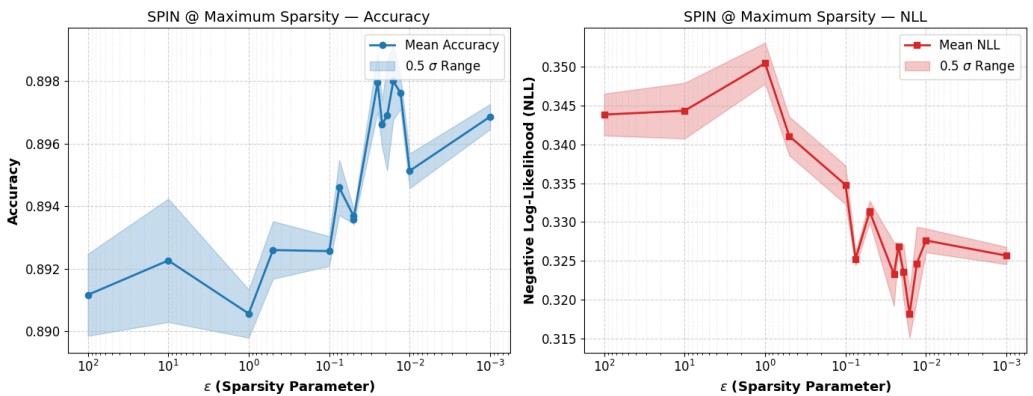

Figure E-9: This figure reports the performance of **ResNet–20** on **CIFAR 10** under different values of $\sigma$. Overall, smaller values of $\sigma$ consistently yield better performance.

## E.2 COMPARISON TO SPIKE-AND-SLAB IN THE LINEAR MODEL

We now compare SPIN to a classical spike-and-slab prior in the same linear-Gaussian setting as Appendix E. We retain the notation $y = Xw^\star + \varepsilon$, $\varepsilon \sim \mathcal{N}(0, \tau^2 I_N)$, and the quadratic approximations

$$L(w) - L(w^\star) \approx \frac{1}{2}(w - w^\star)^\top H(w - w^\star) = \frac{1}{2}\sum_{j=1}^{p} \lambda_j (w_j - w_j^\star)^2,$$

$$\ell_N(w) - \ell_N(w_{\mathrm{DNN}}) \approx \frac{1}{2}(w - w_{\mathrm{DNN}})^\top H_{\mathrm{DNN}}(w - w_{\mathrm{DNN}}),$$

with diagonal Hessians $H = \mathrm{diag}(\lambda_1, \ldots, \lambda_p)$, $\lambda_j > 0$ and $H_{\mathrm{DNN}} = \mathrm{diag}(\kappa_1, \ldots, \kappa_p)$, $\kappa_j > 0$. As in Appendix E, we denote

$$\delta := w_{\mathrm{DNN}} - w^\star, \qquad \Delta := w_{\mathrm{sparse}} - w_{\mathrm{DNN}},$$

with coordinates $\delta_j$ and $\Delta_j$, and the pruning-induced partition

$$I := \{j : \text{important (kept) weight}\}, \qquad R := \{j : \text{pruned weight}\}.$$

### E.2.1 GENERAL GAUSSIAN PRIOR: PER-COORDINATE POSTERIOR RISK

We first derive a per-coordinate formula for the expected excess test error under an *arbitrary* diagonal Gaussian prior. Consider a prior

$$w \sim \mathcal{N}(\mu_{\mathrm{prior}}, \Lambda_{\mathrm{prior}}^{-1}), \qquad \Lambda_{\mathrm{prior}} = \mathrm{diag}(\lambda_{\mathrm{prior},1}, \ldots, \lambda_{\mathrm{prior},p}),$$

combined with the quadratic likelihood approximation around $w_{\mathrm{DNN}}$ with Hessian $H_{\mathrm{DNN}}$. In the diagonal basis, the posterior over each coordinate is

$$w_j \mid D \sim \mathcal{N}(m_j, v_j),$$

with

$$m_j = \frac{\kappa_j (w_{\mathrm{DNN}})_j + \lambda_{\mathrm{prior},j} \mu_{\mathrm{prior},j}}{\kappa_j + \lambda_{\mathrm{prior},j}}, \qquad v_j = \frac{1}{\kappa_j + \lambda_{\mathrm{prior},j}}. \tag{36}$$

The per-coordinate contribution of $w_j$ to the expected excess test error is

$$E_j := \mathbb{E}_{w_j \mid D}\big[L_j(w) - L_j(w^\star)\big] = \frac{1}{2}\lambda_j \Big\{(m_j - w_j^\star)^2 + v_j\Big\},$$

where $L_j(w) := \frac{1}{2}\lambda_j (w_j - w_j^\star)^2$. Using $\delta_j := (w_{\mathrm{DNN}})_j - w_j^\star$ and $m_j - w_j^\star$, we obtain

$$m_j - w_j^\star = \frac{\kappa_j \delta_j + \lambda_{\mathrm{prior},j}(\mu_{\mathrm{prior},j} - w_j^\star)}{\kappa_j + \lambda_{\mathrm{prior},j}}.$$

Substituting into the expression for $E_j$ yields the following generic formula.

---

**Proposition E.3: Per-coordinate risk under a diagonal Gaussian prior**

Under the diagonal approximations for $H$ and $H_{\mathrm{DNN}}$ and a diagonal Gaussian prior $w \sim \mathcal{N}(\mu_{\mathrm{prior}}, \Lambda_{\mathrm{prior}}^{-1})$, the per-coordinate expected excess test error is

$$E_j = \frac{1}{2}\lambda_j \left[ \frac{\big(\kappa_j \delta_j + \lambda_{\mathrm{prior},j}(\mu_{\mathrm{prior},j} - w_j^\star)\big)^2}{(\kappa_j + \lambda_{\mathrm{prior},j})^2} + \frac{1}{\kappa_j + \lambda_{\mathrm{prior},j}} \right]. \tag{37}$$

In particular, the global posterior risk satisfies $\mathcal{E} = \sum_{j=1}^{p} E_j$.

---

### E.2.2 SPIN PRIOR: IMPORTANT VS. REDUNDANT COORDINATES

We now instantiate equation 37 with the SPIN prior. Recall that SPIN uses the sparse solution $w_{\mathrm{sparse}}$ as prior mean, $\mu_{\mathrm{prior}}^{\mathrm{SPIN}} = w_{\mathrm{sparse}}$, and a block-diagonal precision

$$\lambda_{\mathrm{prior},j}^{\mathrm{SPIN}} = \begin{cases} 1/\sigma^2, & j \in I, \\ 1, & j \in R. \end{cases}$$

**Important coordinates $j \in I$.** For $j \in I$,

$$\mu_{\text{prior},j}^{\text{SPIN}} = (w_{\text{sparse}})_j = w_j^\star + \delta_j + \Delta_j,$$

so $\mu_{\text{prior},j}^{\text{SPIN}} - w_j^\star = \delta_j + \Delta_j$, and

$$\lambda_{\text{prior},j}^{\text{SPIN}} = \frac{1}{\sigma^2}.$$

Substituting into equation 37 gives

$$E_j^{\text{SPIN}} = \frac{1}{2}\lambda_j \left[ \frac{\left(\kappa_j \delta_j + \lambda_{\text{prior},j}^{\text{SPIN}}(\delta_j + \Delta_j)\right)^2}{(\kappa_j + \lambda_{\text{prior},j}^{\text{SPIN}})^2} + \frac{1}{\kappa_j + \lambda_{\text{prior},j}^{\text{SPIN}}} \right], \qquad j \in I. \tag{38}$$

In the strong-prior limit $\sigma^2 \to 0$ (equivalently, $\lambda_{\text{prior},j}^{\text{SPIN}} \to \infty$), the posterior concentrates to the sparse point:

$$w_{j,\text{post}}^{\text{SPIN}} \to (w_{\text{sparse}})_j, \qquad v_j \to 0,$$

so

$$E_{j,\text{small}}^{\text{SPIN}} := \lim_{\sigma^2 \to 0} E_j^{\text{SPIN}} = \frac{1}{2}\lambda_j(\delta_j + \Delta_j)^2, \qquad j \in I, \tag{39}$$

which is exactly the strong-prior limit already derived in Lemma E.1.

**Redundant coordinates $j \in R$.** For $j \in R$, SPIN imposes a zero-mean prior:

$$(w_{\text{sparse}})_j = 0, \qquad \mu_{\text{prior},j}^{\text{SPIN}} = 0, \qquad \lambda_{\text{prior},j}^{\text{SPIN}} = 1.$$

Then $\mu_{\text{prior},j}^{\text{SPIN}} - w_j^\star = -w_j^\star$. Under the assumption that redundant coordinates are truly near-zero ($w_j^\star \approx 0$), we approximate $\mu_{\text{prior},j}^{\text{SPIN}} - w_j^\star \approx 0$, so that

$$m_j - w_j^\star \approx \frac{\kappa_j \delta_j}{\kappa_j + 1}, \qquad v_j = \frac{1}{\kappa_j + 1}.$$

Therefore the SPIN contribution on $R$ is approximately

$$E_j^{\text{SPIN}}(j \in R) \approx \frac{1}{2}\lambda_j \left[ \left(\frac{\kappa_j}{\kappa_j + 1}\delta_j\right)^2 + \frac{1}{\kappa_j + 1} \right]. \tag{40}$$

Intuitively, SPIN allows redundant coordinates to weakly track noise in the dense solution (via $\delta_j$) while maintaining non-zero posterior variance.

### E.2.3 SPIKE-AND-SLAB STYLE PRIOR

We now consider a classical spike-and-slab prior of the form

$$w_j \sim (1 - \pi_j)\,\delta_0 + \pi_j\,\mathcal{N}(0, \sigma_{\text{slab}}^2), \tag{41}$$

where $\delta_0$ is a point mass at zero and $\sigma_{\text{slab}}^2$ is the slab variance. To make the comparison analytically tractable, we consider an *idealized* setting in which we take the same partition of coordinates into important and redundant weights as identified by SPIN, and *hypothetically* model them with a spike-and-slab prior—assigning the important weights to the slab component and the redundant weights to the spike component—instead of using SPIN's original prior construction:

- For $j \in I$, the posterior selects the slab component.
- For $j \in R$, the posterior selects the spike component.

**Important coordinates $j \in I$: slab mode.** On $I$, the prior reduces to a Gaussian slab

$$w_j \sim \mathcal{N}(0, \sigma_{\text{slab}}^2), \qquad \lambda_{\text{prior},j}^{\text{slab}} = \frac{1}{\sigma_{\text{slab}}^2},$$

so that $\mu_{\text{prior},j}^{\text{slab}} = 0$, $\mu_{\text{prior},j}^{\text{slab}} - w_j^\star = -w_j^\star$. Applying Proposition E.2.1, we obtain

$$m_j^{\text{slab}} - w_j^\star = \frac{\kappa_j \delta_j - \lambda_{\text{prior},j}^{\text{slab}} w_j^\star}{\kappa_j + \lambda_{\text{prior},j}^{\text{slab}}}.$$

Define the shrinkage factor

$$\gamma_j := \frac{\lambda_{\text{prior},j}^{\text{slab}}}{\kappa_j + \lambda_{\text{prior},j}^{\text{slab}}} \in (0,1), \tag{42}$$

so that

$$m_j^{\text{slab}} - w_j^\star = \delta_j - \gamma_j(w_j^\star + \delta_j) = (1-\gamma_j)\delta_j - \gamma_j w_j^\star. \tag{43}$$

Hence

$$E_j^{\text{slab}} = \frac{1}{2}\lambda_j\left[\left((1-\gamma_j)\delta_j - \gamma_j w_j^\star\right)^2 + \frac{1}{\kappa_j + \lambda_{\text{prior},j}^{\text{slab}}}\right], \qquad j \in I. \tag{44}$$

**Redundant coordinates** $j \in R$**: spike mode.** On $R$, the posterior is dominated by the spike component:

$$w_j \sim \delta_0,$$

so we approximate

$$m_j^{\text{spike}} = 0, \qquad v_j^{\text{spike}} = 0.$$

The per-coordinate risk becomes

$$E_j^{\text{spike}} = \frac{1}{2}\lambda_j(0 - w_j^\star)^2 = \frac{1}{2}\lambda_j(w_j^\star)^2, \qquad j \in R. \tag{45}$$

In the Optimal Brain Damage analysis (LeCun et al., 1989), the curvature-weighted quantity $\lambda_j(w_j^\star)^2$ naturally appears as a measure of the sensitivity of the loss with respect to the parameter $w_j^\star$. When pruning decisions are guided by such curvature-weighted sensitivities, indices assigned to $R$ correspond to directions with small $\lambda_j(w_j^\star)^2$. In this regime, the aggregate spike-and-slab contribution

$$E_{\text{S\&S}}(R) := \sum_{j \in R} E_j^{\text{spike}} = \frac{1}{2}\sum_{j \in R}\lambda_j(w_j^\star)^2 \tag{46}$$

is therefore small compared to the dominant contribution coming from the important coordinates $I$.

For SPIN, using Eq. 40, the contribution on $R$ is approximately

$$E_j^{\text{SPIN}}(j \in R) \approx \frac{1}{2}\lambda_j\left[\left(\frac{\kappa_j}{\kappa_j + 1}\delta_j\right)^2 + \frac{1}{\kappa_j + 1}\right], \tag{47}$$

so that

$$E_{\text{SPIN}}(R) := \sum_{j \in R} E_j^{\text{SPIN}} \approx \frac{1}{2}\sum_{j \in R}\lambda_j\left[\left(\frac{\kappa_j}{\kappa_j + 1}\delta_j\right)^2 + \frac{1}{\kappa_j + 1}\right]. \tag{48}$$

Here the indices in $R$ again correspond to flat, low-sensitivity directions of the loss landscape (small curvature $\lambda_j$ and near-zero signal $w_j^\star$), so both $E_{\text{S\&S}}(R)$ and $E_{\text{SPIN}}(R)$ are curvature-suppressed and remain small compared to the leading contribution on $I$. Thus, the difference between SPIN and spike-and-slab on redundant coordinates is of second order in the overall posterior risk.

### E.2.4 COORDINATE-WISE AND GLOBAL COMPARISON: SPIN VS. SPIKE-AND-SLAB

We now compare SPIN (in the strong-prior regime on $I$) to spike-and-slab in this idealized setting.

**Important coordinates** $j \in I$. For important coordinates, we compare the SPIN risk with a strong prior in Eq. equation 39 to the slab. Neglecting the slab variance term (which is always nonnegative), a sufficient coordinate-wise condition for SPIN to have smaller risk than spike-and-slab is

$$\frac{1}{2}\sum_{j \in I}\lambda_j(\delta_j + \Delta_j)^2 < \frac{1}{2}\sum_{j \in I}\lambda_j\left[\left((1-\gamma_j)\delta_j - \gamma_j w_j^\star\right)^2 + \frac{1}{\kappa_j + \lambda_{\text{prior},j}^{\text{slab}}}\right]. \tag{49}$$

The left-hand side can be expanded as

$$\frac{1}{2}\sum_{j \in I}\lambda_j(\delta_j + \Delta_j)^2 = \frac{1}{2}\sum_{j \in I}\lambda_j\left(\delta_j^2 + 2\delta_j\Delta_j + \Delta_j^2\right). \tag{50}$$

By Eq. 35 (the assumption that pruning preserves the validation error), the additional terms involving $\Delta_j$ can be controlled, leading to the upper bound

$$\frac{1}{2}\sum_{j\in I}\lambda_j(\delta_j+\Delta_j)^2 \;\lesssim\; \underbrace{\frac{1}{2}\sum_{j\in I}\lambda_j\delta_j^2}_{\text{DNN Error}}. \tag{51}$$

Consequently, a convenient sufficient condition ensuring that SPIN has smaller risk than spike-and-slab on the important coordinates is

$$\frac{1}{2}\sum_{j\in I}\lambda_j(\delta_j+\Delta_j)^2 \;\lesssim\; \underbrace{\frac{1}{2}\sum_{j\in I}\lambda_j\delta_j^2}_{\text{DNN Error}} \;<\; \frac{1}{2}\sum_{j\in I}\underbrace{\lambda_j}_{>0}\left[\underbrace{\left((1-\gamma_j)\delta_j-\gamma_j w_j^\star\right)^2}_{>0}+\underbrace{\frac{1}{\kappa_j+\lambda_{\text{prior},j}^{\text{slab}}}}_{>0}\right]. \tag{52}$$

In other words, as long as the underlying (i) DNN is reasonably well trained and (ii) pruning preserves the accuracy, the above inequality can be satisfied, and SPIN achieves a strictly smaller risk than spike-and-slab on $I$ (noting that $\lambda_j > 0$).

**Summary.**   *To summarize, we have analyzed, in a linear-Gaussian model, whether it is more reasonable to model the posterior using a SPIN-style prior constructed from a pruned DNN, rather than a spike-and-slab prior. On the important coordinates $I$, the SPIN prior concentrates around the pruned solution and, under the assumptions that (i) the dense DNN is reasonably well trained and (ii) pruning preserves the validation performance, the strong-prior SPIN risk can be strictly smaller than the risk induced by the spike-and-slab slab component (Eq. 52). On the redundant coordinates $R$, spike-and-slab fixes weights exactly to zero, yielding an excess risk equal to the curvature-weighted quantity $\frac{1}{2}\sum_{j\in R}\lambda_j(w_j^\star)^2$, while SPIN leaves a small amount of posterior variability whose contribution is nevertheless strongly suppressed by the small curvature eigenvalues $\{\lambda_j : j \in R\}$. In this sense, and in line with the interpretation suggested by the Optimal Brain Damage analysis (LeCun et al., 1989), the practically relevant distinction between SPIN and spike-and-slab arises on the important coordinates $I$, where the SPIN-style prior is better aligned with the pruned DNN solution than a zero-centered slab.*

## F   Layer-wise Structural Patterns in Masks from Pruning

In this section, we examine whether the pruning masks produced by global magnitude pruning exhibit meaningful layer-wise structural regularities, and how these structures relate to the posterior uncertainty patterns observed in Bayesian deep learning.

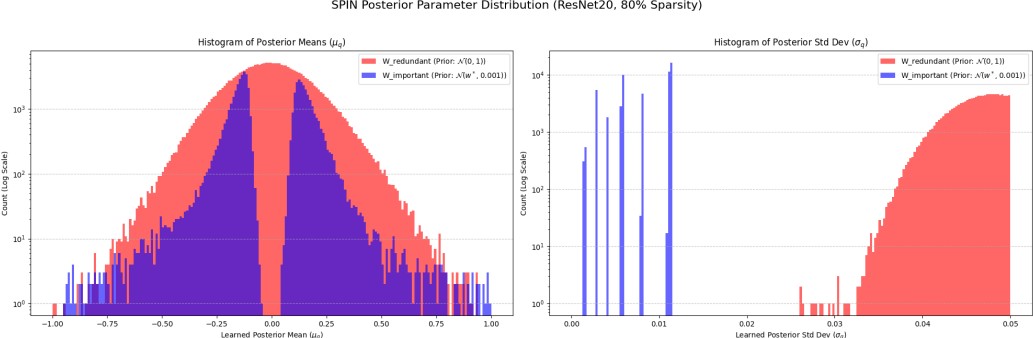

Figure F-10: Posterior distribution of all BNN weights obtained under the SPIN prior of **ResNet-20** trained on **CIFAR-10**. The redundant set $\mathcal{W}_{\text{redundant}}$ contains 214,669 weights and the important set $\mathcal{W}_{\text{important}}$ contains 53,667 weights, corresponding to an overall sparsity level of $80\%$ (i.e., $80\%$ of the weights are pruned). The left panel shows the posterior means for $\mathcal{W}_{\text{redundant}}$ and $\mathcal{W}_{\text{important}}$, while the right panel depicts their posterior standard deviations. Consistent with the construction of the SPIN prior, the weights in $\mathcal{W}_{\text{important}}$ exhibit uniformly smaller posterior standard deviations than those in $\mathcal{W}_{\text{redundant}}$, confirming that coordinates deemed important by pruning are also assigned systematically lower posterior uncertainty.

**Layer-wise Structural Patterns in Pruning Masks.**   Across all experiments, the sparsity patterns induced by global magnitude pruning display a highly consistent and interpretable layer-wise structure. Specifically, the pruning masks naturally form:

$$\underbrace{\text{Dense} \rightarrow \text{Sparse} \rightarrow \text{Dense}}_{\substack{\text{First} \qquad \text{Middle} \qquad \text{Last} \\ \text{Layer} \qquad \text{Layer} \qquad \text{Layer}}} \qquad \text{(C1)}$$

Here, **Dense** indicates that only a small fraction of weights are pruned, whereas **Sparse** indicates that a large portion of weights are pruned. In other words, the early layers undergo minimal pruning, the amount of pruning increases progressively in deeper layers, and the final layers again exhibit relatively little pruning (as visualized in Fig. F-11):

- **Input and output layers remain dense.** The first convolutional layer and the final linear classifier retain most of their weights, highlighting their importance for early feature extraction and final decision mapping. In Fig. F-11, the first convolutional layer has only $28.2\%$ sparsity, and the final layer only $14.5\%$ sparsity—remarkably low proportions given that the overall network is pruned at $80\%$ (Fig. F-10).

- **Middle layers become progressively sparse.** Intermediate residual blocks exhibit significantly higher sparsity levels, frequently exceeding $70$–$80\%$, indicating that these layers contain a much larger set of redundant directions relative to the first and last layers.

This *Dense-Sparse-Dense* (C1) structural hierarchy arises organically from magnitude-based pruning without any explicit architectural constraints, suggesting that pruning implicitly captures the intrinsic layer-wise importance structure of the model.

**Connection to Posterior Uncertainty Patterns.**   Recent empirical studies analyzing posterior samples of neural networks—whether obtained by (i) empirically estimating the neural network

distribution (Fortuin et al., 2021)[7] or (ii) MCMC-based posterior sampling (Sommer et al., 2024)[8]—consistently report a characteristic layer-wise uncertainty structure.

$$
\underset{\text{First Layer}}{\text{Low Uncertainty}} \rightarrow \underset{\text{Middle Layer}}{\text{High Uncertainty}} \rightarrow \underset{\text{Last Layer}}{\text{Low Uncertainty}} \tag{C2}
$$

In these analyses, the first and last layers show highly concentrated posterior distributions, while the middle layers exhibit broad posterior variability.

Remarkably, the structural characteristics observed in the pruning masks (C1) allow SPIN to construct a prior for the BNN that exhibits the corresponding structural form (C2). SPIN transforms this mask structure characteristic (C1) directly into a hierarchical uncertainty characteristic (C2) to the BNN prior:

- Unpruned, important weights are assigned a *small* prior variance ($\sigma = 0.001$), producing low posterior uncertainty.
- Pruned, redundant weights are assigned a *large* prior variance ($\sigma = 1$), allowing high posterior uncertainty.

As a result, SPIN produces a prior uncertainty profile that *closely mirrors* the empirical posterior behavior (i.e., (C2)): The pruning-derived mask, therefore, encodes a meaningful structural hierarchy within the network, and the SPIN prior leverages this hierarchy to shape a posterior.

---

[7]In Fig. 2 of Fortuin et al. (2021), the layer-wise covariance matrices are visualized. The diagonal entries (highlighted with an "X" mark) represent the per-parameter variances, and the figure clearly shows that these variances—and thus the posterior uncertainty—tend to increase as the network depth increases.

[8]Fig. 2 of Sommer et al. (2024) explicitly analyzes this phenomenon and demonstrates the emergence of a characteristic layer-wise uncertainty structure across the network.

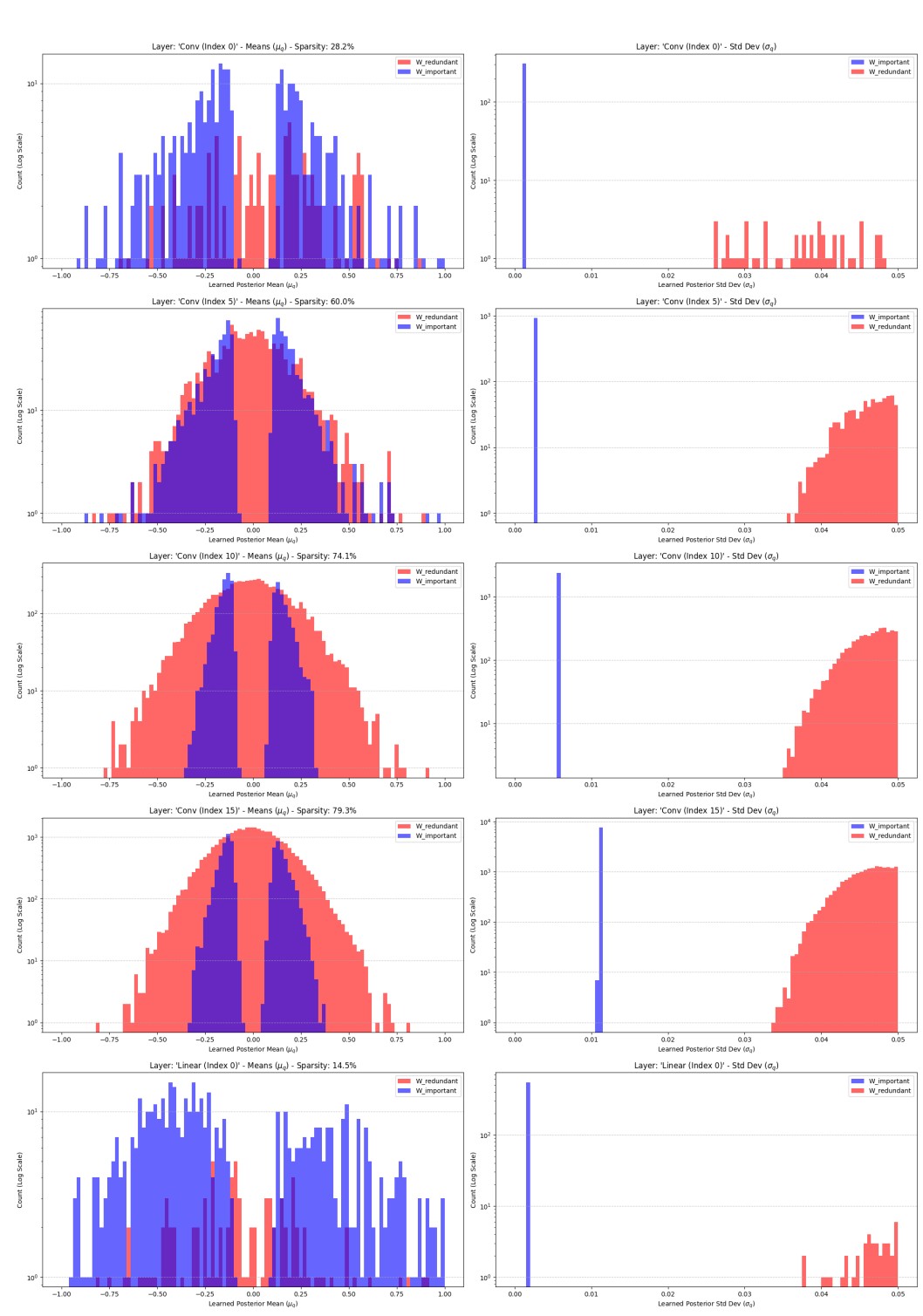

Figure F-11: We visualize the network (**ResNet-20** trained on **CIFAR-10** via SPIN) variational parameters (i.e., $\mu$ and $\sigma$) layer by layer. The first row corresponds to the first convolutional layer, the second row to the 5th convolutional layer, the third row to the 10th convolutional layer, the fourth row to the 15th convolutional layer, and the final row to the last linear layer of the network. The sparsity starts at 28.2% in the early layers, increases to 60–80% in the middle layers, and decreases again to 12% in the final layer. Here, sparsity refers to the proportion of zeros in the mask (i.e., the proportion of pruned weights), which directly indicates the fraction of redundant weights in each layer.

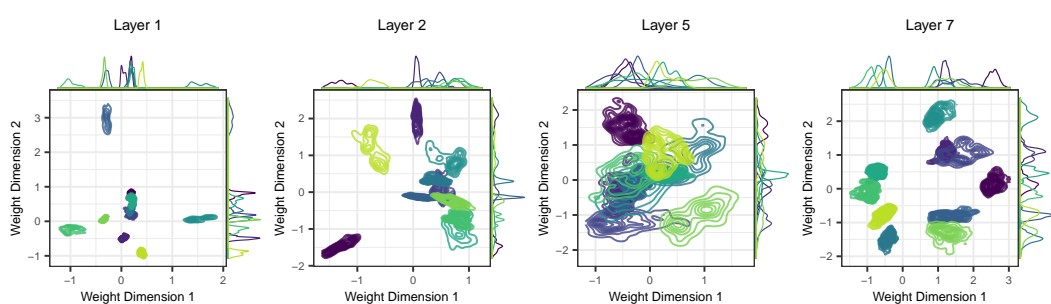

Figure F-12: This figure visualizes the distribution of BNN weights obtained via MCMC-based posterior sampling following (Fig 2. in Sommer et al. (2024)). The first layer exhibits very limited variability, indicating low posterior uncertainty, whereas the intermediate layers (e.g., Layers 2 and 5) display substantially higher dispersion and thus much larger uncertainty. In the final layers, the posterior variability decreases again, yielding an overall *low-uncertainty* → *high-uncertainty* → *low-uncertainty* pattern across depth. This aligns closely with the hierarchical sparsity pattern observed in the pruning-derived mask, which transitions from *dense* → *sparse* → *dense* across layers. In particular, SPIN encodes this dense–sparse–dense structure into the prior as a corresponding low–high–low uncertainty profile, which is then faithfully reflected in the MCMC posterior samples.

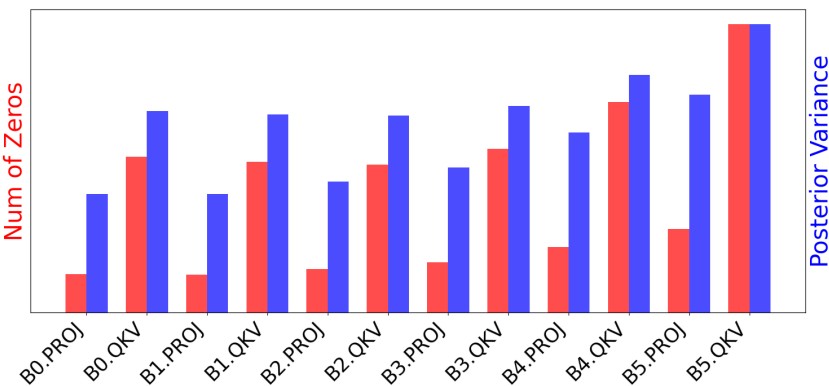

Figure F-13: Within the attention module, the MLP responsible for the output projection consistently exhibits lower sparsity than the MLPs that generate the queries, keys, and values (QKV). SPIN leverages this pattern by assigning lower prior uncertainty to the projection MLP and relatively higher prior uncertainty to the QKV MLPs. As a result, the posterior distribution inherits this structure: the posterior variance of the QKV MLP weights is consistently higher than that of the projection MLP, reflecting the sparsity-induced uncertainty hierarchy encoded by SPIN.

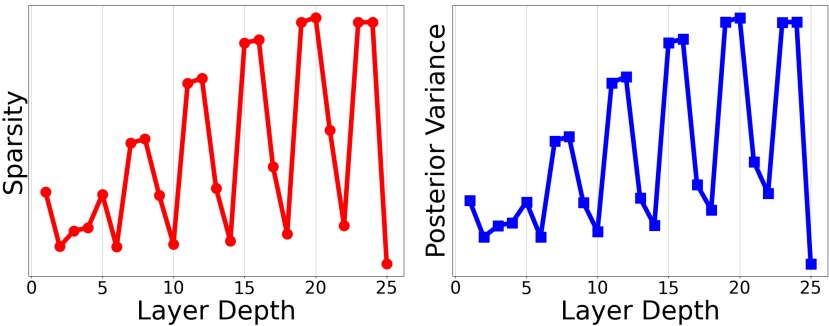

Figure F-14: The left panel shows the layer-wise sparsity of the pruning mask obtained from the **ViT** model, while the right panel visualizes the corresponding weight posterior induced by combining this mask with SPIN. Globally, the network exhibits a *dense → sparse → dense* pattern: the early layers are relatively dense, sparsity increases toward the middle layers, and then decreases again in the final layers. In addition, within each attention block, the QKV MLPs consistently have higher sparsity than the projection MLP, which gives rise to the characteristic zig-zag structure across layers. SPIN incorporates this structure directly into the prior, leading to a posterior in which layer-wise hierarchical importance—both across depth and between QKV vs. projection MLPs—is explicitly reflected in the resulting uncertainty profile.

**Layer Wise Structural Patterns in ViT** We also analyze **ViT** layer-wise pattern on **CIFAR-10**. It is *Dense → Sparse → Dense* across depth, similar to **ResNet-20**. However, within each transformer block, we observe that the attention module MLPs (*query-key–value (QKV)*) are consistently *more pruned* than the MLPs used for projection. In other words, the pruning process implicitly identifies the *QKV*-related submodules as less important than the projection submodules. When combined with SPIN, these masks do not merely encode a coarse hierarchical importance across layers, but also capture *module-level importance* within each layer (e.g., attention-related MLPs vs. projection MLPs). SPIN then maps this structured importance into the prior, assigning lower uncertainty to more important submodules and higher uncertainty to less important ones. Taken together, these observations suggest that the pruning-derived masks are not simply indicators of "zero vs. non-zero" weights; rather, they encode a rich, hierarchical notion of importance across layers and modules. SPIN leverages this structure to construct informative priors that constrain important weights more strongly while allowing unimportant weights to remain flexible.

## F.1 Additional Comparison with Other Sub-Network Approach

The experimental setup is as follows: We first train a **ResNet-20** on **CIFAR-10** for 1000 epochs using SGD with a learning rate of 0.001, weight decay of 0.0001, momentum of 0.99, and without Nesterov acceleration. We then perform pruning for an additional 1000 epochs, again using SGD with a learning rate of 0.0001, weight decay of 0.0001, momentum of 0.99, and no Nesterov, to obtain a family of sparsified sub-networks. Finally, we train the corresponding BNN models (both Pruned BNN and SPIN)for 90 epochs with 30 Monte Carlo samples at test time, using SGD with a learning rate of 0.001, zero weight decay, momentum of 0.9, and without Nesterov acceleration. For each sparsity level, we train both the pruned BNN baseline and SPIN with three different random seeds, and report the mean and standard deviation of the performance in Table F-8. For sparsity levels above 30%, SPIN consistently outperforms the pruned BNN across metrics, and our proposed SPIN$_{@\mathrm{maximum}}$ achieves better performance than any of the pruned BNN baselines. In addition, we compare SPIN against sub-network obtained from the same DNN using the linearized Laplace sub-network (Daxberger et al., 2021), and observe that SPIN yields substantially superior performance, as shown in Table F-9.

Table F-8: Pruned BNN vs. SPIN

| Sparsity | Pruned BNN (Acc) | SPIN (Acc) |
|---|---|---|
| 0% | **0.8908 $\pm$ 0.00065** | 0.87917 $\pm$ 0.00017 |
| 10% | **0.8890 $\pm$ 0.00229** | 0.88153 $\pm$ 0.00097 |
| 20% | **0.89067 $\pm$ 0.00354** | 0.88670 $\pm$ 0.00106 |
| 30% | 0.88967 $\pm$ 0.00314 | **0.89283 $\pm$ 0.00066** |
| 40% | 0.88857 $\pm$ 0.00368 | **0.89440 $\pm$ 0.00051** |
| 50% | 0.88527 $\pm$ 0.00152 | **0.89883 $\pm$ 0.00061** |
| 60% | 0.88153 $\pm$ 0.00092 | **0.89873 $\pm$ 0.00142** |
| 70% | 0.87577 $\pm$ 0.00129 | **0.89937 $\pm$ 0.00221** |
| 80% | 0.86740 $\pm$ 0.00318 | **0.89940 $\pm$ 0.00184** |
| 90% | 0.84297 $\pm$ 0.00284 | **0.89703 $\pm$ 0.00083** |

Table F-9: Comparison with Linearized Laplace Subnetwork (Daxberger et al., 2021)

| Method | Acc | NLL |
|---|---|---|
| Linearized Laplace | 0.8620 | 0.4160 |
| SPIN$_{@\mathrm{maximum}}$ | **0.8970** | **0.3293** |

---

**Algorithm 2 SP**arsity-**I**nformed Priors for Bayesian **N**eural Networks (SPIN): Detailed procedure

---

1: **Input:** Neural network architecture $f_\theta$; training dataset $\mathcal{D}$
2: **Output:** *Hybrid prior $p(\mathbf{w})$ for a BNN.*
3: Train a deterministic DNN $f_{\theta_0}$ on the training data $\mathcal{D}$.
4: Let $L_0$ denote its negative log-likelihood (NLL) on a validation set.
5: Set $\theta_{\text{pruned}} \leftarrow \theta_0$.
     `/* Pruning begins to obtain weights of Sparse DNN */`
6: **repeat**
7:     $\theta_{\text{last\_good}} \leftarrow \theta_{\text{pruned}}$                    ▷ Save the state before this pruning iteration
8:     Prune a fraction of the remaining nonzero weights in $\theta_{\text{pruned}}$.        ▷ `sparsity` increases
9:     Fine-tune $\theta_{\text{pruned}}$ and compute the new NLL, $L_{\text{new}}$.
10: **until** $L_{\text{new}} > L_0$                    ▷ Stop when the NLL does not recover to the original level
     `/* `$\theta_{\text{last\_good}}$` holds the weights at the maximum sparsity */`
11: $\mathcal{W}_{\text{important}} \leftarrow \{\, w_j^* \in \theta_{\text{last\_good}} \mid w_j^* \neq 0 \,\}$.
12: $\mathcal{W}_{\text{redundant}} \leftarrow \theta_0 \setminus \mathcal{W}_{\text{important}}$.
     `/* Set a hybrid prior based on the pruning results */`
13: **for** each weight $w_j$ in $\theta_0$ **do**
14:     **if** the corresponding $w_j^* \in \mathcal{W}_{\text{important}}$ **then**
15:         $p(w_j) \leftarrow \mathcal{N}(w_j \mid w_j^*, \sigma^2)$    ▷ Informative prior with low variance
16:     **else**
17:         $p(w_j) \leftarrow \mathcal{N}(w_j \mid 0, 1)$    ▷ Standard normal prior (uninformative)
18:     **end if**
19: **end for**
20: **return** $p(\mathbf{w})$

---

## F.2  ViT and CNN Hybrid Approach

In the **TinyImageNet–ViT** setting, the baseline DNN itself turns out to be a major bottleneck: as shown in Table B-3, the **ViT** struggles to train reliably, which prevents SPIN from fully realizing its potential. To obtain a stronger initial DNN and thereby maximize the benefit of SPIN, we slightly modify the architecture by prepending a shallow **ResNet**-style convolutional stem, yielding a **CNN+ViT hybrid**. Using this improved backbone, we first learn a well-performing sparse subnetwork via pruning and then apply SPIN with the resulting mask. As summarized in Table F-10, this adjustment leads SPIN to achieve substantially higher accuracy and lower NLL than all competing priors (e.g., 63.33% ACC and 1.5260 NLL).

Table F-10: Performance comparison of different priors on **TinyImageNet** (**CNN+ViT**).

| Model | Acc(↑) | NLL(↓) | ECE(↓) | AUROC (ENT)(↑) |
|---|---|---|---|---|
| Iso. Gaussian | 0.5745 | 1.7675 | **0.0121** | 0.8464 |
| Laplace | 0.5707 | 1.7737 | 0.0187 | 0.8457 |
| Student-t | 0.5605 | 1.8231 | 0.0151 | 0.7868 |
| Spike-and-Slab | 0.5648 | 1.7863 | 0.0176 | 0.7576 |
| MOPED | 0.5695 | 1.8196 | 0.0222 | 0.8475 |
| Horseshoe | 0.5925 | 1.7038 | 0.0290 | 0.8387 |
| SPIN | **0.6333** | **1.5260** | 0.0392 | **0.8693** |