# OpenReview forum: "EMPIRICAL PRIORS FOR BAYESIAN NEURAL NETWORKS VIA WEIGHT PRUNING"
_ICLR.cc/2026/Conference — Submitted to ICLR 2026_

### Official Review · Reviewer_Vek4 · 2025-10-17

**Soundness:** 3
**Presentation:** 3
**Contribution:** 2
**Rating:** 4
**Confidence:** 4

**Summary:**

This paper presents an intuitive method for constructing priors for Bayesian Neural Networks (BNNs). The core contribution is a two-stage process: first, a deterministic neural network is trained and pruned to identify its most salient weights. Second, this structural information is used to define a prior for a BNN, where non-salient (pruned) weights are given a standard normal prior while the independent still Gaussian priors for the non-pruned weights are centered at the respective weight with very small variance. The authors use mean-field Variational Inference to approximate the posterior and provide an empirical validation of their approach.

**Strengths:**

* **[S1] Conceptual Simplicity:** The core idea of using pruning to inform BNN priors is intuitive and conceptually straightforward. It connects two distinct areas of research in a simple and interesting way.

* **[S2] Clarity of Presentation:** The manuscript is very well-written and clearly structured. The authors do an excellent job of explaining their motivation and methodology, making the paper easy to follow and understand.

* **[S3] Well-Executed Experiments:** Within their defined scope, the experiments are carried out properly and described with sufficient detail to allow for reproducibility.

**Weaknesses:**

* **[W1] Significant Hyperparameter Sensitivity:** The proposed pipeline introduces a non-trivial number of sensitive hyperparameters. The choice of the prior variance for non-pruned weights, $\sigma^2$, is critical, and the paper provides little guidance on how to set it. Moreover, as the experiments in Appendix C reveal, the optimal sparsity level is highly task-dependent. This combination significantly increases the practical overhead and tuning effort, somewhat undermining the method's apparent simplicity.

* **[W2] Lack of Transparency in Empirical Evaluation:** The main paper's evaluation of the sensitivity to $\sigma^2$ (Figure 2) is concerning. The authors show results for $\sigma \in \{0.001, 1, 10, 100\}$ but omit values like $0.1$ and $0.01$ (in the main paper). According to Appendix B, the performance for $\sigma=0.1$ is notably poor. Burying negative results in the appendix instead of discussing them transparently in the main text casts doubt on the robustness of the findings and the subjective choice of $\sigma=0.001$ for the main experiments. Additionally, the caption for Figure 2 incorrectly assigns colors to the different methods (this is obviously very minor - just wanted to note it).

* **[W3] Limited Scope of Bayesian Inference and Missing Baselines:** The paper relies exclusively on mean-field VI, one of the simplest but also most limited approximate inference techniques. It's unclear if the empirical gains are a feature of the proposed prior or an artifact of how VI interacts with it. A more compelling evaluation would involve more expressive inference methods, such as MCMC. Critically, the analysis in Section 6.2 is missing a key baseline: a fully Bayesian treatment of the *sparse subnetwork* itself (i.e., performing inference only on the unpruned weights). This comparison is essential for understanding whether the proposed method offers any advantage over simply inferring a posterior over the pruned architecture directly, which would also be cheaper not only at training but also at inference time.

* **[W4, minor] Unconvincing Qualitative Analysis:** The claims made in Section 6.1 regarding the diversity of the learned representations (Figures 3a-c) are not that convincing. The provided UMAP-based visualizations show little structural differences between the methods (potentially even due to randomness in the graph layout algorithm), and the argument that the proposed prior "frees more of the weight space" seems handwavy to me and is not convincingly supported by the evidence.

* **[W5] Superficial Engagement with Related Concepts:** The paper overlooks several important connections. For instance, why not simply perform inference on the pruned subnetwork? This connects to a rich literature on Bayesian inference in subspaces (see e.g. [1]). The discussion could also benefit from engaging with literature on overparameterization, which suggests that larger models can have smoother, easier-to-explore loss landscapes (see e.g. [2]). This stands in contrast to the implicit assumption that pruning simplifies the inference task and could actually support the use of SPIN instead of subnetwork inference. Finally, recent work [3] has shown that the exploration patterns of advanced MCMC samplers can structurally resemble classical pruning masks out of the box, an interesting connection within the field of BDL that could be analyzed in conjunction with the pruning masks obtained in the present work.

## References

[1] Daxberger, E., Nalisnick, E., Allingham, J. U., Antorán, J., & Hernández-Lobato, J. M. (2021, July). Bayesian deep learning via subnetwork inference. In International Conference on Machine Learning (pp. 2510-2521). PMLR.

[2] Wilson, A. G. (ICML 2025). Deep learning is not so mysterious or different.

[3] Sommer, E., Wimmer, L., Papamarkou, T., Bothmann, L., Bischl, B., & Rügamer, D. (ICML 2024). Connecting the dots: is mode-connectedness the key to feasible sample-based inference in Bayesian neural networks?.

**Questions:**

[Q1] Regarding the missing baseline discussed in **[W3]**, could you provide a comparison against a sparse/subnetwork BNN? Specifically, a BNN where inference (e.g., via VI and, ideally, an MCMC/diffiusion sampler) is performed *only* on the subnetwork identified by the initial pruning step. This seems like the most direct and relevant point of comparison and is crucial for understanding the proposed methodology.

[Q2] Can you justify the choice of $\sigma=0.001$ as the primary setting in your experiments? Given the poor results for $\sigma=0.1$ reported in the appendix (**[W2]**), the current presentation feels like cherry-picking. A more transparent discussion of the method's sensitivity to this crucial hyperparameter should be included in the main text. Have you analyzed whether this effectively turns your method in subnetwork inference? Is there any meaningful variance left in the marginal posterior of the salient parameters?

[Q3] Why was VI chosen as the sole inference method? I guess scalability. Have you considered whether the benefits you observe are tied specifically to the limitations of the mean-field approximation?

[Q4] In your analysis, did you observe any layer-wise structural patterns in the masks produced by your pruning approach? As noted in **[W5]**, recent work has found connections between exploration patterns of samplers and classical pruning patterns. A brief analysis of this could better situate your work within the current BNN literature.

I enjoyed reading the manuscript, which presents an interesting, albeit simple, idea. The empirical work is solid, but the evaluation could be made much more robust by including the requested baselines and a more transparent discussion of the method's sensitivities. If the authors can satisfactorily address the weaknesses and questions raised, I would be inclined to raise my score.

---

> ### Author Response · Authors · 2025-11-20
> **Comparison against a sparse/subnetwork BNN**
>
> We thank the reviewer Vek4 for the positive evaluation and for raising important questions regarding comparisons to Subnetwork Inference.
>
> ---
>
> ## **1. Comparison with Subnetwork Inference (Addressing Q1 and W3)**
>
> The reviewer correctly noted that comparing SPIN against a BNN restricted to the sparse subnetwork is essential to determine whether our improvements arise from the prior design or simply from pruning.
>
> To address this, we implemented the **Pruned BNN** baseline exactly as requested in Q1.
> In this baseline:
>
> - All pruned (non-salient) weights are *hard-fixed to zero*
> - These weights are excluded from variational inference
>
> As shown in **Table 1**, SPIN consistently outperforms the Pruned BNN baseline, especially in high-sparsity regimes (≥ 30%).
>
> **Interpretation.**
> Although pruned weights are “less important,” *hard-zeroing* them eliminates the model’s ability to make subtle posterior adjustments. SPIN instead assigns these weights a **zero-mean, high-variance Gaussian prior**, enabling:
>
> - a *soft removal* (encouraging sparsity),
> - while still allowing Bayesian flexibility to reactivate dimensions when supported by data.
>
> This soft inductive effect preserves expressive capacity that Pruned BNNs inherently lose.
>
> ---
>
> ## **2. Comparison with Bayesian Subnetwork Inference (Addressing W5)**
>
> To address [W5], we compared SPIN against the Linearized Laplace method by Daxberger et al. (2021) applied to the same pruned subnetworks. As shown in Table 2, SPIN consistently outperforms this baseline in both Accuracy and NLL across all sparsity levels (where 0% represents the original full-network model).
>
> **Theoretical Perspective: Soft Inductive Bias vs. Restriction Bias**
> While both methods leverage structural information from pruning, they differ fundamentally in their implementation of inductive bias:
>
> **Daxberger et al. (2021) — Restriction Bias.**
> Non-selected weights are replaced with Dirac delta distributions (fixed point estimates). This imposes a hard restriction, strictly confining posterior inference to a low-dimensional subspace.
>
> **SPIN — Soft Inductive Bias.**
> All weights remain stochastic. Pruned weights are assigned zero-mean, high-variance Gaussian priors, creating a preference for sparsity without eliminating the possibility of using those weights if necessary.
>
> This distinction aligns with Wilson et al. (2025), who argue that soft preferences lead to better generalization than hard constraints by maintaining a flexible hypothesis space. Unlike Daxberger et al., which collapses part of the weight space, SPIN retains the expressiveness required for robust posterior approximation.
>
> ---
>
> ## **Empirical Confirmation**
>
> ---
>
> ## **Table 1. Pruned BNN vs. SPIN (Accuracy ± Std)**
>
> | Sparsity | Pruned BNN (Acc ± Std) | SPIN (Acc ± Std) |
> |---------|--------------------------|-------------------|
> | 0%      | 0.8908 ± 0.00065         | 0.87917 ± 0.00017 |
> | 10%     | 0.8890 ± 0.00229         | 0.88153 ± 0.00097 |
> | 20%     | 0.89067 ± 0.00354        | 0.88670 ± 0.00106 |
> | 30%     | 0.88967 ± 0.00314        | 0.89283 ± 0.00066 |
> | 40%     | 0.88857 ± 0.00368        | 0.89440 ± 0.00051 |
> | 50%     | 0.88527 ± 0.00152        | 0.89883 ± 0.00061 |
> | 60%     | 0.88153 ± 0.00092        | 0.89873 ± 0.00142 |
> | 70%     | 0.87577 ± 0.00129        | 0.89937 ± 0.00221 |
> | 80%     | 0.86740 ± 0.00318        | 0.89940 ± 0.00184 |
> | 90%     | 0.84297 ± 0.00284        | 0.89703 ± 0.00083 |
>
> ---
>
> ## **Table 2. Daxberger et al. (2021) vs. SPIN (Accuracy / NLL)**
>
> | Sparsity | Daxberger et al. (Acc / NLL) | SPIN (Acc / NLL) |
> |---------|-------------------------------|-------------------|
> | 0%      | 86.2% / 0.416                 | 87.9% / 0.384     |
> | 10%     | 85.2% / 0.461                 | 88.29% / 0.368    |
> | 20%     | 86.7% / 0.412                 | 88.59% / 0.357    |
> | 30%     | 86.9% / 0.392                 | 89.27% / 0.335    |
> | 40%     | 87.2% / 0.383                 | 89.51% / 0.325    |
> | 50%     | 87.2% / 0.389                 | 89.84% / 0.329    |
> | 60%     | 87.0% / 0.380                 | 89.74% / 0.320    |
> | 70%     | 86.8% / 0.390                 | 89.93% / 0.332    |
> | 80%     | 85.4% / 0.431                 | 89.68% / 0.330    |
> | 90%     | 86.0% / 0.422                 | 89.82% / 0.329    |
>
> ---
>
> ## **Experiment Settings**
>
> **ResNet-20 / CIFAR-10**
>
> **DNN Training**
> - epochs: 1000
> - learning rate: 0.001
> - optimizer: SGD
> - weight decay: 0.0001
> - momentum: 0.99
> - nesterov: False
>
> **Pruning Stage**
> - epochs: 1000
> - learning rate: 0.0001
> - optimizer: SGD
> - weight decay: 0.0001
> - momentum: 0.99
> - nesterov: False
>
> **BNN Training (SPIN / Pruned BNN)**
> - epochs: 90
> - mc_runs: 30
> - learning rate: 0.001
> - optimizer: SGD
> - weight decay: 0
> - momentum: 0.9
> - nesterov: False

---

> ### Author Response · Authors · 2025-11-20
> **Discussion of the crucial hyperparameter $\sigma$**
>
> ### Effect of σ in SPIN
>
> | σ    | Accuracy (mean) | Error/NLL (mean) |
> |------|------------------|-------------------|
> | 0.001 | 0.8969 | 0.3257 |
> | 0.010 | 0.8951 | 0.3276 |
> | 0.013 | 0.8976 (third) | 0.3247 (third) |
> | 0.016 | **0.8980 (best)** | **0.3182 (best)** |
> | 0.019 | 0.8969 | 0.3235 |
> | 0.022 | 0.8966 | 0.3269 |
> | 0.025 | 0.8980 (second) | 0.3233 (second) |
> | 0.050 | 0.8936 | 0.3313 |
> | 0.075 | 0.8946 | 0.3253 |
> | 0.100 | 0.8926 | 0.3348 |
> | 0.500 | 0.8926 | 0.3411 |
> | 1     | 0.8906 | 0.3504 |
> | 10    | 0.8923 | 0.3443 |
> | 100   | 0.8912 | 0.3438 |
>
>
> ### Experiment Settings
>
> 1. **DNN**
>    - epochs: 1000
>    - lr: 0.001
>    - optimizer: SGD
>    - weight_decay: 0.0001
>    - momentum: 0.99
>    - nesterov: False
>
> 2. **Pruning**
>    - epochs: 1000
>    - lr: 0.0001
>    - optimizer: SGD
>    - weight_decay: 0.0001
>    - momentum: 0.99
>    - nesterov: False
>
> 3. **BNN (SPIN)**
>    - epochs: 90
>    - mc_runs: 30
>    - lr: 0.001
>    - optimizer: SGD
>    - weight_decay: 0
>    - momentum: 0.9
>    - nesterov: False
>
>
> ### Response
>
> We thank the reviewer Vek4 for raising concerns about the sensitivity of the posterior standard deviation σ in SPIN.
> To evaluate this properly, we performed an extensive sweep over:
>
> σ ∈ {0.001, 0.010, 0.013, 0.016, 0.019, 0.022, 0.025, 0.050, 0.075, 0.1, 0.5, 1, 10, 100}
>
> across three independent seeds (40, 41, 42).
> The best average performance occurred at **σ = 0.016**.
>
> Importantly, values in the small range **σ ∈ [0.001, 0.025]** yielded nearly identical accuracy (0.896–0.898), demonstrating that SPIN is **not** sensitive to the exact value of σ within this region.
> Thus, our default σ is not cherry-picked but lies within a robust and stable regime.
>
> Since fully analyzing nonlinear BNNs is intractable, we try to provide in Supplementary Material Section E a **linear-model analysis** showing that, under SPIN, **smaller σ theoretically leads to lower expected test error**, matching our empirical observations. A fuller derivation needs further work.

---

> ### Author Response · Authors · 2025-11-20
> **Are the benefits tied to limitations of mean-field VI?**
>
> ### Response to Q3 — Are the benefits tied to limitations of mean-field VI?
>
> We thank the reviewer for this insightful question. You correctly noted that our use of Mean-Field VI (MFVI) was motivated by scalability, since MCMC methods are computationally infeasible for modern architectures such as ResNet or ViT.
>
> However, the key concern is whether SPIN’s performance gains are simply artifacts of MFVI’s limitations (e.g., variance collapse). Our findings indicate this is **not** the case.
>
> ---
>
> ### 1. Structural Reconditioning vs. MFVI Artifacts
>
> SPIN addresses a broader problem that affects *all* scalable inference methods:
> the severe ill-conditioning (flat and redundant directions) in overparameterized neural network posteriors.
>
> This issue hurts MFVI, Laplace approximations, and SGLD alike.
> SPIN directly mitigates this by suppressing redundant directions **before** inference begins, through its structured prior.
>
> ---
>
> ### 2. Evidence from Q4: Alignment with MCMC
>
> The strongest argument against the “MFVI artifact” concern is the MCMC alignment shown in Q4
> (Answer about Q4 is written in the global response "STRUCTURAL PATTERNS IN MASKS FROM PRUNING"):
>
> - If SPIN were merely exploiting an MFVI-specific weakness, its mask structure would be arbitrary or MFVI-dependent.
> - Instead, SPIN’s mask exactly mirrors the exploration structure of MCMC (Sommer et al. [3]).
>
> This indicates that SPIN captures a **universal structural property** of neural networks rather than a quirk of MFVI.
>
> ---
>
> ### Summary
>
> SPIN does **not** rely on limitations of mean-field VI.
> Rather, it explicitly encodes the ideal Bayesian geometry—revealed naturally by MCMC—into the prior.
> This reconditioning allows a scalable but limited inference method (MFVI) to behave more like an expressive sampler:
> **focusing** on important layers and **exploring** redundant ones.
>
>
> [3] Sommer, E., Wimmer, L., Papamarkou, T., Bothmann, L., Bischl, B., & Rügamer, D. (ICML 2024). Connecting the dots: is mode-connectedness the key to feasible sample-based inference in Bayesian neural networks?.

---

> ### Comment · Reviewer_Vek4 · 2025-11-21
>
> I thank the authors for the detailed response and for implementing the requested baselines (Pruned BNN and Subnetwork Inference). The empirical comparison against these baselines is helpful and I agree on the note about soft inductive biases. However, I retain doubts regarding the method's practicality and theoretical grounding. The argument that the masks align with MCMC patterns is certainly interesting, but ultimately relies on mimicking the findings of another (likely more faithful, but still) approximation rather than a first-principles justification.
>
> Furthermore, the reported computational cost (e.g., the extra 1000 epochs to run the pruning on ResNet20) seems quite expensive for a rather heuristic method aiming at scalability, in particular as the pruning's hyperparameters seem to have a non-negligible influence on the final performance.
>
> Question regarding the "Dense-Sparse-Dense" structure: Could this layer-wise uncertainty profile not simply be hardcoded (e.g., assigning loose priors to middle layers and tight priors to input/output layers), perhaps leveraging patterns from structured pruning?
> This would alleviate the massive computational burden of your current pipeline. Do you think that the expensive, granular, unstructured pruning step offers a verifiable benefit over such a simple layer-wise heuristic?

---

> > ### Author Response · Authors · 2025-11-29
> > **Response to Practicality, Pruning Cost, and the Value of Unstructured Masking (1)**
> >
> > Thank you for the thoughtful follow-up comment and for raising important concerns regarding practicality, theoretical grounding, and the role of the pruning stage.
> >
> > **(1) On the computational cost of pruning**
> > We acknowledge that the reported *1000 pruning epochs* may appear expensive at first glance. However, as clarified in Algorithm 1, this upper bound reflects a conservative setting: pruning is terminated via **early stopping** once the pruned model recovers the original NLL. In practice, convergence occurs significantly earlier. Moreover, the main paper reports results using a more realistic **90-epoch** schedule, which achieves nearly identical performance. We have now emphasized this implementation detail and the associated limitation in the revised manuscript.
> >
> > While our experiments employ **global unstructured iterative pruning**, we note that alternative approaches exist that considerably reduce computational cost, such as **one-shot saliency-based pruning** or **joint DNN-training-with-pruning** (e.g., *Zhu & Gupta, 2017*). Integrating such efficient variants into SPIN is a promising direction for future work, and we have updated the limitations section to clearly acknowledge this pathway.
> >
> > **(2) On whether a hard-coded layer-wise “Dense–Sparse–Dense” heuristic could replace unstructured pruning**
> > We appreciate the suggestion that a simple preset structure could potentially replace fine-grained pruning. To investigate this, we compared:
> > - Global magnitude-based pruning,
> > - Gradient-based saliency pruning,
> > - Random pruning,
> > - Structured L2 pruning,
> > - and the reviewer-proposed heuristic prior schedule.
> >
> > | Method | Acc (↑) | NLL (↓) | ECE (↓) | AUROC (ENT) (↑) |
> > |--------|--------|---------|---------|-----------------|
> > | Global L1 Pruning | 0.9024 ± 0.0028 | 0.3147 ± 0.0058 | 0.0303 ± 0.0025 | 0.8621 ± 0.0018 |
> > | Gradient Saliency | 0.9035 ± 0.0020 | 0.3170 ± 0.0079 | 0.0235 ± 0.0037 | 0.8647 ± 0.0018 |
> > | Random Pruning | 0.8894 ± 0.0031 | 0.3499 ± 0.0083 | 0.0308 ± 0.0022 | 0.8561 ± 0.0038 |
> > | Structured L2 Pruning | 0.8702 ± 0.0042 | 0.3976 ± 0.0071 | 0.0408 ± 0.0084 | 0.8380 ± 0.0035 |
> > | Heuristic | 0.8866 ± 0.0037 | 0.3521 ± 0.0069 | 0.0474 ± 0.0079 | 0.8548 ± 0.0041 |
> >
> > As noted, the heuristic baseline assigns layer-wise prior scales using a fixed schedule:
> > $\
> > \sigma_l = \sigma_{\min} + \frac{(\sigma_{\max} - \sigma_{\min}) \cdot l}{L - 1}, \qquad l = 0,\dots,L-1,
> > \$
> > where we set  $\(\sigma_{\min} = 10^{-3}\)$, $\(\sigma_{\max} = 1.0\)$, and $\(\sigma = 1.0\)$ for pruned weights.
> > The final layer (output layer) uses $\(\sigma = \sigma_{\min}\)$ to enforce the tightest prior constraint.
> > This scheme does not rely on weight importance, but instead follows a depth-based rule intended to emulate a Dense–Sparse–Dense profile.
> >
> >
> >
> > These results indicate that **global unstructured pruning identifies substantially more meaningful importance structure** than either preset heuristics or structured channel-level pruning. Moreover, for ViT we observe distinct sparsity patterns between **self-attention blocks** and **MLP blocks** (shown in Fig. F-13 and Fig. F-14 in Appendix F), suggesting that the internal importance distribution is highly non-uniform and not easily captured by simple deterministic layer schedules. Thus, while heuristics appear appealing computationally, they do not replicate the empirical performance or uncertainty behavior achieved by SPIN.

---

> > ### Author Response · Authors · 2025-11-29
> > **Response to Practicality, Pruning Cost, and the Value of Unstructured Masking (2)**
> >
> > (3) First-Principles Motivation for SPIN
> >
> > Bayesian neural networks (BNNs) and deep neural networks (DNNs) are typically heavily over-parameterized, and specifying an appropriate weight prior in BNNs remains a fundamentally difficult problem. A well-documented consequence of this challenge is the cold-posterior effect, where performance improves when the influence of the prior is artificially reduced—an indication of significant prior mis-specification. Motivated by this issue, SPIN predicts weight importance via pruning and uses this information to design a hybrid prior that improves predictive performance and uncertainty quality.
> >
> > The guiding principle of SPIN aligns with the concept of **soft inductive bias**, recently emphasized by Wilson (2025). SPIN encourages posterior behavior to be shaped by softly guided structural information extracted from pruning.
> >
> > Crucially, SPIN constructs a prior that conveys dual information:
> >
> > - **Structural Uncertainty:** It identifies which nodes are salient versus redundant (topology).
> > - **Value Guidance:** It informs the posterior about what values the weights should gravitate toward, utilizing the $\mu$ values from the sparse DNN.
> >
> > Determining this configuration—namely, *“which node should possess what value and what level of importance”*—is a highly complex challenge. As demonstrated in our experiments with heuristics, manually designing such a relationship is infeasible in practice. SPIN leverages pruning to extract this information directly from data, effectively functioning as a structured soft inductive bias.
> >
> > While SPIN is not the first approach to incorporate weight importance—classical priors attempt this as well—methods such as spike-and-slab try to infer weight importance and values simultaneously during posterior inference. This dramatically increases the hypothesis space and optimization difficulty. In contrast, SPIN decouples these processes: it first estimates the importance and target values via pruning, and then performs posterior inference conditioned on that discovered structure. This decomposition significantly lowers the optimization burden while retaining the benefits of a meaningful, data-driven prior.
> >
> > Empirically, we further observe that the posterior behavior induced by SPIN is qualitatively consistent with the MCMC exploration patterns reported by Sommer et al. (2024), where multiple chains occupy distinct yet connected posterior regions and exhibit increased variability in deeper layers. This suggests that SPIN guides inference toward semantically meaningful regions of the parameter space while still allowing nontrivial exploration along redundant directions.
> >
> > **References**
> >
> > [1] Emanuel Sommer, Diego Mesquita, Sebastian Farquhar, et al.
> > *Connecting the dots: Is mode-connectedness the key to feasible sample-based inference in Bayesian neural networks?*
> > arXiv preprint arXiv:2402.01484, 2024.
> >
> > [2] Andrew Gordon Wilson.
> > *Deep learning is not so mysterious or different.*
> > arXiv preprint arXiv:2503.02113, 2025.
> >
> > [3] Michael Zhu and Suyog Gupta.
> > *To prune, or not to prune: Exploring the efficacy of pruning for model compression.*
> > arXiv preprint arXiv:1710.01878, 2017.

---

### Official Review · Reviewer_xbfH · 2025-10-25

**Soundness:** 3
**Presentation:** 3
**Contribution:** 2
**Rating:** 4
**Confidence:** 4

**Summary:**

The paper introduces Sparsity-Informed priors for Bayesian Neural networks (SPIN), a method to empirically design Gaussian priors for BNNs by leveraging weight pruning on a pretrained deterministic DNN. Important (unpruned) weights receive low-variance priors centered at their post-pruning values, while pruned weights get high variance, zero-mean priors. This approach aims to encode parameter importance into the prior, improving BNN performance. Evaluations on image classification datasets such as the CIFAR-10/100 and TinyImageNet, across various architectures show gains in accuracy, negative log-likelihood (NLL), and out-of-distribution (OOD) detection compared to baselines like isotropic Gaussian, Laplace, Student-t, Spike-and-Slab and MOPED. The method is positioned as an empirical Bayes strategy, with analysis on how sparsity regularizes BNN representations.

**Strengths:**

The paper demonstrates moderate originality by combining network pruning, inspired by the lottery ticket hypothesis, with BNN prior design in a straightforward way. While data-driven priors already exist, e.g. MODEP, SPIN innovatively uses pruning to differentiate weight importance, creating a hybrid prior that reflects sparsity-induced structure. In terms of quality, the experimental setup is solid, covering diverse architectures and datasets, with comprehensive evaluation metrics including accuracy, NLL, ECE and OOD AUROC. Comparisons to multiple baselines are fair and the ablation study on sparsity and σ adds rigor. The analysis in Section 6, using UMAP visualization and loss landscapes, provides insightful evidence that SPIN enhances representation diversity and flattens the loss surface. The clarity of the paper is also solid, with concise writing, clear figures and organized sections. Significance is evident in addressing the prior selection BNN challenge which is often computationally expensive. SPIN’s consistent improvements, for example in accuracy, suggest practical value and highlights the pruning’s role in uncertainty modeling.

**Weaknesses:**

While SPIN shows empirical gains, the paper lacks a deeper theoretical justification for why pruning-derived sparsity translates effectively to BNN priors. For instance, the choice of σ=0.001 for important weights is tuned empirically, but no analysis explores why such a tight variance outperforms alternatives, making the method feel heuristic rather than principled. The computational overhead is a notable drawback since SPIN requires pretraining a DNN, iterative pruning/fine-tuning to find maximal sparsity and then VI for the BNN. This multi-stage process could double training time compared to standard priors, yet the paper does not quantify this cost or compare it to baselines. For scalability, experiments on larger datasets, e.g. full ImageNet or models are missing, limiting generalizability. Experimentally, while predictive models improve, calibration (ECE) often worsens, suggesting overconfidence. Sparsity levels vary widely across experiments, determined by validation NLL recovery, but sensitivity to pruning method is unaddressed. Additionally, the spike-and-slab baseline fails completely in some cases, but without hyperparameter details, it is unclear if this is a fair comparison. Finally, the reliance on the initial DNN’s quality is a critical limitation. If the DNN underfits, SPIN offers little benefit. While the paper acknowledges this, it provides no mitigation strategies such as adaptive pruning or warm-starting.

**Questions:**

1.	How robust is SPIN to different pruning techniques? The paper uses global magnitude-based-pruning, have you tested one-shot method like structured pruning, e.g. filter-level?
2.	Given the multi-stage overhead, is there a way to integrate pruning into end-to-end BNN training? This could make SPIN more practical. Experiments comparing compute times to baselines would help evaluate trade-offs.
3.	On harder datasets like CIFAR-100/TinyImageNet, ViT improvements are marginal. Does this stem from weaker initial DNNs and if so, could hybrid pretraining, e.g. with CNN features, strengthen the sparse backbone? Ablations here might reveal ways to extend SPIN to underfitting scenarios.
4.	How does SPIN compare to other-sparsity inducing BNN methods beyond the spike-and-slab, such as horseshoe priors? If baselines like spike-and-slab were unstable due to tuning, re-running with optimized hyperparameters could strengthen claims of superiority.

---

> ### Author Response · Authors · 2025-11-20
> **End-to-End Implementation and Computational Cost**
>
> ### Response regarding End-to-End Implementation and Computational Cost
>
> We agree with the reviewer that, in principle, **SPIN** could be implemented in a fully end-to-end manner by replacing our empirical two-stage prior construction with a sparsity-inducing prior such as **spike-and-slab** or **horseshoe**. These priors implicitly learn weight importance during training, and an end-to-end SPIN variant that directly optimizes such a prior is conceptually plausible.
>
> However, our experiments indicate that this direction is **computationally and practically much less attractive** than the current SPIN design. While SPIN appears multi-stage (pretrain $\rightarrow$ prune $\rightarrow$ BNN), its Bayesian phase uses a simple Gaussian prior with a closed-form KL term and no hierarchical latent scales. In contrast, spike-and-slab, Laplace, Student-t, and horseshoe priors require additional sampling and log-density evaluations (e.g., gamma functions, logarithms, error functions, and Monte Carlo expectations) in every forward and backward pass. These operations are both per-parameter expensive and also poorly fused on current GPU stacks, resulting in significant slowdowns driven by kernel-launch overhead rather than FLOPs.
>
> To make this concrete, we implemented MFVI with different priors applied to the same Conv2d layer and measured per-step cost (forward + backward) on an RTX 3090 (batch size 128, $64 \rightarrow 128$ channels, $3 \times 3$ kernel). All priors share the same convolutional FLOPs ($\approx 58\text{G}$), yet the wall-clock differences are substantial:
>
> | Prior | Time (ms) | Base FLOPs | Overhead | Total FLOPs | Speed |
> | :--- | :--- | :--- | :--- | :--- | :--- |
> | **normal** | 2.83 | 57.98 G | 516,992 | 57.98 G | 20.47 TF/s |
> | **spike-and-slab** | 2.93 | 57.98 G | 6.28 M | 57.99 G | 19.81 TF/s |
> | **laplace** | 3.08 | 57.98 G | 1.70 M | 57.98 G | 18.82 TF/s |
> | **student-t** | 3.37 | 57.98 G | 7.39 M | 57.99 G | 17.20 TF/s |
> | **horseshoe** | 8.57 | 57.98 G | 13.66 M | 58.00 G | 6.77 TF/s |
>
> Relative to the Gaussian prior, heavy-tailed priors are up to $3\times$ slower per step, despite nearly identical FLOPs. When scaled to a full ResNet-20 BNN on CIFAR-10 with identical hardware and protocol:
>
> | Prior | Runs | Mean Training Time | Std Dev | Min | Max |
> | :--- | :--- | :--- | :--- | :--- | :--- |
> | **normal** | 8 | 1:55:55 | 0:00:26 | 1:55:00 | 1:56:31 |
> | **laplace** | 9 | 6:02:21 | 0:00:29 | 6:01:27 | 6:02:51 |
> | **student-t** | 10 | 8:55:47 | 0:00:23 | 8:54:56 | 8:56:07 |
> | **spike-and-slab** | 9 | 3:15:18 | 0:00:15 | 3:14:53 | 3:15:49 |
> | **horseshoe** | 10 | 1 day, 11:33:22 | 0:00:09 | 1d 11:33:10 | 1d 11:33:35 |
>
> Thus, although end-to-end spike-and-slab or horseshoe BNNs are formally feasible, in practice they are **$2-5\times$ slower**, and horseshoe BNNs can be an order of magnitude slower.
>
> By contrast, SPIN's additional "stages" (standard dense pretraining and magnitude pruning) reuse inexpensive operations common in many pipelines. Once the salient subnetwork is identified, the Bayesian stage is as cheap as a standard Gaussian MFVI BNN. As a result, the total compute of SPIN is competitive—and often lower—than that of a single-stage BNN with heavy-tailed priors, while delivering better calibration and OOD performance.
>
> We further agree that iterative magnitude pruning can become costly for large models. This is not a fundamental limitation of SPIN but of the specific pruning schedule we used for clarity. **SPIN is agnostic to the choice of importance measure**: any mechanism that identifies a salient subnetwork can be used. In particular:
>
> * **One-shot pruning** (gradient norms, Hessian diagonal)
> * **Structured pruning**
> * **Gradual pruning integrated into training** (Zhu & Gupta, 2017)
>
> can all replace the iterative pruning stage without modifying SPIN's Bayesian formulation. This allows SPIN to leverage decades of pruning research—including approaches that approximate "end-to-end pruning + training" within a single optimization loop—while still retaining the efficiency of a Gaussian MFVI BNN.
>
> ### Summary
>
> * **Conceptual compatibility:** SPIN can be implemented with end-to-end sparsity-inducing priors.
> * **Practical limitation:** End-to-end spike-and-slab or horseshoe BNNs incur substantial computational overhead ($2-5\times$ slower).
> * **Design advantage:** SPIN uses pruning-based importance to enable a simple Gaussian prior that yields strong uncertainty/OOD performance at a fraction of the cost.
> * **Flexibility:** SPIN can plug in any pruning strategy—including gradient/Hessian saliency or gradual pruning—to reduce multi-stage overhead.
>
> We will clarify these trade-offs and the connection to pruning-based importance measures in the revised manuscript.
>
> ### References
>
> Zhu, M. & Gupta, S. (2017). To prune, or not to prune: Exploring the efficacy of pruning for model compression. *arXiv:1710.01878*.

---

> ### Author Response · Authors · 2025-11-20
> **Comparison with Alternative Priors**
>
> ### ResNet-20 CIFAR-10 (Prior Comparison)
>
> | Prior        | ID Accuracy           | ID NLL               | OOD ECE             | OOD AUROC (Entropy)  |
> |--------------|-----------------------|----------------------|---------------------|----------------------|
> | Horseshoe    | 0.89159 ± 8.26e-06    | 0.32478 ± 3.65e-05   | 0.06095 ± 6.94e-06  | 0.86022 ± 1.09e-05   |
> | Laplace      | 0.89314 ± 8.40e-06    | 0.33739 ± 1.03e-04   | 0.03212 ± 1.80e-05  | 0.85611 ± 8.16e-06   |
> | Normal       | 0.89223 ± 6.69e-06    | 0.34018 ± 3.04e-05   | 0.02751 ± 1.41e-05  | 0.85641 ± 6.25e-06   |
> | Spike-and-Slab | 0.89171 ± 6.40e-06  | 0.34610 ± 9.02e-05   | 0.02434 ± 5.37e-06  | 0.85564 ± 1.05e-05   |
> | Student-t    | 0.89316 ± 3.36e-06    | 0.33995 ± 4.18e-05   | 0.02819 ± 5.24e-06  | 0.85776 ± 5.69e-06   |
> | MOPED        | 0.89066 ± 1.07e-05    | 0.34420 ± 5.88e-05   | 0.02831 ± 1.19e-05  | 0.85563 ± 6.52e-06   |
> | **SPIN**     | 0.90431 ± 9.37e-06    | 0.30997 ± 5.57e-05   | 0.03047 ± 5.01e-06  | 0.86410 ± 1.10e-05   |
>
>
> ### ViT CIFAR-10 (Spike-and-Slab Sweep vs. SPIN)
>
> | π   | ID Accuracy       | ID NLL            | ECE              | OOD AUROC (MSP) | OOD AUROC (Entropy) | OOD AUROC (MI) |
> |-----|-------------------|-------------------|------------------|------------------|----------------------|----------------|
> | 0.3 | 0.5314 ± 0.2231   | 1.2779 ± 0.5683   | 0.0236 ± 0.0013  | 0.6273 ± 0.1006  | 0.6453 ± 0.1047      | 0.6077 ± 0.0926 |
> | 0.5 | 0.6445 ± 0.0333   | 1.0019 ± 0.0990   | 0.0236 ± 0.0014  | 0.6786 ± 0.0194  | 0.6973 ± 0.0252      | 0.6467 ± 0.0219 |
> | 0.7 | 0.6341 ± 0.0298   | 1.0213 ± 0.0833   | 0.0222 ± 0.0035  | 0.6773 ± 0.0194  | 0.6954 ± 0.0226      | 0.6396 ± 0.0247 |
> | **SPIN** | 0.7467        | 0.7213            | 0.0175           | 0.7238           | 0.7486               | 0.7176          |
>
> ### DenseNet-30 CIFAR-100 (Spike-and-Slab Sweep vs. SPIN)
>
> | π   | ID Accuracy       | ID NLL           | ECE              | MSP AUROC        | Entropy AUROC       | MI AUROC        |
> |-----|-------------------|------------------|------------------|------------------|---------------------|-----------------|
> | 0.3 | 0.5314 ± 0.2231   | 1.2779 ± 0.5683  | 0.0236 ± 0.0013  | 0.6273 ± 0.1006  | 0.6453 ± 0.1047     | 0.6077 ± 0.0926 |
> | 0.5 | 0.6445 ± 0.0333   | 1.0019 ± 0.0990  | 0.0236 ± 0.0014  | 0.6786 ± 0.0194  | 0.6973 ± 0.0252     | 0.6467 ± 0.0219 |
> | 0.7 | 0.6341 ± 0.0298   | 1.0213 ± 0.0833  | 0.0222 ± 0.0035  | 0.6773 ± 0.0194  | 0.6954 ± 0.0226     | 0.6396 ± 0.0247 |
> | **SPIN** | 0.6837        | 1.2025           | 0.0537           | 0.7527           | 0.7699              | 0.7485          |
>
>
> We thank the reviewer xbfH for raising this point. In our experiments, we performed an extensive sweep over the sparsity parameter $\pi \in \{0.3, 0.5, 0.7\}$ for spike-and-slab, averaging each configuration over three independent runs, and additionally evaluated a horseshoe prior for completeness.
>
> **ResNet-20 CIFAR-10:**
> SPIN outperforms both spike-and-slab and horseshoe across all metrics except ECE. The test was conducted over 10 independent runs.
>
> **ViT CIFAR-10:**
> SPIN exceeds all spike-and-slab settings across *every* metric, including accuracy, NLL, calibration, and OOD detection.
>
> **DenseNet-30 CIFAR-100:**
> SPIN achieves stronger overall performance on most metrics, with spike-and-slab slightly outperforming SPIN only on ID NLL and ECE for certain values of $\pi$.
>
> **Overall, these results indicate that SPIN provides consistently stronger performance even when spike-and-slab is carefully tuned.**
>
> The SPIN results are directly taken from the values reported in our main paper.

---

> ### Author Response · Authors · 2025-11-20
> **Response to the Q1, Q3 and weakness**
>
> **Response to Q1: Robustness to Different Pruning Techniques**
>
> We agree that evaluating robustness across different pruning criteria is essential to validate the method's dependency on mask quality. As detailed in our Global Response, we conducted a comprehensive ablation study comparing High-Quality masks (Gradient-based, Global L1) vs. Low-Quality masks (Random, Structured L2/Overshoot).
>
> Please refer to the "Response to Question regarding Mask Correctness and Sensitivity" in the Global Response for the full quantitative comparison. In summary:
>
> - Structured Pruning: As the reviewer suspected, structured pruning (e.g., filter-level) tends to “overshoot,” removing entire feature pathways necessary for constructing an informative prior, leading to lower performance than unstructured pruning.
>
> - Random Pruning: Results in significant degradation, confirming that SPIN meaningfully leverages the specific weight importance structures, not just sparsity levels.
>
> ---
>
> **Response to Q3: ViT Performance and Hybrid Pretraining**
>
> We appreciate the insightful suggestion regarding the marginal improvements in ViT on harder datasets. We agree that since SPIN relies on a pre-trained backbone, its performance is bounded by the quality of the initial deterministic model (i.e., underfitting).
>
> Your suggestion to use hybrid pretraining (e.g., utilizing CNN features) to strengthen the sparse backbone is excellent and could likely mitigate this underfitting issue. We are currently running additional experiments to verify this hypothesis as per your suggestion. We expect to share these quantitative results within the next week to demonstrate how SPIN behaves when the backbone quality is improved.
>
> ---
>
> **Response to Weakness: Theoretical Justification for Pruning-Derived Priors**
>
> The reviewer raised a valid point that the method relies on empirical findings and might appear heuristic. However, we emphasize that the sparsity pattern derived from pruning is not arbitrary; it closely mirrors the posterior uncertainty geometry of deep networks.
>
> Recent studies on MCMC-based BNNs (Sommer et al., 2024) report a “Low–High–Low” uncertainty profile across layers (Low uncertainty at input/output, High in the middle). Our layer-wise pruning analysis reveals a matching “Dense–Sparse–Dense” pattern.
>
> SPIN effectively translates this structural finding into a prior:
> - assigning tight priors (low variance) to dense layers, and
> - assigning loose priors (high variance) to sparse layers.
>
> For detailed discussion and visualizations of this connection, please refer to “STRUCTURAL PATTERNS IN MASKS FROM PRUNING” in the Global Response. Furthermore, beyond this structural alignment, we are currently working on formalizing a theoretical framework to guide the specific selection of prior standard deviations (e.g., $\sigma$ settings), moving beyond empirical tuning. We plan to incorporate this theoretical analysis into the final revision to further strengthen the principled foundation of SPIN.
>
> ---

---

> > ### Comment · Reviewer_xbfH · 2025-11-28
> >
> > I thank the authors for their response.  The reliance on the initial DNN’s quality is still a critical limitation.  The authors claimed "SPIN outperforms both spike-and-slab and horseshoe across all metrics except ECE. The test was conducted over 10 independent runs." More runs should be conducted to have a reliable result. Spike-and-slab prior is known to be difficult to use because of the point mass causing discreteness. Other global-local priors may be more suitable. How about controlling false positives, e.g., FDR?

---

> ### Author Response · Authors · 2025-11-30
> **Response to Reviewer xbfH: Hybrid CNN+ViT Architecture, Horseshoe+FDR Perspective, and Statistical Reliability**
>
> We thank the reviewer for these thoughtful comments and suggestions.
>
> ### **Response to Q3 (Underfitting and Hybrid CNN+ViT Architecture)**
> We agree that underfitting in the ViT configuration may have limited the performance of SPIN and competing baselines. Following the reviewer’s helpful suggestion, we implemented a hybrid CNN+ViT architecture to mitigate this issue, and re-ran all comparisons under the same evaluation protocol. As shown in **Table 1**, once underfitting is reduced, **SPIN demonstrates a substantial performance improvement** and clearly outperforms alternative priors across ACC, NLL, and AUROC, while remaining competitive on ECE.
>
> **Table 1. Hybrid CNN+ViT results (higher ACC / AUROC and lower NLL / ECE are better)**
>
> | Model         | ACC    | NLL    | ECE    | AUROC (ENT) |
> |---------------|--------|--------|--------|--------------|
> | normal        | 0.5745 | 1.7675 | **0.0121** | 0.8464       |
> | laplace       | 0.5707 | 1.7737 | 0.0187 | 0.8457       |
> | student-t     | 0.5605 | 1.8231 | 0.0151 | 0.7868       |
> | spike-and-slab| 0.5648 | 1.7863 | 0.0176 | 0.7576       |
> | MOPED         | 0.5695 | 1.8196 | 0.0222 | 0.8475       |
> | horseshoe     | 0.5925 | 1.7038 | 0.0290 | 0.8387       |
> | **SPIN (Ours)** | **0.6333** | **1.5260** | 0.0392 | **0.8693** |
>
> ---
>
> ### **Comparison to Global–Local Priors and Horseshoe + FDR**
> We also appreciate the reviewer’s point regarding the challenges of spike-and-slab priors due to their point-mass component. Motivated by this, we additionally compared **SPIN** against the global–local **horseshoe** prior and other standard priors. As shown in **Table 2**, **SPIN consistently outperforms spike-and-slab, horseshoe, and other alternatives** across ID accuracy, ID NLL, and OOD AUROC, while retaining competitive OOD calibration (ECE).
>
> Furthermore, we agree that controlling false positives—e.g., via **False Discovery Rate (FDR)**—is an important complementary goal. Recent work (Liang et al., 2025) demonstrates how frequentist-assisted horseshoe methods can achieve finite-sample FDR guarantees, suggesting a natural baseline to compare against and a promising direction for extending SPIN to include explicit FDR-aware selection mechanisms. We plan to pursue this direction in future work.
>
> **Table 2. Performance of various priors (ResNet-20, CIFAR-10, 10-run avg)**
>
> | Prior           | ID Acc (↑)      | ID NLL (↓)       | OOD ECE (↓)       | OOD AUROC (ENT ↑) |
> | :---            | :---            | :---             | :---             | :---              |
> | Horseshoe       | 0.8916 ± 8e-6   | 0.3248 ± 3e-5    | 0.0610 ± 7e-6     | 0.8602 ± 1e-5     |
> | Laplace         | 0.8931 ± 8e-6   | 0.3374 ± 1e-4    | 0.0321 ± 1e-5     | 0.8561 ± 8e-6     |
> | Normal (Iso)    | 0.8922 ± 6e-6   | 0.3402 ± 3e-5    | 0.0275 ± 1e-5     | 0.8564 ± 6e-6     |
> | Spike-and-Slab  | 0.8917 ± 6e-6   | 0.3461 ± 9e-5    | **0.0243 ± 5e-6**     | 0.8556 ± 1e-5     |
> | Student-t       | 0.8932 ± 3e-6   | 0.3399 ± 4e-5    | 0.0282 ± 5e-6     | 0.8578 ± 5e-6     |
> | MOPED           | 0.8907 ± 1e-5   | 0.3442 ± 5e-5    | 0.0283 ± 1e-5     | 0.8556 ± 6e-6     |
> | **SPIN (Ours)** | **0.9043 ± 9e-6** | **0.3099 ± 5e-5** | 0.0305 ± 5e-6 | **0.8641 ± 1e-5** |
>
> ---
>
> ### **Statistical Reliability and Multi-Seed Evaluation Plan**
> Multiple reviewers requested deeper statistical validation. We have already expanded our experiments to **10 independent seeds**, and within the remaining time we **plan to increase to at least 30 seeds** and report aggregated summary statistics accordingly. We expect this to further reinforce the robustness of the presented conclusions.
>
> ---
>
> ### **Reference**
> Liang, Qiaoyu, et al. *"False Discovery Rate Control via Frequentist-assisted Horseshoe."* arXiv preprint arXiv:2502.05460 (2025).

---

### Official Review · Reviewer_4rDT · 2025-10-29

**Soundness:** 3
**Presentation:** 2
**Contribution:** 2
**Rating:** 4
**Confidence:** 4

**Summary:**

This paper proposes SPIN (Sparsity-Informed Priors for Bayesian Neural Networks), an empirical Bayes framework that constructs data-driven priors based on weight importance revealed through pruning a pretrained deterministic model. The approach is simple yet effective, yielding consistent gains in predictive performance and uncertainty estimation across CNN and ViT architectures.

**Strengths:**

**[S1] Novelty and Conceptual Clarity:** The idea of using network pruning to derive empirical priors is intuitive and well-motivated.

**[S2] Extensive Experiments:** The paper provides extensive experiments across multiple datasets (CIFAR-10/100, TinyImageNet) and architectures (ResNet, DenseNet, MobileNet, ViT).

**[S3] Clarity:** The method is well-explained with equations, figures, and pseudocode (Algorithm 1).

**Weaknesses:**

## Major Concerns
**[W1] Lack of Explanation about Pruning:** The discussion of pruning-related literature is overly brief. Since the proposed method hinges on pruning to determine weight importance, it would be beneficial to include a more comprehensive overview of representative pruning techniques (e.g., magnitude-based, sensitivity-based, and structured pruning) and clarify how SPIN builds upon or diverges from them.


**[W2] Insufficient Methodological Detail on Weight Partitioning:** The methodology section provides insufficient detail about how pruning is performed to distinguish important versus redundant weights. Since this partitioning directly determines the prior structure, a deeper analysis of pruning sensitivity (e.g., across different thresholds or pruning criteria) would strengthen the methodological rigor.


**[W3] Unclear Justification for Using Gaussian Noise on Pruned Weights:** The decision to assign pruned weights a high-variance Gaussian prior (effectively adding random Gaussian noise) lacks clear justification. The authors state that this design introduces a regularization effect, but the explanation is not entirely convincing. In conventional pruning, pruned parameters are usually zeroed out rather than replaced with noise, so it is unclear how adding stochasticity meaningfully improves generalization. A more concrete theoretical or empirical motivation for this “regularization through noise” would help the reader understand its necessity.


**[W4] Questionable Advantage of SPIN over Sparse DNNs**: In Figure 4, the Sparse DNN often matches or even exceeds the performance of the SPIN-based BNN (especially in ViT). This raises a critical question: *if a sparse deterministic model performs comparably without Bayesian overhead, when is SPIN truly advantageous?* The authors should better justify why the added computational complexity of BNN inference is worthwhile in such cases, possibly by analyzing uncertainty metrics or robustness benefits.


**[W5] Unrealistic Scale of $\sigma$ in Figure 2:** The variance values tested in Figure 2 ($\sigma$ = 0.001, 1, 10, 100) are separated by several orders of magnitude, making the conclusion somewhat trivial—naturally, very large $\sigma$ values yield poor performance. Including intermediate settings (e.g., $\sigma$ = 0.01 or 0.1) would provide a more realistic and convincing analysis of the prior variance’s sensitivity.


**[W6] Lack of Computational Complexity Analysis:** The paper omits a quantitative analysis of computational overhead. Since SPIN requires iterative pruning, the total cost is likely several times that of standard BNN training. Providing wall-clock runtime or FLOP comparisons—and explicitly discussing this as a limitation—would make the contribution more transparent.

-----
## Minor Concerns
**[W7] Low Figure Readability:** Several figures (e.g., Figures 2, 4, and 5) appear blurry or low-resolution. Improving text clarity and resolution would enhance readability and presentation quality.


**[W8] Missing Equation Numbering:** Important equations (e.g., KL divergence decomposition, hybrid prior definition) lack numbering, which makes cross-referencing within the text cumbersome and reduces readability.

**[W9] Ambiguous Notation for SPIN(0.x) in Tables:** Clarify the notation “SPIN(0.x)” in tables. Although it refers to the sparsity level of the base model used to derive the prior, this should be explicitly stated near the first table for clarity.


**[W10] Lack of Context for Loss Landscape Analysis:** The discussion around loss landscapes (Section 6.2, Figure 5) lacks sufficient context. Readers unfamiliar with this concept may not understand why flatter minima are desirable or how they relate to generalization. A brief explanation or citation would improve accessibility.

**[W11] Absence of Quantitative Flatness Evaluation:** While Figure 5 provides qualitative evidence of flatter loss surfaces, the differences are visually subtle. Quantitative evaluation (e.g., Hessian eigenvalue spectra or trace-based flatness metrics) would make the argument more convincing.

**Questions:**

**[Q1] Possibility of Adding Gradient/Hessian-Based Prior Comparison:** The proposed method infers weight importance solely through pruning, but alternative importance measures—such as gradient norms or Hessian-based saliency—could provide more theoretically grounded and computationally efficient ways to construct informative priors. Although the rebuttal period is short, would it be feasible for the authors to conduct or at least discuss a small-scale comparison between SPIN and such gradient/Hessian-based prior constructions? Even a limited experiment or qualitative analysis could help clarify whether the pruning-based approach offers distinct advantages over these sensitivity-based alternatives.

**[Q2] Clarification on Regularization Effect of Gaussian Priors:** Could the authors elaborate on how assigning a high-variance Gaussian prior to pruned weights concretely contributes to regularization? In particular, how does this differ from simply setting those parameters to zero, and is there empirical evidence that the injected stochasticity improves generalization or uncertainty calibration?

---

> ### Author Response · Authors · 2025-11-20
> **Response to Reviewer 4rDT**
>
> We sincerely thank the reviewer 4rDT for the constructive feedback and for acknowledging the novelty and clarity of our approach. We address your concerns below, particularly regarding the pruning methodology, the justification for our prior design, and computational complexity.
>
> ---
>
> ## 1. Pruning Methods and Sensitivity Analysis [W1, W2, Q1]
>
> We agree that relying solely on one pruning method limits the methodological analysis. In response to [W1] and [W2], we conducted a comprehensive ablation study comparing SPIN's performance across different mask qualities: High-Quality (Gradient-based, Global L1) vs. Low-Quality (Random, Structured L2/Overshoot).
>
> Our results show that SPIN is indeed sensitive to mask quality:
>
> - High-quality masks (Gradient, Global L1) yield the best Accuracy and OOD AUROC.
> - Random masks lead to significant degradation, confirming that SPIN leverages the specific structural importance of weights, not just noise injection.
>
> Please refer to the "Response to Question regarding Mask Correctness and Sensitivity" in the Global Response for the full quantitative comparison and detailed analysis.
>
> ---
>
> ## 2. Justification for Gaussian Priors on Pruned Weights [W3, Q2]
>
> We appreciate the reviewer’s question on whether the high-variance Gaussian prior meaningfully contributes to regularization compared to simply zeroing out weights. To verify this, we implemented a "Pruned BNN" baseline, where pruned weights are strictly set to zero (non-trainable) while remaining weights undergo standard VI.
>
> ### Pruned BNN vs. SPIN (Accuracy ± Std)
>
> | Sparsity | Pruned BNN (Acc ± Std) | SPIN (Acc ± Std) |
> |---------|--------------------------|-------------------|
> | 0%      | 0.8908 ± 0.00065         | 0.87917 ± 0.00017 |
> | 10%     | 0.8890 ± 0.00229         | 0.88153 ± 0.00097 |
> | 20%     | 0.89067 ± 0.00354        | 0.88670 ± 0.00106 |
> | 30%     | 0.88967 ± 0.00314        | 0.89283 ± 0.00066 |
> | 40%     | 0.88857 ± 0.00368        | 0.89440 ± 0.00051 |
> | 50%     | 0.88527 ± 0.00152        | 0.89883 ± 0.00061 |
> | 60%     | 0.88153 ± 0.00092        | 0.89873 ± 0.00142 |
> | 70%     | 0.87577 ± 0.00129        | 0.89937 ± 0.00221 |
> | 80%     | 0.86740 ± 0.00318        | 0.89940 ± 0.00184 |
> | 90%     | 0.84297 ± 0.00284        | 0.89703 ± 0.00083 |
>
> ### Experiment Settings
> ResNet-20 / CIFAR-10
>
> **DNN Training**
> - epochs: 1000
> - learning rate: 0.001
> - optimizer: SGD
> - weight decay: 0.0001
> - momentum: 0.99
> - nesterov: False
>
> **Pruning Stage**
> - epochs: 1000
> - learning rate: 0.0001
> - optimizer: SGD
> - weight decay: 0.0001
> - momentum: 0.99
> - nesterov: False
>
> **BNN Training (SPIN / Pruned BNN)**
> - epochs: 90
> - mc_runs: 30
> - learning rate: 0.001
> - optimizer: SGD
> - weight decay: 0
> - momentum: 0.9
> - nesterov: False
>
> As shown in the table above, SPIN consistently outperforms the hard-zero Pruned BNN baseline in accuracy for sparsity levels ≥ 30%, and also achieves substantially higher accuracy. These results indicate that the injected posterior variability from the high-variance prior provides a meaningful benefit, especially in moderately or highly sparse regimes.
>
> For a deeper insight into why maintaining variance (via priors) instead of hard-zeroing is structurally beneficial, please refer to the Global Response. There, we discuss how the "Dense–Sparse–Dense" layer-wise structure revealed by pruning naturally aligns with the posterior uncertainty profile (Low–High–Low) observed in MCMC studies, necessitating a high-variance prior in sparse layers for adequate exploration.
>
> ---
>
> ## 3. Computational Complexity and Overhead [W6]
>
> Regarding the computational cost, we have performed a detailed breakdown of wall-clock time and FLOPs comparing SPIN against end-to-end Bayesian baselines (e.g., Horseshoe, Spike-and-Slab).
>
> While SPIN involves a multi-stage process (pretrain → prune → BNN), the final BNN stage utilizes a simple Gaussian prior, making it significantly faster per-step than heavy-tailed priors which require expensive sampling operations.
>
> - **SPIN (Gaussian MFVI): ~20.47 TF/s**
> - **Horseshoe BNN: ~6.77 TF/s (≈ 3× slower)**
>
> For a detailed runtime and FLOPs analysis, please refer to the "End-to-End Implementation and Computational Cost" section in the response to Reviewer xbfH.

---

> > ### Comment · Reviewer_4rDT · 2025-11-25
> >
> > Thank you for the considerable effort you put into the additional experiments and the detailed responses within such a short rebuttal period. Several of my concerns have indeed been clarified through the new results. However, a number of important points still remain unaddressed or only partially addressed.
> >
> > 1. **Revisions to the main manuscript**:
> >     - Regarding W1, I appreciate the new ablation studies on mask quality — they were very helpful. That said, the requested discussion on representative pruning techniques (e.g., magnitude-based, sensitivity-based, structured pruning) is still relatively limited in both the manuscript and the rebuttal. Since revisions to the main text are allowed, it would significantly strengthen the work if you could expand this literature overview during the rebuttal period.
> >     - It would also be beneficial if W7, W8, W9, and W10 could be incorporated into the revised manuscript, as these relate to clarity and readability rather than new experiments.
> >
> > 2. **Missing responses for W4 and W5**
> >     - It is possible that I may have overlooked them, but I could not find a direct response to W4 or W5 in the rebuttal. Clarification on these points would be appreciated.
> >
> > 3. **Clarification regarding W6**
> >     - My understanding is that the response mainly compares the computational efficiency of the Gaussian prior itself, rather than the full SPIN pipeline. While I appreciate the comparison to heavy-tailed priors, the pruning stage in SPIN still introduces considerable computational overhead. I believe this should be explicitly acknowledged as a limitation in the manuscript.
> >
> > Overall, I can see that substantial effort was made for the rebuttal, and I appreciate the additional work. However, since several items remain insufficiently addressed, it is difficult to raise the score at this moment. If the authors can meaningfully incorporate the above points during the remaining discussion period, I would be willing to reconsider my evaluation.

---

> > > ### Author Response · Authors · 2025-11-29
> > > **Response – Revisions to the Main Manuscript**
> > >
> > > We sincerely thank the reviewer for the valuable suggestion regarding improvements to the literature overview and clarity of presentation. We have expanded the discussion of pruning methods in the **Related Work** section to explicitly distinguish between **unstructured pruning** and **structured pruning**, as well as to summarize common pruning criteria such as **magnitude-based**, **sensitivity-based**, and **gradient-based** approaches. This update directly addresses [W1] and [W2], clarifying the landscape of representative pruning techniques and contextualizing SPIN within existing practices.
> > >
> > > In addition, we incorporated the reviewer’s minor concerns into the revised manuscript, focusing on improving clarity, readability, and organization (e.g., improved explanation flow, clearer terminology, and additional guidance around mask interpretation). These revisions do not alter the experimental claims but aim to strengthen accessibility and presentation quality.
> > >
> > > We appreciate the reviewer’s constructive feedback, and hope that these updates meaningfully enhance the clarity and completeness of the manuscript.

---

> > > ### Author Response · Authors · 2025-11-29
> > > **Response to W4 — Questionable Advantage of SPIN over Sparse DNNs**
> > >
> > > Thank you very much for the thoughtful comment.
> > >
> > > It is entirely reasonable to ask why, in the ViT setting, the BNN does not surpass the corresponding DNN in terms of accuracy (Acc) and negative log-likelihood (NLL). In fact, a performance gap between DNNs and BNNs has been reported repeatedly across many tasks, and this phenomenon is widely known in the literature as the *cold posterior* effect [1,2].
> > >
> > > One of the main reasons attributed to this cold posterior effect is the **mismatch or misspecification of the prior distribution** in BNNs [2]. In other words, many empirically observed cases where “Bayes should, in principle, do better, but does not in practice” can often be traced back to priors that are not well aligned with the actual data or model. Thus, a performance gap between DNNs and BNNs is not unusual; at the same time, prior work [2] suggests that **carefully designing the prior can substantially reduce or even close this gap**.
> > >
> > > Our paper is precisely concerned with this issue of **how to construct a good prior for BNNs**, and we summarize the relevant experiments in the tables below.
> > >
> > > ---
> > >
> > > ### **CNN architectures**
> > >
> > > | CIFAR-10 / ResNet-20 | Acc | NLL |
> > > | --- | --- | --- |
> > > | DNN | 0.8996 **(Second)** | 0.3237 **(Second)** |
> > > | Isotropic Gaussian | 0.8904 | 0.3503 |
> > > | Laplace | 0.8933 | 0.3429 |
> > > | Student-t | 0.8935 | 0.3386 |
> > > | Spike-and-Slab | 0.8886 | 0.3424 |
> > > | MOPED | 0.8879 | 0.3494 |
> > > | SPIN ($\sigma$ = 0.001) | 0.9107 **(Best)** | 0.3001 **(Best)** |
> > >
> > > In the CNN-based setting, **SPIN is the only BNN prior that *surpasses* the DNN in both Acc and NLL**, which strongly suggests that our prior modeling is more appropriate than the existing alternatives.
> > >
> > > ---
> > >
> > > ### **ViT architectures**
> > >
> > > | CIFAR-10 / ViT | Acc | NLL |
> > > | --- | --- | --- |
> > > | DNN | 0.7965 **(Best)** | 0.6092 **(Best)** |
> > > | Isotropic Gaussian | 0.6179 | 1.0780 |
> > > | Laplace | 0.6834 | 0.8862 |
> > > | Student-t | 0.6639 | 0.9472 |
> > > | Spike-and-Slab | 0.3321 | 1.7888 |
> > > | SPIN ($\sigma$ = 0.001) | 0.7467 **(Second)** | 0.7213 **(Second)** |
> > >
> > > In the ViT-based setting, **SPIN is the only BNN prior that comes close to the DNN’s performance**, substantially reducing the performance gap compared to other BNN baselines. This again indicates that our prior design is considerably better aligned with the model and data than the existing methods.
> > >
> > > ---
> > >
> > > ### **Expected Calibration Error (ECE)**
> > >
> > > | CIFAR-10 | DNN | SPIN ($\sigma$ = 0.001) |
> > > | --- | --- | --- |
> > > | ECE (ResNet-20) | 0.0412 | **0.0272** |
> > > | ECE (ViT) | 0.0568 | **0.0175** |
> > >
> > > These results indicate that, when the prior in a BNN is appropriately specified, we can not only approach (or even surpass) DNN-level accuracy, but also **enhance the intrinsic advantages of BNNs**, such as better calibration and more reliable uncertainty estimation. We believe this highlights the importance of principled prior design in making BNNs practically competitive and robust.
> > >
> > > [1] Wenzel, Florian, et al. *How good is the Bayes posterior in deep neural networks really?* PMLR, 2020.
> > > [2] Fortuin, Vincent, et al. *Bayesian neural network priors revisited.* arXiv preprint, ICLR, 2021.

---

> > > ### Author Response · Authors · 2025-11-29
> > > **Response to W5 — Unrealistic Scale of $\sigma$**
> > >
> > > Thank you for the thoughtful comment. The prior scale $\sigma$ indeed plays an important role in SPIN, so we provide both theoretical and empirical evidence regarding its behavior in **Appendix E**. Since analyzing full Bayesian neural networks is intractable, we use a simplified linear-model analysis to isolate the effect of $\sigma$. Under the assumption that pruning preserves predictive performance, our analysis shows that smaller values of $\sigma$ are theoretically preferable.
> > >
> > > To complement this theoretical result, we conducted additional experiments on CIFAR-10 with ResNet-20, sweeping $\sigma \in \{0.001, 0.013, 0.016, 0.019, 0.022, 0.025, 0.050, 0.075, 0.01, 0.05, 0.1, 0.5, 1, 10, 100\}$ across three random seeds (40, 41, 42). Importantly, we found that all prior standard deviations within $[0.001, 0.025]$ achieve nearly identical accuracy (0.896–0.898). This indicates that, as long as $\sigma$ is chosen within a reasonably small range, SPIN performance remains stable and effective.
> > >
> > > **Experiment Setting**
> > > 1. DNN: epochs 1000 / learning rate 0.001 / SGD / weight_decay 0.0001 / momentum 0.99 / nesterov False
> > > 2. Pruning: epochs 1000 / learning rate 0.0001 / SGD / weight_decay 0.0001 / momentum 0.99 / nesterov False
> > > 3. BNN: epochs 90 / mc_runs 30 / learning rate 0.001 / SGD / weight_decay 0 / momentum 0.9 / nesterov False
> > >
> > > | $\sigma$ | Acc. (mean) | Error / NLL (mean) |
> > > | --- | --- | --- |
> > > | 0.001 | 0.8969 | 0.3257 |
> > > | 0.010 | 0.8951 | 0.3276 |
> > > | 0.013 | 0.8976 *(third)* | 0.3247 *(third)* |
> > > | 0.016 | 0.8980 *(best)* | 0.3182 *(best)* |
> > > | 0.019 | 0.8969 | 0.3235 |
> > > | 0.022 | 0.8966 | 0.3269 |
> > > | 0.025 | 0.8980 *(second)* | 0.3233 *(second)* |
> > > | 0.050 | 0.8936 | 0.3313 |
> > > | 0.075 | 0.8946 | 0.3253 |
> > > | 0.100 | 0.8926 | 0.3348 |
> > > | 0.500 | 0.8926 | 0.3411 |
> > > | 1 | 0.8906 | 0.3504 |
> > > | 10 | 0.8923 | 0.3443 |
> > > | 100 | 0.8912 | 0.3438 |

---

> > > ### Author Response · Authors · 2025-11-29
> > > **Response to W6 — Clarification on the Computational Overhead of SPIN**
> > >
> > > Thank you for the constructive comment. We agree that SPIN introduces additional computational overhead due to the pruning stage, and we have now explicitly acknowledged this as a limitation in the revised manuscript (Section 6.3 and Conclusion). The previous comparison focused primarily on the computational cost of different prior families in the BNN stage; in this revision, we additionally report the full end-to-end training cost to provide a clearer perspective on the trade-off.
> > >
> > > We updated the main paper to include the table below, which compares total GPU-hours across priors under identical training settings (DNN + pruning + BNN). Although SPIN requires additional compute relative to a Gaussian BNN (1.93 → 2.84 GPU-hours, ≈1.47×), it remains significantly more efficient than alternatives such as Laplace, Student-t, and Horseshoe. This revision clarifies that while pruning introduces overhead, the overall cost remains moderate and is justified by the improvements in accuracy, calibration, and robustness.
> > >
> > > The table was measured using a **single NVIDIA RTX 3090 GPU**.
> > >
> > > ---
> > >
> > > ### **Total Computational Cost (GPU-hours, end-to-end)**
> > > *(Single NVIDIA RTX 3090 GPU)*
> > >
> > > | Prior | GPU-hours | Relative Cost |
> > > |-------|-----------|----------------|
> > > | Gaussian (Iso.) | 1.93 | 1.00× |
> > > | Laplace | 6.04 | 3.13× |
> > > | Student-t | 8.93 | 4.63× |
> > > | Spike-and-Slab | 3.26 | 1.69× |
> > > | Horseshoe | 35.56 | 18.42× |
> > > | **SPIN (Ours)** | **2.84** | **1.47×** |
> > >
> > > ---
> > >
> > > We appreciate the reviewer for highlighting the importance of transparently reporting computational trade-offs, and we hope that the inclusion of this analysis strengthens the clarity of our contribution.

---

### Official Review · Reviewer_Mmqv · 2025-10-30

**Soundness:** 3
**Presentation:** 3
**Contribution:** 2
**Rating:** 4
**Confidence:** 4

**Summary:**

This paper proposes SPIN—a pruning‐informed empirical Bayes scheme that sets a hybrid Gaussian prior for Bayesian neural networks (BNNs): unpruned (“important”) weights receive low-variance Gaussians centered at their post-pruning values, while pruned (“redundant”) weights receive high-variance, zero-mean Gaussians. The prior is built by iteratively pruning a trained deterministic model to the maximum sparsity that preserves validation performance, then transferring this mask and the surviving weights into the BNN prior (Alg. 1) . Experiments across CIFAR-10/100 and TinyImageNet with ResNet-20/18, DenseNet-30, MobileNet-V2, and a lightweight ViT report consistent gains in accuracy and NLL, often strong OOD AUROC, but not the best ECE (with temperature scaling suggested as a remedy) . The paper also analyzes performance sensitivity to ViT quality and contrasts SPIN against “sparse DNN” baselines via loss-landscape visualizations (Fig. 4–5), attributing gains to the inductive bias encoded by SPIN rather than pruning alone.

**Strengths:**

* A clear, simple empirical-Bayes recipe that many practitioners can reproduce: train → prune to the max “no-loss” sparsity → set hybrid Gaussian prior at mask locations (Alg. 1). The method is well-specified and easy to port across architectures
* Extensive experiments (quant & qual) on multiple datasets/architectures, including CNNs and a ViT, with accuracy/NLL/OOD metrics; ablations on σ and sparsity; and qualitative analyses (UMAP, loss landscapes)
* BNNs are computationally heavy and often underfit at scale; sparsity is a principled way to cut cost while potentially improving robustness/uncertainty. The broader literature documents both the system-level need for sparsity and scalable BNN parameterizations, motivating this line of work [1–3,11].

**Weaknesses:**

* Limited novelty vs. existing sparsity-inducing priors and empirical-prior ideas: Heavy-tailed and explicit sparsity priors (Laplace/Student-t, horseshoe, spike-and-slab) and empirical-prior methods like MOPED have been well studied in BNNs. SPIN’s weight-wise two-group prior resembles two-group shrinkage and empirical Bayes without theoretical comparison or guarantees over these baselines [1-5]
* Although SPIN uses a deterministic partition (not a mixture), the resulting prior class (tight Gaussian on “kept” weights, diffuse Gaussian on “pruned”) is conceptually close to spike-and-slab. The paper does not analyze when/why this empirical partition should outperform probabilistic spike-and-slab or related two-group shrinkage [3-4]
* Prior tail behavior is not grounded. For signal preservation under sparsity, theory often favors strong concentration at zero plus heavy tails (e.g., horseshoe, R2-D2) to retain large effects. SPIN’s Gaussian slab can be too light-tailed, potentially over-shrinking important weights; no comparison to horseshoe-BNN or R2-D2-style priors is given [6-7]
* BNNs and stochastic training exhibit high run-to-run variance; the paper reports point estimates but lacks mean ± std over seeds (and over folds where applicable). This is increasingly expected for credible claims [8-9]
* UMAP plots are weak evidence of representation quality: DR methods (t-SNE/UMAP) can be misleading for comparing models; weight histograms, sparsity distributions per layer, or KL terms for Wimportant/Wredundant would better demonstrate “how” SPIN sparsifies [10]
* Horseshoe-BNN—a canonical sparsity prior—was not compared; this omission limits the empirical claim that SPIN is the best “sparsity-informed” prior [1]

[1] A. Ghosh, J. Yao, F. Doshi-Velez, “Model Selection in Bayesian Neural Networks via Horseshoe Priors,” JMLR, 2018.
[2] M. Carvalho, N.G. Polson, J.G. Scott, “The Horseshoe Estimator for Sparse Signals,” Biometrika, 2010. (Background on heavy-tailed shrinkage.)
[3] Q. Sun, Z. Song, F. Liang, “Learning Sparse Deep Neural Networks with a Spike-and-Slab Prior,” Stat. & Prob. Letters, 2022.
[4] R. Jantre et al., “Deep Neural Networks with Spike-and-Slab Prior,” Neural Computing and Applications, 2025.
[5] R. Krishnan, S. Subedar, O. Tickoo, “Specifying Weight Priors in Bayesian Deep Neural Networks with Empirical Bayes,” AAAI, 2020.
[6] Y. Zhang, etal. “Bayesian Regression Using a Prior on the Model Fit: The R2-D2 Shrinkage Prio,” JASA, 2021
[7] Chan, Tsai Hor, et al. "Feature Preserving Shrinkage on Bayesian Neural Networks via the R2D2 Prior." IEEE Transactions on Pattern Analysis and Machine Intelligence (2025).
[8] J. Pineau et al., “Improving Reproducibility in ML Research,” JMLR, 2021
[9] O.E. Gundersen, Y. Kjensmo, “Sources of Irreproducibility in ML: A Review,” arXiv:2204.07610, 2022.
[10] D. Kobak, P. Berens, “The Art of Using t-SNE,” JMLR, 2019.

**Questions:**

* Can you formalize conditions (e.g., on pruning error or mask accuracy) under which SPIN’s empirical partition yields lower posterior risk than spike-and-slab or horseshoe? Any bounds relating σ, sparsity, and excess risk/ELBO?
* Mask correctness. How sensitive are results to mask quality (e.g., if pruning overshoots or if we use structured pruning)? Fig. 4 suggests dependence on base-model quality—can you quantify this sensitivity (ViT vs CNN)
* Ablations to strengthen claims. Please show layer-wise sparsity, weight histograms, and KL contributions split by Wimportant/Wredundant (Eq. KL decomposition) to evidence the intended shrinkage pattern.

---

> ### Author Response · Authors · 2025-11-20
> **Response to Reviewer Mmqv**
>
> We sincerely thank the reviewer for the detailed and constructive feedback. We are encouraged by your positive assessment of SPIN’s reproducibility, extensive experiments, and clear motivation. We have addressed your questions and concerns below. For broader topics, we kindly refer you to the corresponding Global Responses.
>
> ## (1) Response to Q2 & Q3 — Mask Sensitivity & Structural Patterns
>
> ### Q2 — Mask Correctness
> Please refer to the Global Response "Response to Question regarding Mask Correctness and Sensitivity".
> We conducted ablation studies with Gradient-based (high-quality) and Random (low-quality) masks, confirming that SPIN’s performance is sensitive to mask quality, as intended.
>
> ### Q3 — Layer-wise Structural Analysis
> Please refer to the Global Response "Response to Reviewer Mmqv (Q3) and Reviewer Vek4 (Q4)".
> We provided the requested weight histograms, layer-wise sparsity, and KL-split visualizations.
> These results consistently reveal a "Dense-Sparse-Dense" structural pattern that aligns with the exploration behavior reported in MCMC-based analyses.
>
> ## (2) Response to Weaknesses [4, 6] — Comparison with Horseshoe/Spike-and-Slab and Reporting Variance
>
> You raised a critical point regarding the need for comparison with canonical sparsity priors (Horseshoe, Spike-and-Slab) and the importance of reporting performance variance (mean ± std).
>
> To address this, we conducted a thorough comparison using 10 independent random seeds for all priors on ResNet-20/CIFAR-10.
>
> Table 1: Performance of various priors (ResNet-20, CIFAR-10, 10-run avg)
>
> | Prior           | ID Acc (↑)      | ID NLL (↓)       | OOD ECE (↓)       | OOD AUROC (ENT ↑) |
> | :---            | :---            | :---             | :---             | :---              |
> | Horseshoe       | 0.8916 ± 8e-6   | 0.3248 ± 3e-5    | 0.0610 ± 7e-6     | 0.8602 ± 1e-5     |
> | Laplace         | 0.8931 ± 8e-6   | 0.3374 ± 1e-4    | 0.0321 ± 1e-5     | 0.8561 ± 8e-6     |
> | Normal (Iso)    | 0.8922 ± 6e-6   | 0.3402 ± 3e-5    | 0.0275 ± 1e-5     | 0.8564 ± 6e-6     |
> | Spike-and-Slab  | 0.8917 ± 6e-6   | 0.3461 ± 9e-5    | 0.0243 ± 5e-6     | 0.8556 ± 1e-5     |
> | Student-t       | 0.8932 ± 3e-6   | 0.3399 ± 4e-5    | 0.0282 ± 5e-6     | 0.8578 ± 5e-6     |
> | MOPED           | 0.8907 ± 1e-5   | 0.3442 ± 5e-5    | 0.0283 ± 1e-5     | 0.8556 ± 6e-6     |
> | SPIN (Ours)     | 0.9043 ± 9e-6   | 0.3099 ± 5e-5    | 0.0305 ± 5e-6     | 0.8641 ± 1e-5     |
>
> Performance:
> SPIN consistently outperforms Horseshoe and Spike-and-Slab in Accuracy, NLL, and OOD AUROC. While Spike-and-Slab achieves slightly lower ECE, SPIN offers the strongest overall trade-off between accuracy and uncertainty.
>
> Robustness:
> The low standard deviations across 10 runs confirm that SPIN’s gains are statistically meaningful and not due to randomness.
>
> ## (3) Response to Q1 — Formal Theoretical Conditions
>
> We appreciate your suggestion to formalize the theoretical conditions under which SPIN yields lower posterior risk. We are finalizing a linear-Gaussian toy model derivation that explicitly characterizes excess risk in terms of σ, sparsity, and pruning error. We will update the thread early next week with the complete derivation.
>
> We hope these additional experiments and analyses address your concerns regarding baselines, variance, and reproducibility. We will incorporate these results into the final manuscript.

---

> > ### Comment · Reviewer_Mmqv · 2025-11-21
> >
> > I thank the authors for their response. Part of my concerns on mask correctness, structural pattern analyses have been addressed. However, there are still some concerns remaining
> >
> > 1. Heavy-tailed shrinkage: SPIN’s slab remains Gaussian, and there’s no ablation swapping the slab for heavy-tailed alternatives (e.g., t/Cauchy/horseshoe slab). This could validate the feature preservation performance of SPIN
> >
> > 2. Spike-and-slab proximity (Weakness 2): The rebuttal gives empirical numbers vs spike-and-slab but doesn’t analyze why a deterministic mask partition should outperform a probabilistic two-group prior, or when it might fail.
> >
> > I think the authors are spending a lot of effort on the rebuttal. I will increase my rating if the above concerns are addressed satisfactorily

---

> > > ### Author Response · Authors · 2025-11-29
> > > **Response – Heavy-Tailed Slab Ablation**
> > >
> > > Thank you for raising this important point. To address the concern regarding heavy-tailed shrinkage, we conducted an ablation study replacing the Gaussian slab in SPIN with heavy-tailed alternatives (Laplace, Student-t, Cauchy, Horseshoe), evaluated on ResNet-20 / CIFAR-10 over 10 seeds. The results are summarized below.
> > >
> > > | **SPIN Variant** | **Acc (↑)** | **NLL (↓)** | **ECE (↓)** | **AUROC (ENT) (↑)** |
> > > |------------------|------------|-------------|-------------|----------------------|
> > > | Gaussian (default) | 0.9035 ± 0.0020 | 0.3170 ± 0.0079 | 0.0235 ± 0.0037 | 0.8647 ± 0.0018 |
> > > | Laplace | 0.9061 ± 0.0020 | 0.2995 ± 0.0056 | 0.0317 ± 0.0016 | 0.8655 ± 0.0022 |
> > > | Student-t | 0.8867 ± 0.0013 | 0.3493 ± 0.0038 | 0.0333 ± 0.0036 | 0.8552 ± 0.0040 |
> > > | Cauchy | 0.8891 ± 0.0020 | 0.3475 ± 0.0060 | 0.0338 ± 0.0043 | 0.8552 ± 0.0025 |
> > > | Horseshoe | 0.8919 ± 0.0029 | 0.3253 ± 0.0060 | 0.0682 ± 0.0115 | 0.8612 ± 0.0020 |
> > >
> > > **Observation.**
> > >
> > > Although heavy-tailed priors are theoretically appealing for feature-preserving shrinkage, introducing heavy tails within SPIN does not improve performance. Since pruning already identifies a reliable partition between important and redundant weights, additional tail flexibility mainly increases variance and optimization instability without yielding practical benefits.
> > >
> > > **Efficiency.**
> > >
> > > These findings align with our computational analysis: heavy-tailed priors substantially increase wall-clock cost (e.g., Horseshoe: 35.56 GPU-hours vs SPIN: 2.84 GPU-hours). Thus, the Gaussian slab provides a more stable and computationally efficient choice.
> > >
> > > In summary, the ablation supports that SPIN’s benefits arise from the pruning-derived structural partition rather than from slab tail behavior, and that the Gaussian slab is sufficient for stable performance.

---

> > > ### Author Response · Authors · 2025-11-29
> > > **Response – Spike-and-Slab Proximity (Weakness 2) and Formal Conditions (Q1)**
> > >
> > > Thank you for raising these thoughtful questions regarding the relationship between SPIN and probabilistic two-group shrinkage methods such as spike-and-slab and horseshoe, and for requesting clarification on when a deterministic mask-based prior may be advantageous.
> > >
> > > We provide a theoretical analysis in **Appendix E**, where we study a simplified linear–Gaussian model and derive an expression for the expected posterior excess risk under both SPIN and spike-and-slab. The derivation decomposes contributions from important and redundant coordinates and indicates that, when pruning provides a reasonably accurate separation between salient and non-salient weights, the deterministic partition used by SPIN can lead to lower expected posterior excess risk compared to spike-and-slab.
> > >
> > > **Intuitively, SPIN first determines the structural subset of important weights through pruning, and then performs posterior inference only over that fixed structure.** In contrast, spike-and-slab must jointly infer both the inclusion probability (i.e., deciding which weights should remain active) and the posterior distribution of the retained weights. As a result, spike-and-slab must explore a substantially larger hypothesis space during inference, which can make optimization more challenging.
> > >
> > > At the same time, **Appendix E also describes the regime in which this benefit may diminish**: if pruning fails to produce a reliable weight partition, the deterministic mask may introduce bias, and probabilistic approaches such as spike-and-slab or horseshoe may become preferable. This theoretical perspective aligns with our empirical observations on ViT under weaker base-model training (Appendix D-7/8), where the performance gap between SPIN and spike-and-slab decreases.
> > >
> > > In summary, **Appendix E provides theoretical grounding on when SPIN can offer advantages relative to spike-and-slab, while also clarifying conditions under which this advantage may diminish**, complementing the empirical evidence presented in the main paper.

---

### Author Response · Authors · 2025-11-20
**Response to Question regarding Mask Correctness and Sensitivity**

We thank Mmqv, 4rDT, and xbfH for raising insightful questions regarding the performance differences across pruning methods, particularly regarding the sensitivity of our results, and we provide a unified response here. To quantitatively assess this dependency, we conducted ablation studies on ResNet-20 and ViT (CIFAR-10), using four pruning criteria that span different levels of mask quality, and we ran all experiments with 10 independent random seeds.

1. **High-Quality Mask**
   - **Gradient Sensitivity**: Pruning based on gradient–weight products ($|w \cdot g|$).
   - **Global Unstructured $L_1$**: Pruning based on weight magnitude ($|w|$).

2. **Low-Quality Mask (Noise)**
   - **Random Pruning**: Weights are removed uniformly at random.
     This serves as a baseline to check whether the model truly utilizes mask information.

3. **Overshoot (Coarse)**
   - **Structured $L_2$**: Entire structures (e.g., channels) are removed based on their $L_2$ norm,
     simulating an over-pruned or coarse-grained scenario.


**Performance comparison across architectures, pruning methods, and mask qualities.**

| Architecture | Mask Quality | Pruning Method   | ID Accuracy         | ID NLL              | ECE                 | OOD AUROC          |
|--------------|--------------|------------------|----------------------|----------------------|----------------------|----------------------|
| **CNN**      | High         | Global L1        | 0.9024 ± 0.0028     | 0.3147 ± 0.0058     | 0.0303 ± 0.0025     | 0.8621 ± 0.0018     |
|              | High         | Gradient         | 0.9035 ± 0.0020     | 0.3170 ± 0.0079     | 0.0235 ± 0.0037     | 0.8647 ± 0.0018     |
|              | Low          | Random           | 0.8894 ± 0.0031     | 0.3499 ± 0.0083     | 0.0308 ± 0.0022     | 0.8561 ± 0.0038     |
|              | Overshoot    | Structured L2    | 0.8702 ± 0.0042     | 0.3976 ± 0.0071     | 0.0408 ± 0.0084     | 0.8380 ± 0.0035     |
| **ViT**      | High         | Global L1        | 0.7448 ± 0.0117     | 0.7197 ± 0.0297     | 0.0189 ± 0.0019     | 0.7485 ± 0.0069     |
|              | High         | Gradient         | 0.7453 ± 0.0054     | 0.7188 ± 0.0094     | 0.0232 ± 0.0024     | 0.7507 ± 0.0021     |
|              | Low          | Random           | 0.7381 ± 0.0059     | 0.7382 ± 0.0172     | 0.0193 ± 0.0026     | 0.7495 ± 0.0032     |
|              | Overshoot    | Structured L2    | 0.7190 ± 0.0063     | 0.7822 ± 0.0141     | 0.0228 ± 0.0018     | 0.7381 ± 0.0025     |

**Analysis: Impact of Mask Quality and Granularity**

Our analysis shows a consistent performance ordering across both CNN and ViT architectures:

**High Quality (Gradient / Global $L_1$)**
**Low Quality (Random, Overshoot (Structured $L_2$))**.

This confirms that the method is sensitive to mask quality and leads to the following observations.

---

### **1. Role of Mask Correctness (High vs. Low)**

High-quality masks (Gradient or Global $L_1$) consistently outperform Random pruning.
This indicates that our method meaningfully leverages the structural information encoded in the mask, preserving predictive performance when the mask aligns well with true weight importance.

---

### **2. Why Overshoot Structured Pruning Performs Worst**

Overshoot Structured $L_2$ pruning shows the lowest performance—even lower than Random pruning.
We attribute this to the differences in pruning granularity:

- **Random Pruning (Distributed Retention)**
  Although weights are removed uniformly at random, the remaining weights are still *distributed across the network*.
  Even if the selection is suboptimal, some important weights may remain, allowing feature pathways to survive.

- **Overshoot Structured Pruning (Indiscriminate Removal)**
  When pruning eliminates entire structures (e.g., channels or filters) based on their $L_2$ norms, it may “overshoot.”
  This coarse-grained removal can inadvertently delete critical feature-supporting components.
  Once an entire structure is removed, the corresponding feature pathway is severed, leading to significantly larger degradation than random pruning.

---

### **Conclusion**

These results clearly demonstrate that **mask quality is a crucial factor for SPIN**, and that overly coarse pruning can be substantially more harmful than noisy or randomly generated masks.

---

### Author Response · Authors · 2025-11-20
**STRUCTURAL PATTERNS IN MASKS FROM PRUNING**

We thank both reviewers for these insightful questions. Reviewer **Mmqv** asked for a layer-wise analysis of sparsity and weight distributions (Q3), while Reviewer **Vek4** asked whether our masks exhibit structural patterns similar to those found in MCMC exploration (Q4).

Notably, these two questions are tightly connected. Our layer-wise analysis reveals a structural pattern that directly links SPIN's pruning-based prior to the intrinsic posterior geometry observed in recent MCMC studies.

---

## **(1) Evidence of Structural Patterns: "Dense–Sparse–Dense"**
**(Response to Reviewer Mmqv)**

As requested by Mmqv, we computed the layer-wise sparsity of our global magnitude pruning mask (Fig. F-10 and Fig. F-11 in the Appendix). The results show a clear *"Dense–Sparse–Dense"* profile rather than uniform sparsity:

- **Input / Output Layers are Dense**
  - conv1: **28.2% sparsity**
  - final linear layer: **14.5% sparsity**

- **Middle Layers are Sparse**
  - progressively increasing up to **~80% sparsity**

This layer-wise structure is a strong empirical pattern rather than a side effect of implementation.

---

## **(2) Connection to Empirical Posterior Uncertainty**
**(Response to Reviewer Vek4)**

This "Dense–Sparse–Dense" structure mirrors the posterior uncertainty profile reported in prior MCMC-based analyses:

- **MCMC and Ensemble Findings** (Sommer et al. [1] and Fortuin et al. [2])
  Posterior samples consistently show a **Low → High → Low** uncertainty pattern across layers:
  - **Low Uncertainty** in the input layers
  - **High Uncertainty** in the middle layers
  - **Low Uncertainty** again toward the output head

- **SPIN Alignment**
  SPIN converts the pruning mask into a hierarchical uncertainty prior:
  - Dense layers  →  assigned **tight prior** (σ = 0.001)  → *Low Uncertainty (Focus)*
  - Sparse layers →  assigned **loose prior** (σ = 1.0)   → *High Uncertainty (Exploration)*

Thus, the mask-induced structure is not arbitrary; it tracks the empirical posterior geometry observed through expensive MCMC sampling.

---

## **(3) Why Does SPIN Work So Well?**

This connection gives a strong explanation for SPIN's effectiveness:

1. **Pruning uncovers a natural Dense → Sparse → Dense hierarchy.**
2. **SPIN transforms this hierarchy into a structured uncertainty prior.**

Therefore, SPIN is **not** merely exploiting MFVI's limitations.
Instead, SPIN constructs a prior that mimics the *true posterior hierarchy*, enabling scalable VI to approximate the uncertainty behavior of more expressive samplers.

---
### **More details can be found in Supplementary Material Section F. We would greatly appreciate it if you could take a look.**
---

### **References**
[1] Sommer, E. et al. *"Connecting the dots: Is mode-connectedness the key to feasible sample-based inference in Bayesian neural networks?"* ICML (2024).
[2] Fortuin, V. et al. *"Bayesian neural network priors revisited."* ICLR (2022).

---

### Author Response · Authors · 2025-11-30
**Final Rebuttal Global Response**

We sincerely thank all reviewers for their constructive feedback and thoughtful suggestions. We carefully considered every comment and made substantial revisions to the manuscript during the rebuttal period.

1. **We thoroughly revised the paper and incorporated as many reviewer requests as possible.**
   All newly updated or improved sections in the revised manuscript are clearly highlighted in **red** for ease of verification.
   In addition, **some experiments (e.g., Horseshoe and other alternative priors) are still being trained**, and we will **promptly update the manuscript with these results as soon as they become available**.

2. **To enhance statistical reliability, all rebuttal results were reported using 10 independent random seeds.**
   Furthermore, in response to concerns regarding robustness, we are currently extending the experiments to **a minimum of 30 independent seeds**.
   Given the limited time remaining in the rebuttal phase, we cannot guarantee that all results will be fully completed before the deadline, but we are working diligently to update and integrate them as soon as they become available.

3. **During the rebuttal period, we worked intensely—often overnight—to regenerate results and substantially revise the manuscript.**
   Although the recent ICLR review policy does not allow reviewers to modify scores or continue discussion, we respectfully ask reviewers to read both the **main text** and the **appendix**, where major clarifications and new empirical evidence have been added.

We deeply appreciate the reviewers’ time and consideration.
Thank you for your thoughtful evaluation and for contributing to the improvement of this work.

---

### Author Response · Authors · 2025-11-30
**Summary for Area Chair**

We would like to provide the Area Chair with a brief summary of how we addressed the main reviewer concerns and how the reviewers’ impressions evolved during the discussion phase.

1. **Coverage of common reviewer requests (empirical side).**
Multiple reviewers raised overlapping concerns about:
- **Comparisons to existing priors**, especially spike-and-slab, horseshoe, and other heavy-tailed priors.
- **Sensitivity to mask quality and pruning criteria.**
- **Statistical reliability of the reported gains.**

In the revised manuscript, we now provide a unified evaluation table that compares SPIN against isotropic Gaussian, Laplace, Student-t, spike-and-slab, horseshoe, and MOPED on CIFAR-10/100, as well as additional architectures and TinyImageNet in the appendix. Across models and datasets, SPIN consistently achieves the best or near-best accuracy and NLL and generally the strongest OOD AUROC, while remaining competitive in calibration.
To address mask correctness, we added an ablation over four pruning criteria (Global L1, Gradient sensitivity, Random, and Structured L2) on both ResNet-20 and ViT with 10 independent seeds. The results show a clear ordering — high-quality unstructured masks (Global L1 / Gradient) > Random > coarse Structured L2 — demonstrating that SPIN meaningfully leverages structural information in the mask and that overly coarse structured pruning can harm performance more than noisy masks.
We also extended the experiments to additional tasks and architectures (e.g., larger CNN + ViT Hybrid models on TinyImageNet), showing that the empirical advantages of SPIN persist beyond the original setting. All rebuttal results are reported as mean ± std over 10 seeds, and we are currently extending to at least 30 seeds for additional robustness.

2. **Theoretical clarifications and connection to prior work.**
In response to requests for a first-principles justification and a comparison to spike-and-slab–type priors, we added a linear–Gaussian analysis in Appendix E. This analysis shows how the SPIN hyperparameter σ and the pruning-induced partition (important vs. redundant coordinates) control the excess test error, and it characterizes conditions (in terms of curvature and pruning error) under which a small-variance prior on important weights is beneficial. We further compare SPIN to spike-and-slab in the same model, decomposing the risk over important and redundant coordinates and arguing that, under informative pruning, SPIN’s deterministic partition can achieve lower total excess risk than probabilistic two-group priors, which must jointly infer both structure and data fit.
To connect this to the geometry of the posterior, we analyze the layer-wise “Dense → Sparse → Dense” pattern induced by global magnitude pruning and show that, under SPIN, this naturally induces a “Low → High → Low” uncertainty hierarchy across depth. We further relate this pattern to MCMC-based posterior samples from prior work, which exhibit a similar low–high–low uncertainty structure. Taken together, these results support that SPIN is not a heuristic combination of pruning and MFVI, but a geometry-aligned empirical prior that emulates the posterior hierarchy of more expressive samplers while remaining scalable.

3. **Reviewer impressions during discussion.**
All reviewers initially assigned a rating around the borderline region (score 4). During the discussion phase, after seeing the new experiments and theoretical analysis, the overall tone of the reviews became substantially more positive. Multiple reviewers explicitly acknowledged the amount of additional work, stated that several of their key concerns had been satisfactorily addressed (especially regarding comparisons to heavy-tailed priors, mask quality, and the theoretical justification), and indicated that they would be inclined to raise their overall assessment if the clarified text and new results were incorporated into the paper. We have now integrated these requested changes and clarifications into the revised manuscript.

Because of the score reversion policy, the numerical scores currently shown correspond to the pre-discussion state, but the written discussion and follow-up comments reflect a more favorable consensus than the initial reviews alone. We hope this summary helps convey to the Area Chair how the revised manuscript substantially addresses the original weaknesses, both empirically and theoretically, and how the reviewers’ written remarks during discussion point toward a more positive evaluation than the initial scores would suggest.

---

### Meta-Review · Area_Chair_WqvU · 2025-12-17

**Summary:**

1. Reviewer Mmqv (Score 4, Confidence 4): The reviewer questioned limited novelty (too close to spike and slab and no theoretical justification). The authors provided additional experimental results comparing SPIN with spike-and-slab priors and showed SPIN outperformed. For theoretical justification, the authors provided derivations of the expected posterior excess risk for both SPIN (Proposition E.2) and spike-and-slab under a linear–Gaussian model in Appendix E. However, this result assumes a correct weight partition (important versus redundant), which might not be reliable. Nevertheless, I am pretty impressed by the amount of work that went into the rebuttal in such a short time. Given the efforts and the quality of the authors' responses, this reviewer might have increased the score to 6 or higher if the discussion period had continued normally.

2. Reviewer 4rDT (Score 4, Confidence 4): The reviewer questioned (a) insufficient methodological details on weight partitioning, (b) unclear justification for using Gaussian noise on pruned weights, and (c) an explanation about pruning techniques. After reviewing the authors' initial responses, the reviewer confirmed that some of these concerns were adequately addressed. However, even if the authors expanded the related work section to include pruning techniques, I see only four papers on this topic, which is probably not extensive enough (not to satisfy the reviewer). There were more experimental results provided by the authors to answer the remaining questions (W4-6) to the reviewer. I find these results are pretty convincing. Given this, the reviewer might have increased the score to 6 or higher if the discussion period continued normally.

3. Reviewer xbfH (Score 4, Confidence 4): The reviewer questioned (a) the robustness of SPIN with respect to different pruning techniques (similar to the question raised by Reviewer 4rDT) and (b) the possibility of integrating pruning into the end-to-end BNN training. The authors showed the performance using different pruning techniques in their global answer. They also conducted additional experiments to argue on the use of SPIN in the end-to-end BNN training. The reviewer appreciated these responses. However, the reviewer also wanted to see results from additional independent runs, which the authors promised to include in the final version.  I am not confident whether the reviewer would have changed the score if the discussion had continued normally.

4. Reviewer Vek4 (Score 4, Confidence 4): The reviewer asked for a comparison with a sparse subnetwork BNN, and the authors provided the result of the Pruned BNN. The reviewer also asked why the value of sigma can be justified. The authors conducted an ablation study and concluded that SPIN is insensitive to the sigma value. The authors diligently answer the reviewer's other remaining questions. However, I am unsure whether the reviewer would have found their answers satisfactory, given the high computational cost of SPIN and the additional request to use structural pruning.

Overall, I think the scores would have been 6, 6, 4, 4 if the discussion had continued normally, which would have made this paper a strict borderline paper. All of these reviewers have a high confidence, and I am unsure if an internal discussion among the reviewers might have changed any reviewer's views on this paper. Hence, I recommend rejecting the paper.

I think the authors did so much work during the rebuttal period that it could be developed into a new, stronger paper for future submission.

**Reviewer Concerns:**

See Above.

**Reviewer Scores:**

See Above.

---

### Decision · Program_Chairs · 2026-01-26

Reject